# An Analytical Theory of Spectral Bias in the Learning Dynamics of Diffusion Models

**Binxu Wang**
Kempner Institute, Harvard University
Boston, MA, USA
binxu_wang@hms.harvard.edu

**Cengiz Pehlevan**
SEAS, Harvard University
Cambridge, MA, USA
cpehlevan@seas.harvard.edu

## Abstract

We develop an analytical framework for understanding how the generated distribution evolves during diffusion model training. Leveraging a Gaussian-equivalence principle, we solve the full-batch gradient-flow dynamics of linear and convolutional denoisers and integrate the resulting probability-flow ODE, yielding analytic expressions for the generated distribution. The theory exposes a universal inverse-variance spectral law: the time for an eigen- or Fourier mode to match its target variance scales as $\tau \propto \lambda^{-1}$, so high-variance (coarse) structure is mastered orders of magnitude sooner than low-variance (fine) detail. Extending the analysis to deep linear networks and circulant full-width convolutions shows that weight sharing merely multiplies learning rates—accelerating but not eliminating the bias—whereas local convolution introduces a qualitatively different bias. Experiments on Gaussian and natural-image datasets confirm the spectral law persists in deep MLP-based UNet. Convolutional U-Nets, however, display rapid near-simultaneous emergence of many modes, implicating local convolution in reshaping learning dynamics. These results underscore how data covariance governs the order and speed with which diffusion models learn, and they call for deeper investigation of the unique inductive biases introduced by local convolution.

## 1 Introduction

Diffusion models create rich data by gradually transforming Gaussian noise into signal, a paradigm that now drives state-of-the-art generation in vision, audio, and molecular design [1, 2, 3]. Yet two basic questions remain open. (i) Which parts of the data distribution do these models learn first, and which linger unlearned—risking artefacts under early stopping? (ii) How does architectural inductive bias shape this learning trajectory? Addressing both questions demands that we track the evolution of the full generated distribution during training and relate it to the network's parameterization.

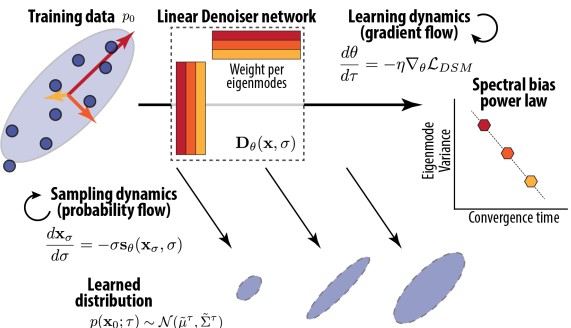

Figure 1: **Spectral-bias schematic.** Learning and sampling together impose a variance-ordered bias along covariance eigenmodes.

We tackle the learning puzzle through the simplest tractable setting—linear denoisers—where datasets become equivalent to a Gaussian with matched mean and covariance. In this regime we solve, in closed form, the nested dynamics of

39th Conference on Neural Information Processing Systems (NeurIPS 2025).

gradient-flow of the weights and the probability-flow ODE that carries noise into data, leading to an analytical characterization of the evolution of the generated distribution. The analysis exposes an inverse-variance spectral law: the time required for an eigen-mode to match target variance scales like $\tau_k \propto \lambda_k^{-\alpha}$, so high-variance directions corresponding to global structure are mastered orders of magnitude sooner than low-variance, fine-detail directions. Extending the analysis to deep linear and linear convolutional nets, we show how convolutional architecture redirect this bias to Fourier or patch space, and acclerate convergence via weight sharing.

**Main contributions** 1. **Closed-form distribution dynamics**. We derive exact weight and distributional trajectories for one-layer, two-layer linear, and convolutional denoisers under full-batch DSM training. 2. **Inverse-variance spectral bias**. The theory reveals and quantifies a spectral-law ordering of mode convergence, offering one mechanistic explanation for early-stop errors. 3. **Empirical validation in nonlinear neural nets**. Experiments on Gaussian and natural-image datasets confirm the spectral-law in deep MLP-based diffusion. 4. **Convolutional architectural shape learning dynamics**. Experiments on convolutional UNet, showing rapid patch-first learning dynamics different from fully-connected architectures.

## 2 Related Work and Motivation: Spectral Bias in Distribution Learning

**Spectral structure of natural data** Many natural signals have interesting spectral structures (e.g. image [4], sound [5], video [6]). For natural images, their covariance eigenvalues decay as a power law, and the corresponding eigenvectors can align with semantically meaningful patterns [4]. For faces, for instance, leading eigenmodes capture coarse, low-frequency shape variations, whereas tail modes encode fine-grained textures [7, 8]. Analyzing spectral effect on diffusion learning can therefore show which type of features the model acquires first and which remain slow to learn.

**Hidden Gaussian Structure in Diffusion Model** Recent work has shown, for most diffusion times, the learned neural score is closely approximated by the linear score of a Gaussian fit to the data, which is usually the best linear approximation [9, 10]. Crucially, this Gaussian linear score admits a closed-form solution to the probability-flow ODE, which can be exploited to accelerate sampling and improve its quality [11]. Moreover, this same linear structure has been linked to the generalization–memorization transition in diffusion models [10]. In sum, across many noise levels, the Gaussian linear approximation is a predominant structure in the learned score. Thus, we hypothesize it will have a significant effect on the learning dynamics of score approximator. From this perspective, **our contribution** is to elucidate the learning process of this linear structure.

**Learning theory for regression and deep linear networks** Gradient dynamics in regression are well-studied, with spectral bias and implicit regularisation emerging as central themes [12, 13, 14]. In Sec. 4.1, we show that the loss of a linear diffusion model reduces to ridge regression, letting us import those results directly. Our analysis also builds on learning theory of deep linear networks (including linear-convolutional and denoising autoencoders) [15, 16, 17, 18]. We extend these insights to modern diffusion-based generative models, offering closed-form description of how the generated distribution itself evolves during training.

**Diffusion learning theory** Several recent theory studies address diffusion models from a spectral perspective but tackle different questions. [19, 20, 21, 22] document spectral bias in the *sampling* process after training; our focus is on how that bias arises during *training*. [23] study stochastic sampling assuming an optimal score, orthogonal to our analysis of training dynamics. Sharing our interest in training, [24] analyze learning of mixtures of *spherical* Gaussians to recover component means, whereas we tackle *anisotropic* covariances and track reconstruction of the full covariance. [25] characterises optimal score and distribution under constraints; results from our convolutional setup can be viewed through that lens.

## 3 Background

### 3.1 Score-based Diffusion Models

Let $p_0(\mathbf{x})$ be the data distribution of interest, and for each noise level $\sigma > 0$ define $p(\mathbf{x}; \sigma) = \left(p_0 * \mathcal{N}(0, \sigma^2 \mathbf{I})\right)(\mathbf{x}) = \int p_0(\mathbf{y}) \mathcal{N}(\mathbf{x} \mid \mathbf{y}, \sigma^2 \mathbf{I}) \, d\mathbf{y}$. The associated *score function* is $\nabla_{\mathbf{x}} \log p(\mathbf{x}; \sigma)$,

i.e. the gradient of the log–density at noise scale $\sigma$. In the EDM framework [26], one shows that the "probability flow" ODE

$$\frac{d\mathbf{x}}{d\sigma} = -\sigma \nabla_{\mathbf{x}} \log p(\mathbf{x}; \sigma) \tag{1}$$

exactly transports samples from $p(\,\cdot\,; \sigma_T)$ to $p(\,\cdot\,; \sigma)$ as $\sigma$ decreases. In particular, integrating from $\sigma_T$ down to $\sigma = 0$ recovers clean data samples from $p_0$. We adopt the EDM parametrization for its notational simplicity; other common diffusion formalisms are equivalent up to simple rescalings of space and time [26]. To learn the score of a data distribution $p_0(\mathbf{x})$, we minimize the denoising score matching (DSM) objective [27] with a function approximator. We reparametrize the score function with the 'denoiser' $\mathbf{s}_\theta(\mathbf{x}, \sigma) = (\mathbf{D}_\theta(\mathbf{x}, \sigma) - \mathbf{x})/\sigma^2$, then at noise level $\sigma$ the DSM objective reads

$$\mathcal{L}_\sigma = \mathbb{E}_{\mathbf{x}_0 \sim p_0,\, \mathbf{z} \sim \mathcal{N}(0, \mathbf{I})} \left\| \mathbf{D}_\theta(\mathbf{x}_0 + \sigma \mathbf{z}; \sigma) - \mathbf{x}_0 \right\|_2^2. \tag{2}$$

To balance the loss and importance of different noise scales, practical diffusion models all adopt certain weighting functions in their overall loss $\mathcal{L} = \int_\sigma d\sigma \; w(\sigma) \, \mathcal{L}_\sigma$.

### 3.2 Gaussian Data and Optimal Denoiser

To motivate our linear score approximator set up, it is useful to consider the optimal score and the denoiser of a Gaussian distribution. For Gaussian data $\mathbf{x}_0 \sim \mathcal{N}(\boldsymbol{\mu}, \boldsymbol{\Sigma}), \mathbf{x}_0 \in \mathbb{R}^d$ and $\boldsymbol{\Sigma}$ is a positive semi-definite matrix. When noising $\mathbf{x}_0$ by Gaussian noise at scale $\sigma$, the corrupted $\mathbf{x}$ satisfies $\mathbf{x} \sim \mathcal{N}(\boldsymbol{\mu}, \boldsymbol{\Sigma} + \sigma^2 \mathbf{I})$, for which the *Bayes-optimal* denoiser is an *affine function* of $\mathbf{x}$.

$$\mathbf{D}^*(\mathbf{x}; \sigma) = \boldsymbol{\mu} + (\boldsymbol{\Sigma} + \sigma^2 \mathbf{I})^{-1} \boldsymbol{\Sigma} (\mathbf{x} - \boldsymbol{\mu}) \tag{3}$$

For Gaussian data, minimizing (2) yields $\mathbf{D}^*$. This solution has an intuitive interpretation, i.e. the difference of the state $\mathbf{x}$ and distribution mean was projected onto the eigenbasis and shrinked mode-by-mode by $\lambda_k/(\lambda_k + \sigma^2)$. Thus, according to the variance $\lambda_k$ along target axis, modes with variance significantly higher than noise $\lambda_k \gg \sigma^2$ will be retained; modes with variance much smaller than noise will be "shrinked" out. Effectively $\sigma^2$ defines a threshold of signal and noise, and modes below which will be removed. This intuition similar to Ridge regression is made exact in Sec. 4.1.

## 4 Learning in Diffusion Models with a Linear Denoiser

**Problem set-up.** Throughout the paper, we assume the denoiser at each noise scale is *linear (affine) and independent across scales*:

$$\mathbf{D}(\mathbf{x}; \sigma) = \mathbf{W}_\sigma \mathbf{x} + \mathbf{b}_\sigma. \tag{4}$$

Since the parameters $\{\mathbf{W}_\sigma, \mathbf{b}_\sigma\}$ are decoupled across noise scales, each $\sigma$ can be analysed independently. Through further parametrization, this umbrella form captures linear residual nets, deep linear nets, and linear convolutional nets (see Sec. 5).

We train on an arbitrary distribution $p_0$ with mean $\boldsymbol{\mu}$ and covariance $\Sigma$ by gradient flow on the *full-batch* DSM loss, i.e. the exact expectation over data *and* noise (2). (In practice, one cannot sample all $\mathbf{z}$ values, but the full-batch limit yields clean closed-form dynamics.)

This setting lets us dissect analytically the role of **data spectrum**, **model architecture** ($\mathbf{W}_\sigma$ parametrisation), and **loss variant** in shaping diffusion learning.

### 4.1 Diffusion learning as ridge regression

**Gaussian equivalence.** For any joint distribution $p(X, Y)$ the quadratic loss

$$\mathcal{L}(\mathbf{W}, \mathbf{b}) = \mathbb{E}_{p(X,Y)} \left\| \mathbf{W} X + \mathbf{b} - Y \right\|^2$$

depends on $p$ only through the first two moments of $(X, Y)$; see App. C.1.1 for proof. Hence a linear denoiser trained on arbitrary $p_0$ interacts with the data *solely* via its mean $\boldsymbol{\mu}$ and covariance $\Sigma$.

**Instance for diffusion.** Under EDM loss (2), the noisy input–target pair is $X = \mathbf{x}_0 + \sigma \mathbf{z}$, $Y = \mathbf{x}_0$, giving $\Sigma_{XX} = \Sigma + \sigma^2 I$, $\Sigma_{YX} = \Sigma$.

**Gradient and optimum.** Differentiating and setting gradients to zero yields

$$\nabla_{\mathbf{W}_\sigma}\mathcal{L}_\sigma = -2\Sigma + 2\mathbf{W}_\sigma(\Sigma + \sigma^2\mathbf{I}) + \nabla_{\mathbf{b}_\sigma}\mathcal{L}_\sigma\,\boldsymbol{\mu}^\top, \qquad \nabla_{\mathbf{b}_\sigma}\mathcal{L}_\sigma = 2\big(\mathbf{b}_\sigma - (\mathbf{I} - \mathbf{W}_\sigma)\boldsymbol{\mu}\big), \quad (5)$$

$$\mathbf{W}_\sigma^* = \Sigma(\Sigma + \sigma^2\mathbf{I})^{-1}, \qquad\qquad\qquad\qquad \mathbf{b}_\sigma^* = (\mathbf{I} - \mathbf{W}_\sigma^*)\boldsymbol{\mu}, \qquad (6)$$

$$\min \mathcal{L}_\sigma = \sigma^2 \operatorname{Tr}\big[\Sigma(\Sigma + \sigma^2\mathbf{I})^{-1}\big].$$

Thus the optimal linear denoiser reproduces the denoiser for the Gaussian approximation of data (3), and its best achievable loss is set purely by the data spectrum.

**Other objectives.** While the main text focuses on the EDM loss (2), we have worked out the gradients, optima, and learning dynamics for several popular variants used in diffusion and flow-matching [28] literature; these results are summarised in Tab. C.4 (derivations in App. C.4).

**Ridge viewpoint.** Because

$$\mathcal{L}_\sigma = \mathbb{E}_{\mathbf{x}\sim p_0}\big\|\mathbf{W}\mathbf{x} + \mathbf{b} - \mathbf{x}\big\|^2 + \sigma^2\|\mathbf{W}\|_F^2,$$

full-batch diffusion at noise scale $\sigma$ is simply auto-encoding with ridge regularisation of strength $\sigma^2$ (App. C.2.1; cf. [29]). We will exploit classic ridge-regression results when analyzing learning dynamics in the following sections.

## 4.2 Weight Learning Dynamics of a Linear Denoiser

With the gradient structure in hand, we solve the full-batch gradient–flow ODE,

$$\frac{d\mathbf{W}_\sigma}{d\tau} = -\eta\nabla_{\mathbf{W}_\sigma}\mathcal{L}_\sigma, \qquad \frac{d\mathbf{b}_\sigma}{d\tau} = -\eta\nabla_{\mathbf{b}_\sigma}\mathcal{L}_\sigma, \qquad\qquad (\text{GF})$$

where $\tau$ is training time and $\eta$ the learning-rate.

**Zero-mean data ($\boldsymbol{\mu} = 0$): Exponential convergence mode-by-mode** Because the gradients to $\mathbf{W}, \mathbf{b}$ decouple (5), the dynamics is simplified on the eigenbasis of the covariance. We diagonalize the covariance, $\Sigma = \sum_{k=1}^d \lambda_k \mathbf{u}_k \mathbf{u}_k^\top$, with orthonormal principal components (PC) $\mathbf{u}_k$ and eigenvalues $\lambda_k \geq 0$ (the mode variances). Projecting (GF) onto this basis yields the closed-form solution (derivation in App. D.1):

$$\mathbf{b}_\sigma(\tau) = \mathbf{b}_\sigma(0)\,e^{-2\eta\tau}, \qquad \mathbf{W}_\sigma(\tau) = \mathbf{W}_\sigma^* + \sum_{k=1}^d \big[\mathbf{W}_\sigma(0) - \mathbf{W}_\sigma^*\big]\mathbf{u}_k\mathbf{u}_k^\top e^{-2\eta\tau(\sigma^2+\lambda_k)}. \qquad (7)$$

*Interpretation.* Each eigenmode projection of the weight $\mathbf{W}_\sigma\mathbf{u}_k$ converges to the optimal value $\mathbf{W}_\sigma^*\mathbf{u}_k$ exponentially with rate $(\sigma^2 + \lambda_k)$; hence (i) the weights at larger noise $\sigma$ generally converge faster; (ii) at a fixed $\sigma$, high-variance $\lambda_k$ modes converge first, while modes buried beneath the noise floor ($\lambda_k \ll \sigma^2$) share the same slower timescale. Fig. 2A illustrates this spectrum-ordered convergence, with high-variance modes reaching their optima before the low-variance ones (see also 5A).

**Non-centred data ($\boldsymbol{\mu} \neq 0$): Interaction of mean and covariance learning.** A non-zero mean introduces a rank-one coupling between $\mathbf{W}$ and $\mathbf{b}$ (matrix $M$ in Prop D.1). Eigenmodes of weights overlapping with the mean ($\mathbf{u}_k^\top\boldsymbol{\mu} \neq 0$) now interact with $\mathbf{b}$, producing transient overshoots and other non-monotonic effects; orthogonal modes retain the exponential convergnece above. App. D.2 gives the full linear-system analysis and two-dimensional visualisations (Fig. 27).

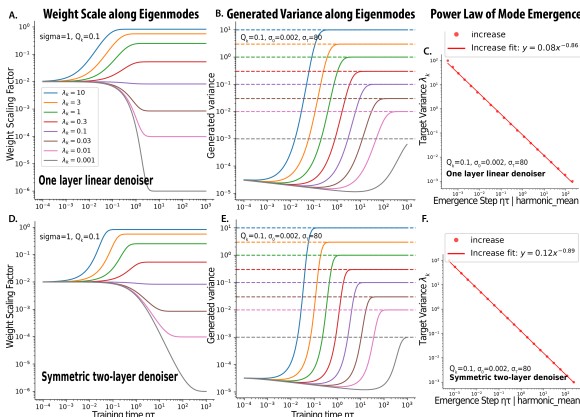

Figure 2: **Learning dynamics per eigenmode.** *Top:* one-layer linear denoiser. *Bottom:* two-layer symmetric denoiser. (A,D) Weight trajectories $\mathbf{u}_k^\top\mathbf{W}_\sigma(\tau)\mathbf{u}_k$ ($\sigma = 1$). (B,E) Generated-variance $\tilde{\lambda}_k$ versus target variance $\lambda_k$. (C,F) Power-law relation between emergence time $\tau_k^*$ and $\lambda_k$.

### 4.3 Sampling Dynamics during Training

For diffusion models, our goal is the generated distribution, obtained by integrating the probability–flow ODE (PF-ODE) backwards from a large $\sigma_T$ to a $\sigma_{min} \approx 0$,

$$\frac{d\mathbf{x}}{d\sigma} = -\sigma^{-1}\big[(\mathbf{W}_\sigma - I)\mathbf{x} + \mathbf{b}_\sigma\big], \tag{PF}$$

initialized with Gaussian noise $\mathbf{x}_T \sim \mathcal{N}(0, \sigma_T^2\mathbf{I})$. For linear denoiser, the PF-ODE is an inhomogeneous *affine* system, so its solution $\mathbf{x}_\sigma$ is necessarily an *affine function* of the initial state $\mathbf{x}_T$ [30], $\mathbf{x}(\sigma_0) = A(\sigma_0; \sigma_T)\,\mathbf{x}(\sigma_T) + c(\sigma_0; \sigma_T)$. Since the map is affine, the distribution of $\mathbf{x}(\sigma_0)$ remains Gaussian, with covariance $\sigma_T^2 A(\sigma_0; \sigma_T) A^\top(\sigma_0; \sigma_T)$.

However, in general, the **state-transition matrix** $A(\sigma_0; \sigma_T)$ is hard to evaluate, as it involves *time-ordered matrix exponential*, and the weight matrices at different noise scales $\mathbf{W}_\sigma$ may *not commute*. The analysis below—and our closed-form results—hinges on situations where *commutativity* is maintained by *gradient flow or architectural bias*, thus removing the time-ordering operator.

**Lemma 4.1** (PF-ODE solution for commuting weights). *If the linear denoiser* $\mathbf{D}(\mathbf{x}; \sigma) = \mathbf{W}_\sigma\mathbf{x} + \mathbf{b}_\sigma$ *satisfies* $[\mathbf{W}_\sigma, \mathbf{W}_{\sigma'}] = 0$ *for all* $\sigma, \sigma'$, *then for any* $0 < \sigma_0 < \sigma_T$,

$$\mathbf{x}(\sigma_0) = A(\sigma_0, \sigma_T)\,\mathbf{x}(\sigma_T) + c(\sigma_0, \sigma_T), \qquad A(\sigma_0, \sigma_T) = \exp\Big[-\int_{\sigma_0}^{\sigma_T} \frac{\mathbf{W}_s - \mathbf{I}}{s}\, ds\Big].$$

*Interpretation.* For each common eigenvector $\mathbf{u}_k$, the term $(\mathbf{u}_k^\top \mathbf{W}_\sigma \mathbf{u}_k - 1)/\sigma$ is the instantaneous expansion (or contraction) rate of the sample variance along $\mathbf{u}_k$; the final variance is obtained by integrating this rate over noise scales $\sigma$ (see App. C.5).

**When does commutativity hold?** This arises in three common settings. (i) *At convergence*, this is satisfied by the optimal weights $\mathbf{W}_\sigma^*$ (6), which jointly diagonalize on eigenbasis of $\Sigma$. In such case, we recover the the closed-form solution to PF-ODE for Gaussian data, as found by [9, 31]. (ii) *During training of linear denoisers,* if weights are initialized to be aligned with eigenbasis of $\Sigma$, then gradient flow keeps them aligned, preserving commutativity (iii) For *linear convolutional denoisers,* circulant weights share the Fourier basis and commute by construction (see Sec. 5.2). In these cases, the sampling process can be understood mode-by-mode. Here we show the explicit solution for one layer linear denoiser.

**Proposition 4.2** (Dynamics of generated distribution in one layer case). *Assume (i) zero-mean data, (ii) aligned initialization* $\mathbf{W}_\sigma(0) = \sum_k Q_k\,\mathbf{u}_k\mathbf{u}_k^\top$, *and (iii) gradient flow, full-batch training with learning rate* $\eta$. *Then, while training the one-layer linear denoiser, the generated distribution at time $\tau$ is* $\mathcal{N}(\tilde{\mu}, \tilde{\Sigma})$ *with* $\tilde{\Sigma} = \sum_k \tilde{\lambda}_k(\tau)\mathbf{u}_k\mathbf{u}_k^\top$ *and*

$$\tilde{\lambda}_k(\tau) = \sigma_T^2\,\frac{\Phi_k^2(\sigma_0, \tau)}{\Phi_k^2(\sigma_T, \tau)}, \quad \Phi_k(\sigma, \tau) = \sqrt{\lambda_k + \sigma^2}\,\exp\Big[\tfrac{1-Q_k}{2}\,\mathrm{Ei}\big(-2\eta\tau\sigma^2\big)e^{-2\eta\tau\lambda_k} - \tfrac{1}{2}\,\mathrm{Ei}\big(-2\eta\tau(\sigma^2+\lambda_k)\big)\Big]$$

*where* $\mathrm{Ei}$ *is the exponential-integral function. (derivation in App. D.3)*

**Spectral bias.** Figure 2B traces the variance trajectory $\tilde{\lambda}_k(\tau)$ for each eigen-mode. All modes begin with the same initialization-induced level, then follow sigmoidal curves to their targets, but *in descending order of $\lambda_k$* We define the first-passage time $\tau_k^*$ as the training time at which $\tilde{\lambda}_k(\tau)$ reaches the geometric (or harmonic) mean of its initial and target values. We find the first-passage time obeys an inverse law $\tau_k^* \propto \lambda_k^{-\alpha}$, $\alpha \approx 1$, (Fig. 2C), which implies that learning a mode with variance $1/10$ smaller takes roughly 10 times longer to converge. With larger weight initialization (larger $Q_k$), the initial variance is closer to the target variance of some modes, then the inverse law splits into separate branches for modes with rising vs. decaying variance (Fig. 5B, Fig. 6).

*Practical implication.* This suggests when training stops earlier, the distribution in higher variance PC spaces have already converged, while low-variance ones—often the perceptual finer points such as letter strokes or finger joints—are under-trained. This could be an explanation for the familiar "wrong detail" artefacts in diffusion samples.

## 5 Deep and Convolutional Extensions

After analyzing the simplest linear denoiser, we set out to examine the effect of architectures via different parametrizations of the weights, specifically deeper linear models and linear convolutional networks. In the following, we will assume $\mu = 0$ and focus on learning of covariance.

## 5.1 Deeper linear network

Consider a depth-$L$ linear denoiser $\mathbf{D}(\mathbf{x}, \sigma) = \mathbf{W}_L \cdots \mathbf{W}_1 \mathbf{x}$ , where—for notational clarity—we suppress the explicit $\sigma$-dependence of weights. We assume **aligned initialization**, where for singular decomposition of each matrix, $\mathbf{W}_\ell(0) = U_\ell \Lambda_\ell V_\ell^\top$, the right basis of each layer matching the left basis of the next, $V_{\ell+1} = U_\ell, \forall \ell = 1, \ldots, L - 1$, and with $U_L = V_1 = U$ where $U$ diagonalizes data covariance $\Sigma$. Then the total weight at initialization is $\mathbf{W}_{tot}(0) = \prod_{\ell=1}^{L} \mathbf{W}_\ell(0) = U(\prod_{\ell=1}^{L} \Lambda_\ell)U^\top$, With aligned initialization, every eigenmode learns independently—mirroring classical results [15, 32]. In our case, this also implies that the total weight $\prod_l \mathbf{W}_l$ shares the eigenbasis $U$ across training and noise scales, thus commute, making sampling tractable.

One especially illuminating case is the two-layer symmetric network, where $\mathbf{D}(\mathbf{x}, \sigma) = P_\sigma P_\sigma^\top \mathbf{x}$.

**Proposition 5.1** (Dynamics of weight and distribution in two layer linear model). *Assume (i) centered data $\mu = 0$; (ii) the weight matrix is initialized aligned, i.e. $P_\sigma(0)P_\sigma(0)^\top = \sum_k Q_k \mathbf{u}_k \mathbf{u}_k^\top$, then the gradient flow ODE admits a closed-form solution (derivation in App. E.1)*

$$\mathbf{W}_\sigma(\tau) = P_\sigma(\tau)P_\sigma(\tau)^\intercal = \sum_k \frac{\lambda_k}{\sigma^2 + \lambda_k} \mathbf{u}_k \mathbf{u}_k^\intercal \left( \frac{Q_k}{(\frac{\lambda_k}{\sigma^2 + \lambda_k} - Q_k)e^{-8\eta\lambda_k\tau} + Q_k} \right) \tag{8}$$

*The generated distribution at time $\tau$ is $\mathcal{N}(\tilde{\mu}, \tilde{\Sigma})$ with $\tilde{\Sigma} = \sum_k \tilde{\lambda}_k(\tau)\mathbf{u}_k \mathbf{u}_k^\top$ and $\tilde{\lambda}_k(\tau) = \sigma_T^2 \frac{\Phi_k^2(\sigma_0)}{\Phi_k^2(\sigma_T)}$*

$$\Phi_k(\sigma) = (\sigma)^{\frac{(1-Q_k)e^{-8\eta\tau\lambda_k}}{Q_k + (1-Q_k)e^{-8\eta\tau\lambda_k}}} \left[ \lambda_k e^{-8\eta\tau\lambda_k} + Q_k \left( 1 - e^{-8\eta\tau\lambda_k} \right) \left( \lambda_k + \sigma^2 \right) \right]^{\frac{Q_k}{2Q_k + 2(1-Q_k)e^{-8\eta\tau\lambda_k}}}$$

*Interpretation.* The learning dynamics of weights and variance along different principal components are visualized in Fig.2 D-F. Compared to one-layer case, here, the weight converges along the PCs via sigmoidal dynamics, with the emergence time (reaching harmonic mean of initial and final value) $\tau_k^* = \ln 2/(8\eta\,\lambda_k)$. As for generated distribution, we find similar relationship between the target variance and emergence time $\tau_k^* \propto \lambda_k^{-\alpha}$, $\alpha \approx 1$. For the more general non-aligned initialization, we show the non-aligned parts of weight will follow non-monotonic rise-and-fall dynamics (App. E.1.2). Extensions to non-symmetric two layer model and deeper model were studied in App. F, which have similar bias but lack clean expressions.

## 5.2 Linear convolutional network

We consider a linear denoiser with convolutional architecture, $\mathbf{D}(\mathbf{x}, \sigma) = \mathbf{w}_\sigma * \mathbf{x}$ where samples $\mathbf{x} \in \mathbb{R}^N$ have 1d spatial structure, and a width $K$ convolution filter $\mathbf{w}_\sigma$ operates on it. The analysis could be easily generalized to 2d convolution. With circular boundary condition, $\mathbf{w}_\sigma$ defines a circulant weight matrix $\mathbf{W}_\sigma \in \mathbb{R}^{N \times N}$, where $\mathbf{w}_\sigma * \mathbf{x} = \mathbf{W}_\sigma \mathbf{x}$. One favorable property of circulant matrices is that they are diagonalized by *discrete Fourier transform $F$* [33].

$$\mathbf{W}_\sigma = F\Gamma_\sigma F^* \qquad F_{mk} := \frac{1}{\sqrt{N}} \exp\left( -2\pi i \frac{mk}{N} \right) \tag{9}$$

Thus all weights $\mathbf{W}_\sigma$ commutes, which allows us to leverage Lemma 4.1, and solve the sampling dynamics mode-by-mode on the Fourier basis, leading to following result.

**Proposition 5.2.** *Linear convolutional denoisers with circular boundary can only model stationary Gaussian processes (GP), with independent Fourier modes, proof in App.G.2.*

**Learning dynamics of full-width filter $K = N$** When convolution filter $\mathbf{w}_\sigma$ is as large as the signal, the gradient flow is diagonal and unconstrained in the Fourier domain. Thus, the analyses in Sec. 4 re-emerge with variance of Fourier mode $\tilde{\Sigma}_{kk}$ taking the place of $\lambda_k$.

**Proposition 5.3** (Full-width circular convolution learning dynamics). *Let $\mathbf{D}(\mathbf{x}, \sigma) = \mathbf{w}_\sigma * \mathbf{x}$, with full-width filter $K = N$, and train $\mathbf{w}_\sigma$ by full-batch gradient flow at rate $\eta$. Then the weights at noise $\sigma$ and its spectral representation $\gamma$ evolves as*

$$\mathbf{w}_\sigma(\tau) = \frac{1}{\sqrt{N}} F^* \gamma(\tau, \sigma) \quad ; \quad \gamma_k(\tau, \sigma) = \gamma_k^*(\sigma) + \left( \gamma_k(\tau, \sigma) - \gamma_k^*(\sigma) \right)e^{-2N\eta(\sigma^2 + \tilde{\Sigma}_{kk})\tau} \tag{10}$$

*where $\gamma_k^*(\sigma) = \tilde{\Sigma}_{kk}/(\sigma^2 + \tilde{\Sigma}_{kk})$ and $\tilde{\Sigma}_{kk} = [F^*\Sigma F]_{kk}$ is the variance of Fourier mode.*

*The generated distribution has diagonal covariance in the Fourier basis and follows* exactly *Prop. 4.2 after the replacement $\lambda_k \to \tilde{\Sigma}_{kk}, \eta \to N\eta, U \to F$. (derivation in App. G.3)*

Table 1: **Summary of theory**. exp. and sigm. denotes exponential and sigmoidal convergence. xN denotes the $N$ time speed up due to weight sharing.

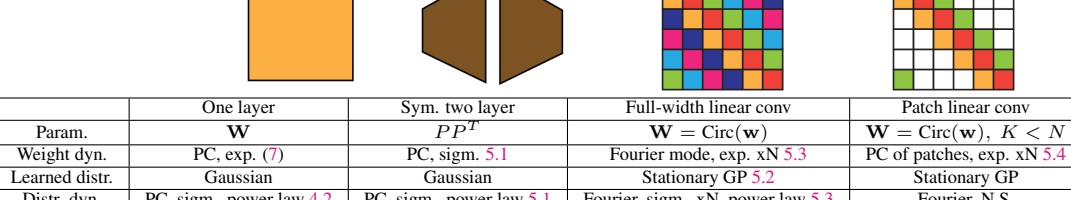

| | One layer | Sym. two layer | Full-width linear conv | Patch linear conv |
|---|---|---|---|---|
| Param. | $\mathbf{W}$ | $PP^T$ | $\mathbf{W} = \mathrm{Circ}(\mathbf{w})$ | $\mathbf{W} = \mathrm{Circ}(\mathbf{w})$, $K < N$ |
| Weight dyn. | PC, exp. (7) | PC, sigm. 5.1 | Fourier mode, exp. xN 5.3 | PC of patches, exp. xN 5.4 |
| Learned distr. | Gaussian | Gaussian | Stationary GP 5.2 | Stationary GP |
| Distr. dyn. | PC, sigm., power law 4.2 | PC, sigm., power law 5.1 | Fourier, sigm., xN, power law 5.3 | Fourier, N.S. |

*Interpretation.* The weight and distribution dynamics mirror the fully-connected case, with spectral bias towards higher variance *Fourier modes*; convolutional weight sharing simply multiplies every rate by $N$, accelerating convergence without altering the inverse-variance law.

Notably, the learned distribution is asymptotically equivalent to the *Gaussian approximation to the original training data with all possible spatial shifts* as augmentations (proof in App. G.3.2). This is one case where **equivariant architectural constraints** *facilitates creativity* as discussed in [25]. Similarly, two-layer linear conv net with full-width filter can be treated as in Sec.5.1.

**Learning dynamics of local filter** $K < N$   When the convolution filter has a limited bandwidth $K \neq N$, the Fourier domain dynamics get constrained, so it is easier to work with the filter weights. Let $r$ be the half-width of the kernel ($K = 2r + 1$). Define the circular patch extractor $\mathcal{P}_r(\mathbf{x}) = \left[\mathbf{x}_{i-r:i+r}\right]_{i=1}^N \in \mathbb{R}^{K \times N}$, and the patch covariance $\Sigma_{patch} = \frac{1}{N}\mathbb{E}_{\mathbf{x}}\left[\mathcal{P}_r(\mathbf{x})\,\mathcal{P}_r(\mathbf{x})^\top\right] \in \mathbb{R}^{K \times K}$.

**Proposition 5.4** (Patch-convolution learning dynamics)**.** *For the circular convolutional denoiser,* $\mathbf{D}(\mathbf{x}, \sigma) = \mathbf{w}_\sigma * \mathbf{x}$ *trained by full-batch gradient flow with step size* $\eta$. *Let* $\mathbf{e}_0 \in \mathbb{R}^K$ *be the one-hot vector with a single* 1 *at the center position* $r + 1$ *(1-indexed). (derivation in App. G.4)*

$$\mathbf{w}_\sigma(\tau) = \mathbf{w}_\sigma^* + \exp\left[-2N\eta\tau(\sigma^2 I + \Sigma_{patch})\right]\left(\mathbf{w}_\sigma(0) - \mathbf{w}_\sigma^*\right), \quad \mathbf{w}_\sigma^* = (\sigma^2 I + \Sigma_{patch})^{-1}\Sigma_{patch}\mathbf{e}_0.$$

*Interpretation.* Training with a narrow convolutional filter reduces to ridge regression in patch space. Under gradient flow, filter converges along eigenmodes of patch $\Sigma_{patch}$: modes with larger variance converges sooner, those with smaller variance later, preserving the inverse-variance law. It also enjoys the $N$ times speed up given by weight sharing, accelerating progress without altering the ordering. The sampling ODE remains diagonal in Fourier space, so the generated distribution will be a stationary Gaussian process with local covariance structure shaped by the learned patch denoiser, though its exact form needs numerical integration to spell out. This setting is similar to the *equivariant and local score machine* described in [25], but with the additional linear constraint.

*Simulation.* We numerically simulated the dynamics of the sample distribution for linear patch-convolution denoisers using FFHQ dataset (details in App. B.1.2). The spectral scaling exponents depend systematically on the convolutional patch size $P$: smaller kernels produced shallower, and in some cases even inverted, scaling relations (Tab. 2), potentially due to stronger coupling between more Fourier modes.

## 6 Empirical Validation of the Theory in Practical Diffusion Model Training

**General Approach**   To test our theoretical predictions about the evolution of generated distribution (esp. covariance), we resort to the following method: 1) we fix a training dataset $\{\mathbf{x}_i\}$ and compute its empirical mean $\boldsymbol{\mu}$ and covariance $\boldsymbol{\Sigma}$. We then perform an eigen-decomposition of $\boldsymbol{\Sigma}$, obtaining eigenvalues $\lambda_k$ and eigenvectors $\mathbf{u}_k$. 2) Next, we train a diffusion model on this dataset by optimizing the DSM objective with a neural network denoiser $\mathbf{D}_\theta(\mathbf{x}, \sigma)$. 3) During training, at certain steps $\tau$, we generate samples $\{\mathbf{x}_i^\tau\}$ from the diffusion model by integrating the PF-ODE (1). We then estimate the sample mean $\tilde{\boldsymbol{\mu}}^\tau$ and sample covariance $\tilde{\boldsymbol{\Sigma}}^\tau$. Finally, we compute the variance of the generated samples along the eigenbasis of training data, $\tilde{\lambda}_k^\tau = \mathbf{u}_k^\intercal \tilde{\boldsymbol{\Sigma}}^\tau \mathbf{u}_k$. To stress test our theory and maximize its relevance, we'd keep most of the training hyperparameters as practical ones.

### 6.1 Multi-Layer Perceptron (MLP)

To test our theory about *linear and deep linear network* (Prop.4.2,5.1), we used a Multi-Layer Perceptron (MLP) inspired by the SongUnet in EDM [26, 34] (details in App. I.2). We found

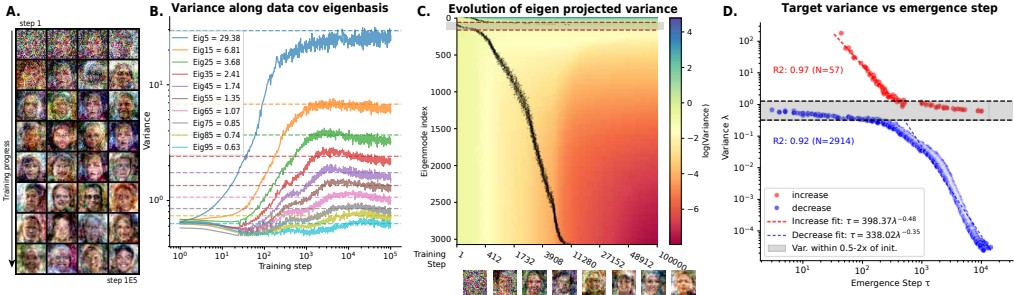

Figure 3: **Spectral Learning Dynamics of MLP-UNet (FFHQ32). A.** Generated samples during training. **B.** Evolution of sample variance $\tilde{\lambda}_k(\tau)$ across eigenmodes during training. **C.** Heatmap of variance trajectories along all eigenmodes, with dots marking mode emergence times $\tau^*$ (first-passage time at the geometric mean of initial and final variances). The **gray zone** (0.5–2× target variance) indicates modes starting too close to their target, causing unreliable $\tau^*$ estimates. **D.** Power-law scaling of $\tau^*$ versus target variance $\lambda_k$. A separate law was fit for modes with **increasing** and **decreasing** variance, excluding the middle gray-zone eigenmodes for stability.

this architecture effective in learning distribution like point cloud data (Fig. 29). We kept the preconditioning, loss weighting and initialization the same as in [26].

**Experiment 1: Zero-mean Gaussian Data x MLP** We first consider a zero mean Gaussian $\mathcal{N}(\mathbf{0}, \boldsymbol{\Sigma})$ in $d$ dimension as training distribution, with covariance defined as a randomly rotated diagonal matrix with log normal spectrum (details in App. I.4). During training, the generated variance of each eigenmode follows a sigmoidal trajectory toward its target value $\lambda_k$; modes with larger $\lambda_k$ cross the plateau sooner (Fig.10**A**). We mark the emergence time $\tau^*$ as the step at which the variance reaches the geometric mean of its initial and asymptotic values (Fig. 10**B**). Across both high- and low-variance modes, $\tau^*$ obeys an inverse power-law, $\tau^* \propto \lambda_k^{-\alpha}$. With higher-dimensional Gaussians the exponent is estimated more precisely and remains close to 1: for $d = 256$, $\alpha = 1.08$; for $d = 512$, $\alpha_{\text{incr}} = 1.05$ and $\alpha_{\text{decr}} = 1.13$ (Fig. 10**C**). The scaling breaks down only for modes whose initial variance is already near $\lambda_k$; in that regime the trajectory is less sigmoidal and $\tau^*$ becomes ill-defined. This result shows that despite many non-idealistic conditions e.g. *deeper network*, *nonlinear activation function*, *residual connections*, *normal weights initialization*, *shared parametrization of denoisers at different noise level*, the prediction from the linear network theory is still *quantitatively* correct.

**Experiment 2: Natural Image Datasets x MLP** Next, we validated our theory on natural image datasets. We flattened the images as a vectors, and trained a deeper and wider MLP-UNet to learn the distribution. Using FFHQ as our running example, monitoring the generated samples throughout training (Fig. 3A), despite heavy noise early on, the coarse facial contours—corresponding to the mean and top principal components of human face distribution [7]—emerge quickly, whereas high-frequency details (lower PCs) only appear later. We note that this spectral ordering effect of training dynamics is reminiscent and similar to that in the sampling dynamics after training [19, 22].

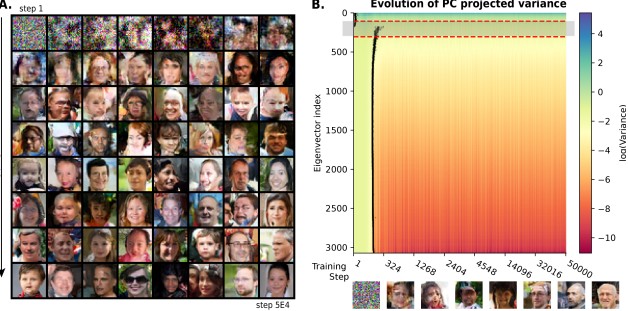

Figure 4: **Learning dynamics of UNet differs | FFHQ32. A.** Sample trajectory from CNN-UNet. **B.** Variance evolution along covariance eigenmodes. (c.f. Fig. 3A.C.)

Quantitatively, the sample covariance $\tilde{\Sigma}^\tau$ rapidly aligns with and becomes close to diagonal in the data eigenbasis $U$ (Fig. 11). The top eigenmodes' variances, $\tilde{\lambda}_k(\tau)$, follow sigmoidal trajectories converging to their targets, and their "emergence times" $\tau_k^*$ increase down the spectrum (Fig. 3**B,C**). We exclude a **central band** of modes whose initial variances lie within 0.5–2× the target, since their undulating learning dynamics make first-passage time estimates unreliable. After this exclusion, modes with **increasing** and **decreasing** variance each exhibit a clear power-law scaling between

emergence step $\tau^*$ and target variance $\lambda_k$, with exponents $-0.48$ ($R^2 = 0.97$, $N = 57$) and $-0.35$ ($R^2 = 0.92$, $N = 2{,}914$), respectively (Fig. 3**D**). Although the observed spectral bias is slightly attenuated relative to the Gaussian case and linear theory prediction, it remains robust and consistent across datasets (MNIST, CIFAR-10, FFHQ32 and AFHQ32) (App. B.2.2). This shows that even with natural image data, the distributional learning dynamics of MLP-based diffusion still suffers from slower convergence speed for lower eigenmodes.

## 6.2 Convolutional Neural Networks (CNNs)

Next we turn to the convolutional U-Net—the work-horse of image-diffusion models [34, 35]. For a full-width linear convolutional network our analysis predicts an inverse-variance law in Fourier space (Prop. 5.3). The patch-convolution variant lacks a clear forecast on distribution, so the following experiments probe empirically whether—and how—its learning dynamics is affected by spectral bias.

**Experiment 3: Natural Image Datasets x CNN UNet**  Training on the same FFHQ dataset, the distributional learning trajectory of CNN-UNet is *markedly different* from the MLPs: early in training, we do not see contour of face, but locally coherent patches, reminiscent of Ising models (Fig. 4**A**.). Visually and variance-wise, the CNN-based UNet converge much faster and better than the MLP-based UNet, matching the N-fold speed-up from weight sharing (Prop. 5.4; Fig. 4**B**). When projecting onto the data eigenbasis, all eigenmodes with increasing variance rise simultaneously, while eigenmodes with decreasing variance co-decay at a later time, giving an effective power-law exponent $\alpha \approx 0$; Thus, spectral bias is essentially absent (Fig. 13**C.D.**).

*Why is spectral bias absent?* On the **theory** side, the likely cause is locality: local convolutional filters couple neighbouring pixels, binding many Fourier modes into one learning unit. Because sampling remains diagonal in Fourier space, a broad band of modes is amplified simultaneously, attenuating the spectral ordering, as we observed numerically in App. B.1.2. In line with this, early in training, the CNN denoiser is indeed well-approximated by a local linear filter (Fig. 15).

On the **empirical** side, the key factor appears to be network width. We systematically varied the channel number and depth of deep convolutional denoisers (App. B.2.4), and found that—regardless of depth—narrower networks (e.g. $ch = 4$) exhibit slower convergence and a stronger spectral ordering, consistent with the patch-convolution theory (Fig. 22,23). In contrast, wide networks with many channels ($ch = 128$) learn spectral modes almost instantaneously, similar to our observations in practical UNet. In hindsight, the analytic theory effectively assumes a convolution with the same number of channels as the input (e.g., RGB = 3), so the ratio between the network's channel and the input channel likely governs the deviation from theoretical predictions.

A complete analytic treatment of convolutional U-Net training dynamics is left for future work.

## 7   Discussion

In summary, we presented closed-form solutions for training denoisers with linear, deep linear or linear convolutional architecture, under the DSM objective on arbitrary data. This setup allows for a precise *mode-wise understanding* of the gradient flow dynamics of the denoiser and the evolution of the learned distribution: covariance eigenmode for deep linear network and Fourier mode for convolutional networks. For both the weights and the distribution, we showed analytical evidence of *spectral bias*, i.e. weights converge faster along the eigenmodes or Fourier modes with high variance, and the learned distribution recovers the true variance first along the top eigenmodes. These theoretical results are summarized in Tab. 1.

We hope these results can serve as a solvable model for spectral bias in the diffusion models through the nested training and sampling dynamics. Furthermore, our analysis is not limited to the diffusion and the DSM loss, in App. H, we showed a similar derivation for the spectral bias in flow matching models [28, 36].

**Relevance of our theoretical assumptions**  We found, for the purpose of analytical tractability, we made many idealistic assumptions about neural network training, 1) linear neural network, 2) small or orthogonal weight initialization, 3) "full-batch" gradient flow, 4) independent evolution of weights at each noise scale. In our MLP experiments, we found even when all of these assumptions were somewhat violated, the general theoretical prediction is correct, with modified power coefficients.

This shows most of these assumptions could be relaxed in real life, and the spectrum of data indeed have a large effect on the learning dynamics, esp. for fully connected networks.

**Inductive bias of the local convolution**    In our CNN experiments, however, the theoretical predictions from linear models deviate: the spectral bias in learning speed does not directly apply to the distribution of full images. Although our theory predicts that filter-weight learning dynamics are governed by the patch covariance, the ultimate image distribution is shaped by the convolution of those filters. To date, many learning-theory analyses for diffusion models assume MLP-like architectures [24]. For future theoretical work on the learning dynamics of practical diffusion models, a rigorous treatment of the local convolutional structure—and its frequency-coupling effects—will likely be essential, rather than relying on full-width convolution analyses [37].

**Implications for high channel inputs**    Our ablation suggests that the ratio between network width and input channel count may underlie the observed deviations from theoretical predictions—specifically, the absent of spectral bias and the near-simultaneous convergence of eigenmodes. While most image and latent representations traditionally have few channels (e.g., 3 for RGB, 4 for latent diffusion [38]), recent architectures employ much higher channel counts—such as DC-AEs with 64–128 channels [39] or encoder-based diffusion models with $ch = 768$ [40]. In these regimes, where input channel becomes comparable to that of the UNet or DiT, the theory's predictions may become increasingly relevant for understanding, regularizing, and stabilizing training dynamics in high–channel-dimensional diffusion models.

**Broader Impact**    Although our work is primarily theoretical, the inverse scaling law could offer valuable insights into how to improve the training of large-scale diffusion or flow generative models.

**Acknowledgements.**    We are grateful to Jacob Zavatone-Veth, Yongyi Yang, Michael Albergo, Hugo Cui, Yue Lu, and Ekdeep Lubana for their insightful discussions and valuable pointers to relevant literature. We appreciate Haim Sompolinsky and Zhengdao Chen for their meticulous proofreading and commenting on earlier versions of the manuscript. We also thank Thomas Fel for providing code formatting snippet. This work has been made possible in part by a gift from the Chan Zuckerberg Initiative Foundation to establish the Kempner Institute for the Study of Natural and Artificial Intelligence, and by the Institute's generous support and computing resources. B.W. is supported by Kempner fellowship. C.P. is supported by an NSF CAREER Award (IIS-2239780), DARPA grants DIAL-FP-038 and AIQ-HR00112520041, the Simons Collaboration on the Physics of Learning and Neural Computation, and the William F. Milton Fund from Harvard University.

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

# Appendix

# Contents

# A Extended Related works

Beyond the closely related works reviewed in the main text, here we are some spiritually related lines of works that inspired ours.

**Spectral effect in the sampling process of diffusion models**   Many works have observed that during the sampling process of diffusion models [22, 41]: low spatial frequency aspects of the sample (e.g. layout) were specified first in the denoiser, before the higher frequency ones (e.g. textures). This phenomenon has been understood through the natural statistics of images (e.g. power-law spectrum) [19] and theory of diffusion [11], and recently through the lens of stochastic localization [42]. Basically, low frequency aspects usually have higher variance, thus were later to be corrupted by noise, so earlier to be generated during sampling process.

In our current work, we extend this line of thought to consider the spectral effects on the training dynamics of diffusion models.

**Inductive bias of deep networks**   There as been a rich history of studying the inductive bias or implicit regularization effect of deep neural network and gradient descent. Deep neural networks have been reported to tend to find low-rank solutions of the task [43], and deeper networks could find it difficult to learn higher-rank target functions. This finding has also been leveraged to facilitate low-rank solutions by over-parameterizing linear operations in a deep networks (e.g. linear [44] or convolution [45] layers).

**Implicit bias of convolutional neural networks**   When the neural network has convolutional structures in it, what kind of inductive bias or regularization effect does it bring to the function approximator?

People have attacked this by analyzing (deep) linear networks. [37] analyzed the inductive bias of gradient learning of the linear convolution network. In their case, the kernel is as wide as the signal and with circular boundary condition, thus convolution is equivalent to pointwise multiplication in Fourier space, which simplified the problem a lot. Then they can derive the learning dynamics of each Fourier mode. This result can be unified with other linear network approaches [18].

[46] further analyzed the inductive bias of the linear convolutional network with non-trivial local kernel size (neither pointwise nor full image) and multiple channels, and provided analytical statements about the inductive bias. However, they also found less success for closed form solutions for even two-layer convolutional networks with finite kernel width.

From an algebraic and geometric perspective, [47, 48] have analyzed the geometry of the function space of the deep linear convolutional network, which is equivalent to the space of polynomials that can be factorized into shorter polynomials.

**Deep image prior and spectral bias in CNN**   On the empirical side, one intriguing result comes from the famous Deep Image Prior (DIP) experiment of Ulyanov et al. [49]. They showed that if a deep convolutional network (e.g. UNet) is used to regress a noisy image as target with pure noise as input, then when we employ early stopping in the optimization, the neural network will produce a cleaner image, thus performing denoising. To understand this method, [50] showed empirical evidence that deep convolutional networks tend to fit the low *spatial frequency* aspect of the data first. Thus, given the different spectral signature of natural images and noise, networks will fit the natural image before the noise. As a corollary, they showed that if the noise has more low frequency components, then neural network will fail to denoise those low frequency corruptions from image.

People have also looked at the inductive bias of untrained convolutional neural networks. Theoretically, [51] showed that infinite-width convolutional network at initialization is equivalent to spatial Gaussian process (random field), and the authors used this Bayesian perspective to understand the Deep image prior.

We noticed that this line of works in deep image prior has intriguing conceptual connection to our current work, i.e. the spectral bias of learning a function with convolutional architecture tend to learn lower frequency aspect first. Comparing diffusion models to DIP, diffusion models regress clean images from many randomly sampled noisy images; on the contrary DIP regress the clean images on a single noise pattern.

**Neural Tangent Kernel**   A widely recognized technique for analyzing the learning dynamics of deep neural network is the neural tangent kernel. For example, an infinitely wide network would be similar to a kernel machine, where the learning dynamics will be linearized and reduce to exponential convergence along different eigenmode of the tangent kernel.

Using neural tangent kernel (NTK) techniques, by inspecting the eigenvalues of the NTK associated with functions of different frequency, [52] has been able to show that given uniform data on sphere assumption, and simple neural network architectures (two-layer fully connected network with ReLU nonlinearity), neural networks learn lower-frequency functions faster, with learning speed quadratically related to the frequency. Later they lifted the spatial uniformity assumption [53], and derived how convergence speed and eigenvalues depend on the local data density.

These insights have been leveraged in classification problems to show that early stopping can lead neural networks to learn smoother functions, thus being robust to labeling noise [54] .

What about convolutional architecture? With some similar NTK techniques, using a simplified architecture, [55] proved that the learning dynamics of the convolutional network will preferably learn the lower spatial frequency aspect of target image first. Their proof technique is also based on the relationship between over-parametrized neural network and the tangent kernel. The proof is based on a simpler generator architecture: one convolutional layer with ReLU nonlinearity and fixed readout vector. They numerically showed the same effect for deeper architectures. This result provided further theoretical foundation for the Deep Image Prior.

# B Extended Results

## B.1 Extended Visualization of Theoretical Results

### B.1.1 Scaling curves of diffusion training (EDM) - fully connected

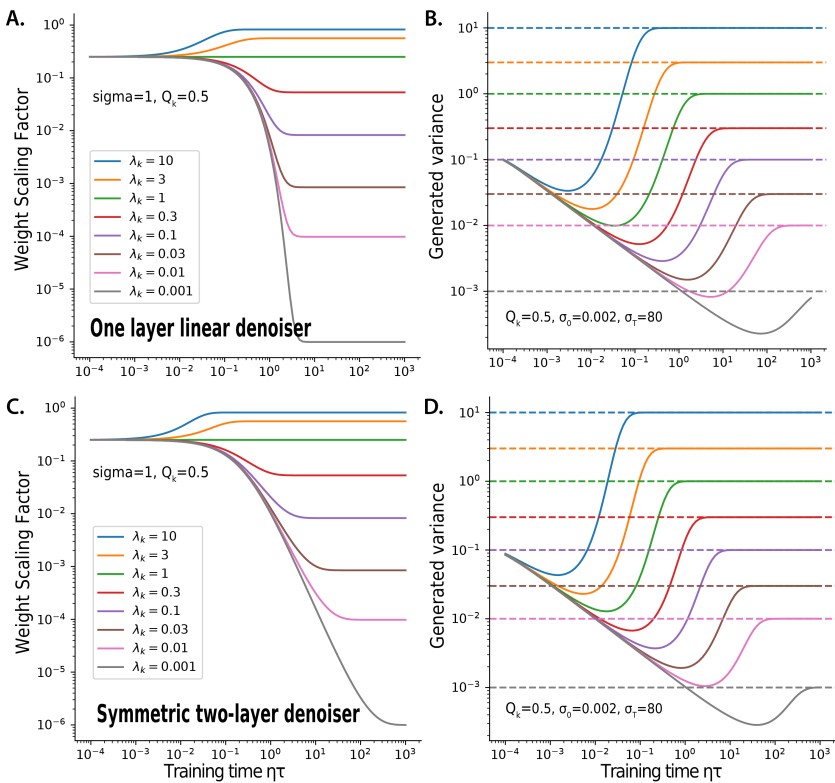

Figure 5: **Learning dynamics of the weight and variance of the generated distribution per eigenmode (continued) Top** Single layer linear denoiser. **Bottom** Symmetric two-layer denoiser. **A.C.** Learning dynamics of $\mathbf{u}_k^\mathsf{T}\mathbf{W}(\tau)\mathbf{u}_k$. **B.D.** Learning dynamics of the variance of the generated distribution $\tilde{\lambda}_k$, as a function of the variance of the target eigenmode $\lambda_k$. This case with larger amplitude weight initialization $Q_k = 0.5$.

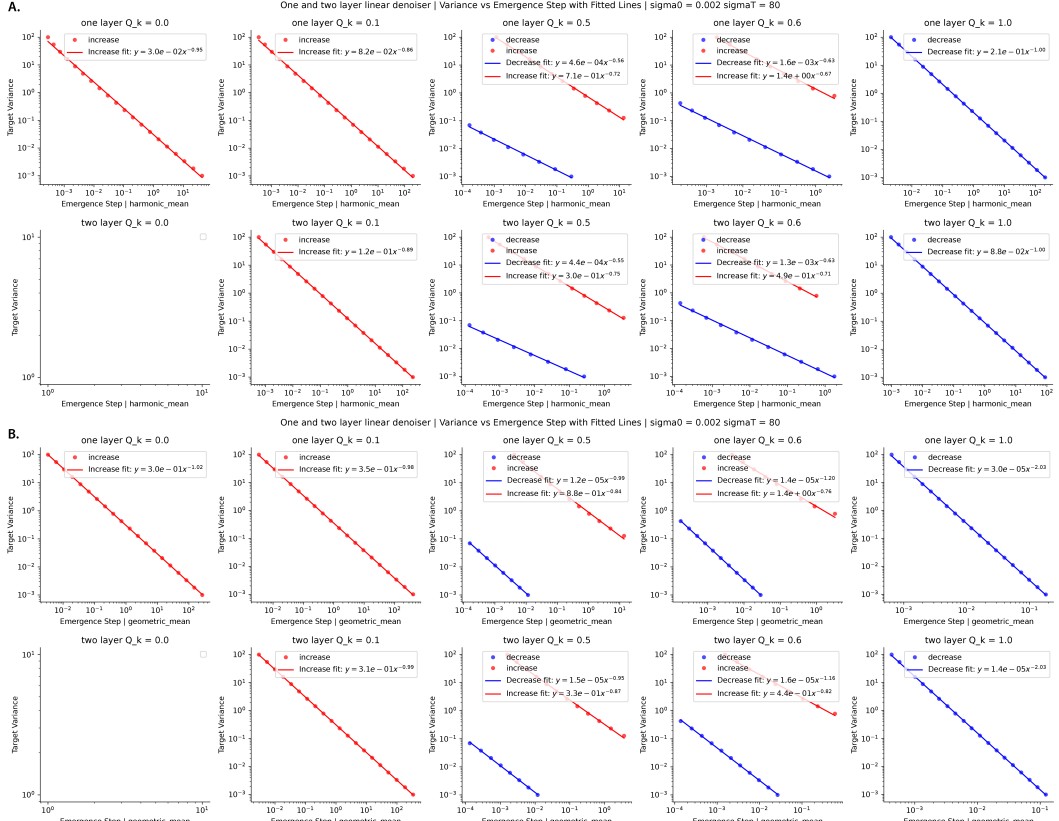

Figure 6: **Power law relationship between mode emergence time and target mode variance for one-layer and two-layer linear denoisers.** Panels (A) and (B) respectively plot the Mode variance against the Emergence Step for different values of weight initialization $Q_k \in \{0.0, 0.1, 0.5, 0.6, 1.0\}$ (columns), for one layer and two layer linear denoser (rows). We used $\sigma_0 = 0.002$ and $\sigma_T = 80$. The emergence steps were quantified via different criterions, via harmonic mean in **A**, and geometric mean in **B**. Within each panel, red markers and lines denote the modes where their variance increases; blue markers and lines denote modes that "decrease" their variance. The solid lines show least-squares fits on log-log scale, giving rise to the $y = a\,x^b$ type relation. Comparisons reveal a systematic power-law decay of variance with respect to the Emergence Step under both the harmonic-mean and geometric-mean definitions. Note, the $Q_k = 0$ and two layer case was empty since zero initialization is an (unstable) fixed point, thus it will not converge.

### B.1.2 Scaling curves of diffusion training (EDM) - linear convolution

**Method**  We performed numerical simulations to study the learning dynamics of patch-based convolutional filters. Using the `FFHQ32` dataset as an example, we first extracted local patch statistics to compute the optimal linear filter at each noise scale $w_\sigma^*$. Then, applying Proposition 5.4, we obtained the learned filter at different training times and solved the corresponding PF-ODE in the spatial domain to generate samples. From these samples, we computed their variance along eigenmodes and analyzed the resulting scaling laws.

**Key factors influencing scaling behavior.**  Our simulations revealed two critical factors affecting the observed power-law relationship between image eigenvalues and convergence time: the filter patch size $P$ and the initialization of filter weights.

When the filter weights were initialized as an identity matrix (without scaling), the initial sample variance became excessively large, and the resulting power-law exponent was around $-0.4$—already weaker than that predicted by the linear-case solution. A more practical alternative is to initialize the filter weights as a scaled identity:

$$W_0 = \frac{\sigma_{\text{data}}^2}{\sigma^2 + \sigma_{\text{data}}^2} I,$$

inspired by the skip-connection coefficient $c_{\text{skip}}$ in the EDM preconditioning scheme. Under this initialization, we observed both increasing and decreasing variance modes during training.

**Effect of patch size.**  The specific scaling relations depend strongly on the patch size $P$. When $P$ decreases, the scaling exponent for the decreasing modes approaches zero—from about $-0.4$ for $P = 15$ to about $-0.21$ for $P = 3$ ($3 \times 3$ convolution). The overall convergence time also increases for smaller patch sizes. For the modes with increasing variance, smaller patches further attenuate the scaling exponent toward $0.0$ ($P = 7$) or even flip the trend ($0.48$ for $P = 3$). Intuitively, a small $3 \times 3$ convolution couples many frequency components in Fourier space, so multiple modes are learned simultaneously during generation.

**Summary.**  Although we no longer have an analytical expression for the scaling coefficients, the numerical simulations allow us to predict the scaling relations empirically. These results indicate that the patch size in convolutional architectures significantly modulates the scaling law. We hypothesize that the attenuated or flat scaling behavior observed in practical UNet architectures trained on natural images may, in part, stem from their reliance on small ($3 \times 3$) convolutional filters.

Table 2: **Spectral convergence time computed for patch linear convolutional networks on FFHQ32** at varying patch size.

| Patch size $P$ | Increasing scaling | Decreasing scaling |
|:---:|:---:|:---:|
| 3  | $0.06\lambda^{0.48}$  | $11.88\lambda^{-0.21}$ |
| 5  | $0.05\lambda^{0.24}$  | $3.66\lambda^{-0.36}$  |
| 7  | $0.05\lambda^{0.01}$  | $3.24\lambda^{-0.37}$  |
| 11 | $0.07\lambda^{-0.22}$ | $2.61\lambda^{-0.40}$  |
| 15 | $0.08\lambda^{-0.28}$ | $2.57\lambda^{-0.40}$  |
| 25 | $0.09\lambda^{-0.30}$ | $2.54\lambda^{-0.40}$  |

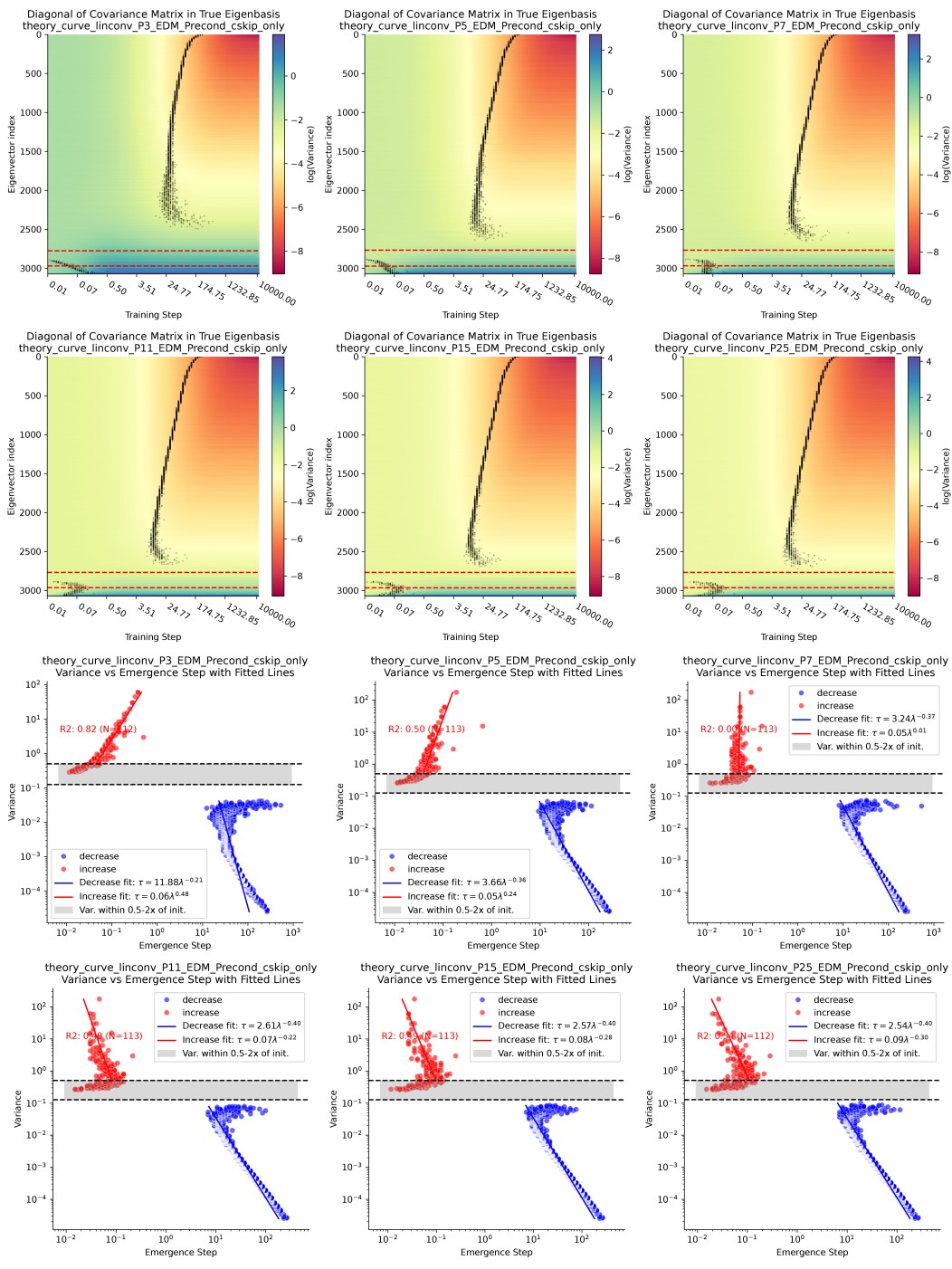

Figure 7: **Numerical simulation of spectral learning curve for linear convolutional network with local patch filter with patch size** $P = 3, 5, 7, 11, 15, 25$**, using FFHQ32 dataset. Top** same format as Fig.3**C**. **Bottom** same format as Fig.3**D**.

### B.1.3 Scaling curves of flow matching

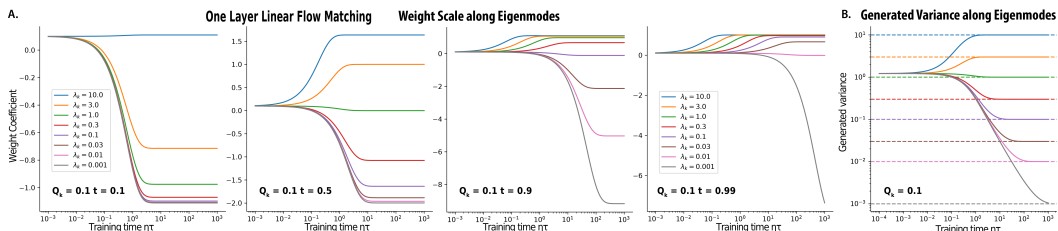

Figure 8: **Learning dynamics of the weight and variance of the generated distribution per eigenmode, for one layer linear flow matching model** Similar plotting format as Fig. 2. **A.** Learning dynamics of weights $\mathbf{u}_k^\mathsf{T}\mathbf{W}(\tau;t)\mathbf{u}_k$ for various time point $t \in \{0.1, 0.5, 0.9, 0.99\}$. **B.** Learning dynamics of the variance of the generated distribution $\tilde{\lambda}_k$, as a function of the variance of the target eigenmode $\lambda_k \in \{10, 3, 1, 0.3, 0.1, 0.03, 0.01, 0.001\}$. Weight initialization is set at $Q_k = 0.1$ for every mode.

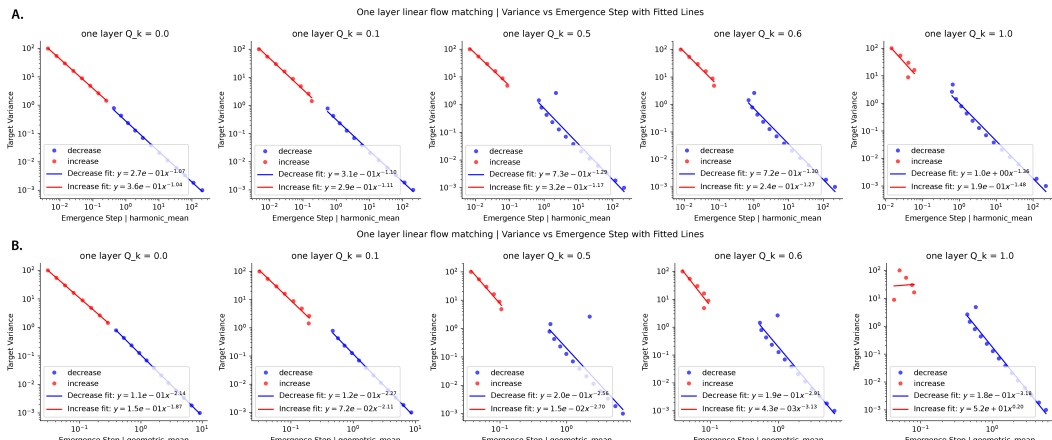

Figure 9: **Power law relationship between mode emergence time and target mode variance for one-layer linear flow matching.** Panels (A) and (B) respectively plot the Mode variance against the Emergence Step for different values of weight initialization $Q_k \in \{0.0, 0.1, 0.5, 0.6, 1.0\}$ (columns), for one layer linear flow model. The emergence steps were quantified via different criterions, via harmonic mean in **A**, and geometric mean in **B**. We used the same plotting format as in Fig. 6. Comparisons reveal a systematic power-law decay of variance with respect to the Emergence Step under both the harmonic-mean and geometric-mean definitions.

## B.2 Extended Empirical Results

### B.2.1 MLP-UNet Gaussian training experiments

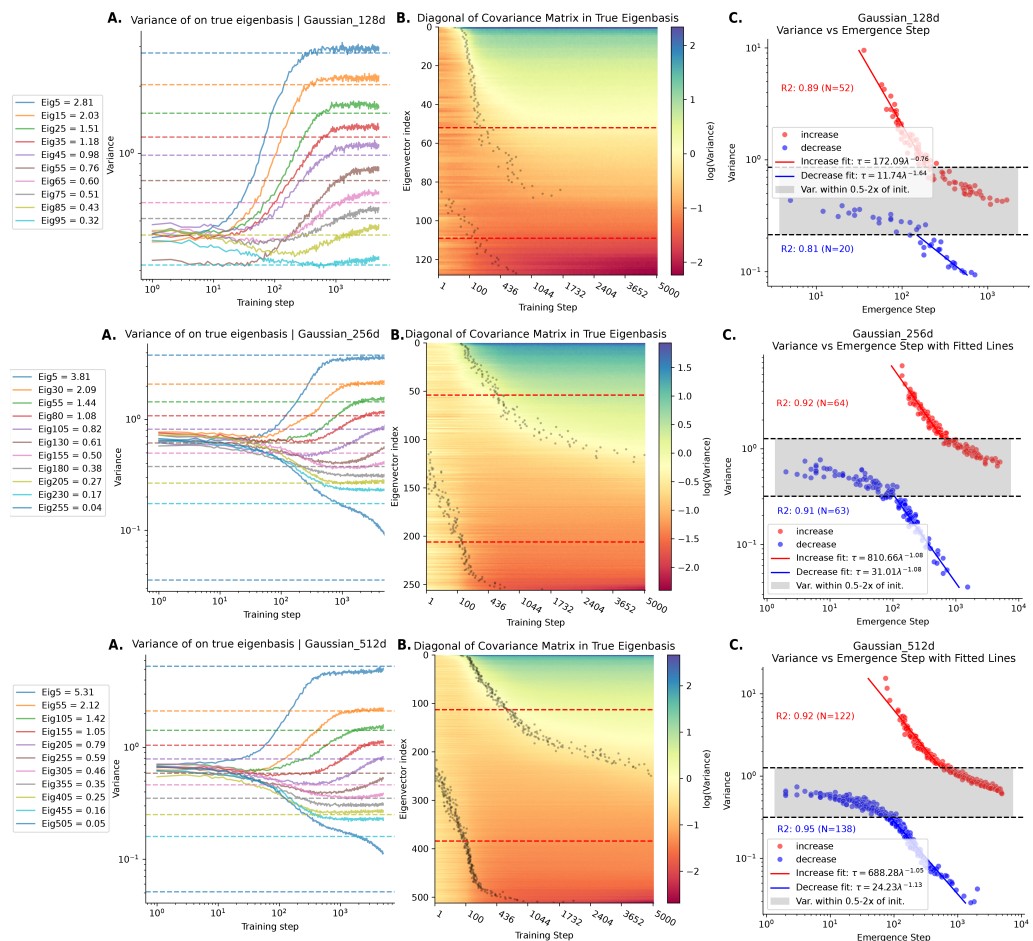

Figure 10: **Spectral Learning Dynamics of MLP-UNet (Gaussian-rotated).** (same layout and analysis procedure as main Fig. 3) Top, middle, bottom show cases for 128d, 256d and 512d Gaussian. **A.** Evolution of sample variance $\tilde{\lambda}_k(\tau)$ across eigenmodes during training. **B.** Heatmap of variance trajectories along all eigenmodes, with dots marking mode emergence times $\tau^*$ (first-passage time at the geometric mean of initial and final variances). The gray zone (0.5–2× target variance) indicates modes starting too close to their target, causing unreliable $\tau^*$ estimates. **C.** Power-law scaling of $\tau^*$ versus target variance $\lambda_k$, excluding gray-zone eigenmodes for stability.

### B.2.2 MLP-UNet natural image training experiments

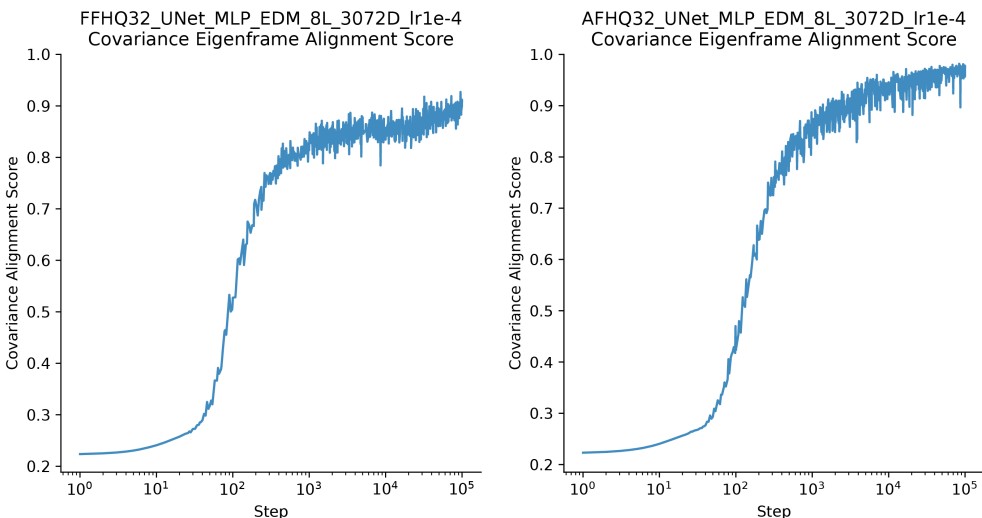

Figure 11: **Dynamical alignment onto the covariance eigenframe of data (MLP-UNet, FFHQ32, AFHQ32).** Alignment score $\chi$ as function of training step. Alignment score defined as the sum of square of diagonal entries of the rotated sample covariance on the training data eigenframe $U^T \tilde{\Sigma}_\tau U$, divided by the sum of square of all entries. This quantifies how well the training data eigenframe diagonalizes the generated sample covariance. It will be $\chi = 1$ if $U$ is the eigenbasis of $\tilde{\Sigma}_\tau$.

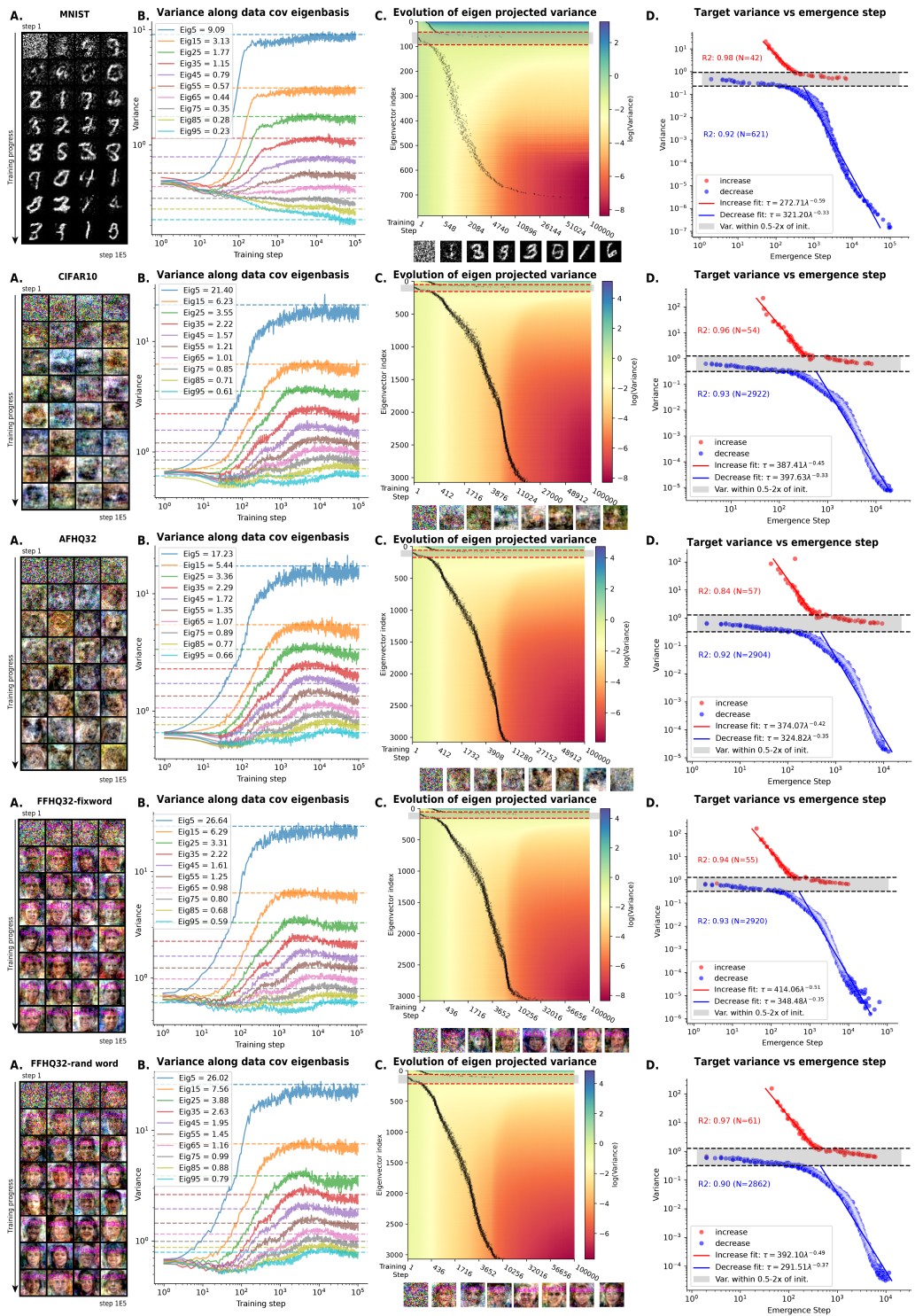

Figure 12: **Spectral Learning Dynamics of MLP-UNet (MNIST, CIFAR10, AFHQ32, FFHQ32-fixword, random word). A.** Generated samples during training. **B.** Evolution of sample variance $\tilde{\lambda}_k(\tau)$ across eigenmodes during training. **C.** Heatmap of variance trajectories along all eigenmodes, with dots marking mode emergence times $\tau^*$ (first-passage time at the geometric mean of initial and final variances). The **gray zone** (0.5–2× target variance) indicates modes starting too close to their target, causing unreliable $\tau^*$ estimates. **D.** Power-law scaling of $\tau^*$ versus target variance $\lambda_k$. A separate law was fit for modes with **increasing** and **decreasing** variance, excluding the middle gray-zone eigenmodes for stability.

## B.2.3 CNN-UNet training experiment

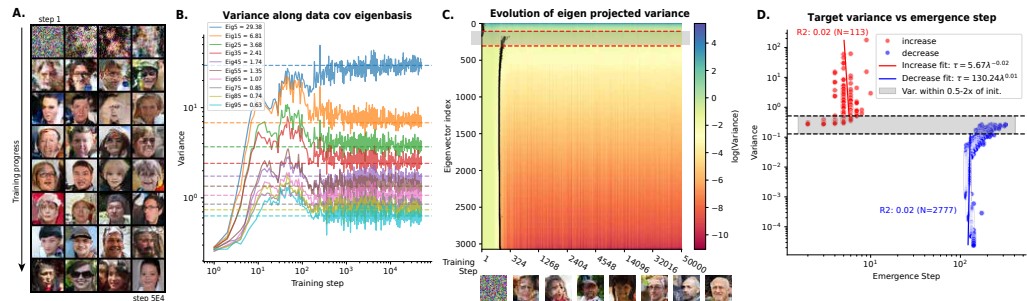

Figure 13: **Spectral Learning Dynamics of CNN-UNet (FFHQ32).** (same layout and analysis procedure as main Fig. 3) **A.** Generated samples during training. **B.** Evolution of sample variance $\tilde{\lambda}_k(\tau)$ across eigenmodes during training. **C.** Heatmap of variance trajectories along all eigenmodes, with dots marking mode emergence times $\tau^*$ (first-passage time at the geometric mean of initial and final variances). The gray zone (0.5–2× target variance) indicates modes starting too close to their target, causing unreliable $\tau^*$ estimates. **D.** Power-law scaling of $\tau^*$ versus target variance $\lambda_k$, excluding gray-zone eigenmodes for stability.

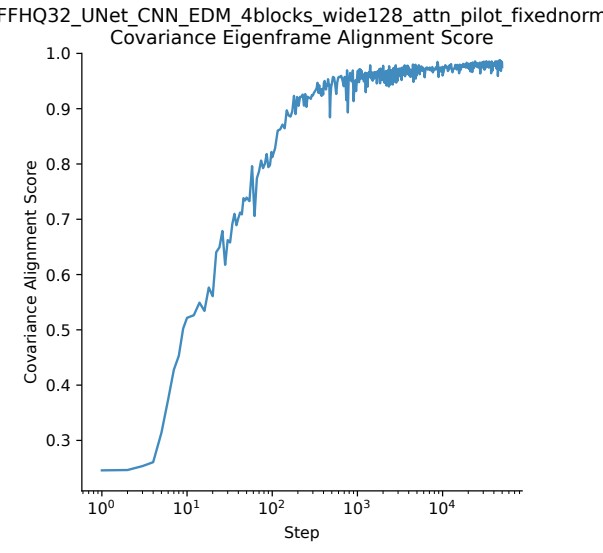

Figure 14: **Dynamical alignment onto the covariance eigenframe of data (CNN-UNet, FFHQ32).** Alignment score $r$ as function of training step. Same analysis as Fig.11.

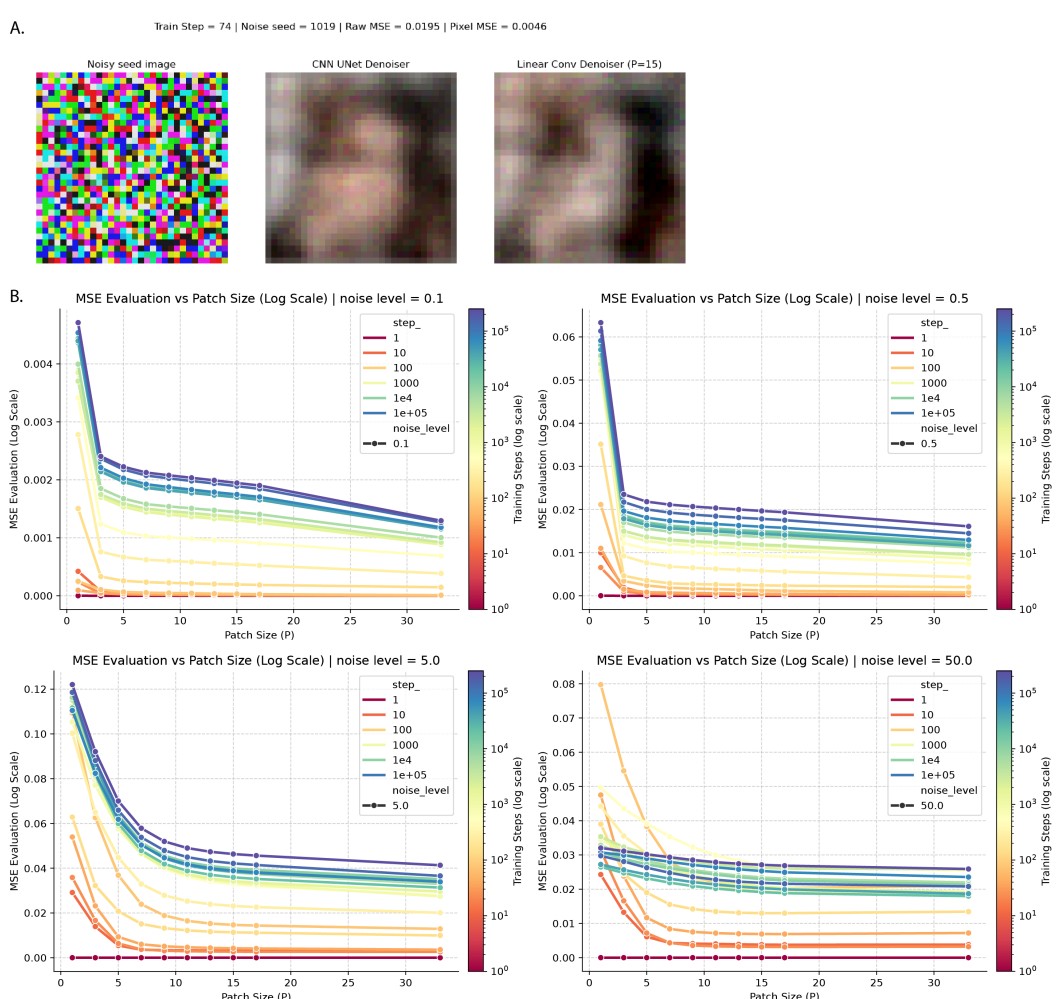

Figure 15: **UNet denoiser can be approximated by linear convolution early in training (CNN-UNet, FFHQ32). A.** Early in training, the UNet denoiser output can be well approximated by a linear convolutional layer, with a patch size $P$. **B.** The approximation error as a function of patch size $P$, training time $\tau$ and noise scale $\sigma$. Generally, early in training, the denoiser is very local and linear, well approximated by a linear convolutional layer.

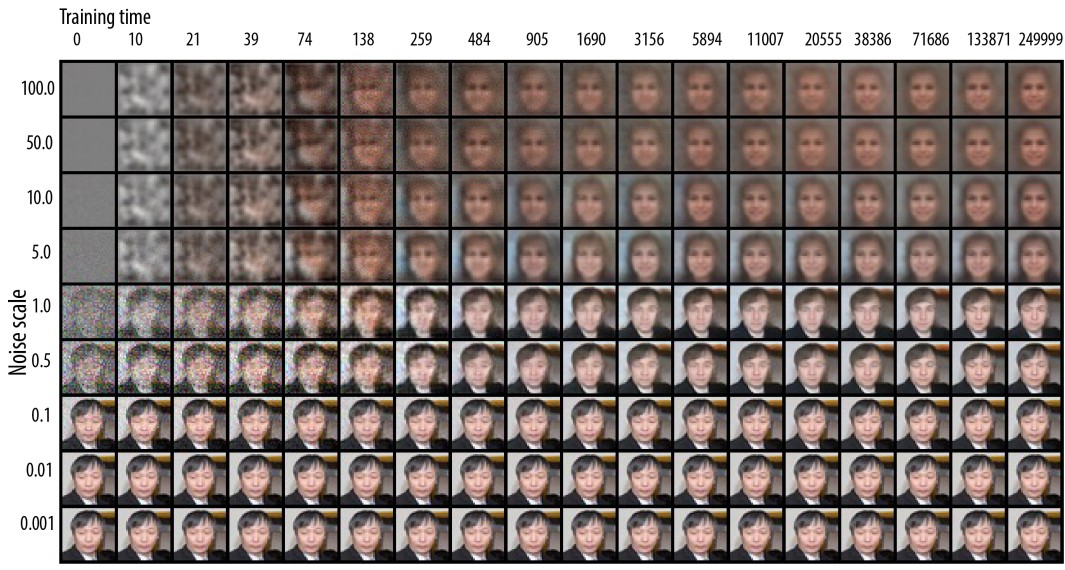

Figure 16: **Visualizing the denoiser training dynamics with a fixed image and noise seed (CNN-UNet, FFHQ32).** $D(x + \sigma z, \sigma)$ as a function of training time $\tau$ and noise scale $\sigma$.

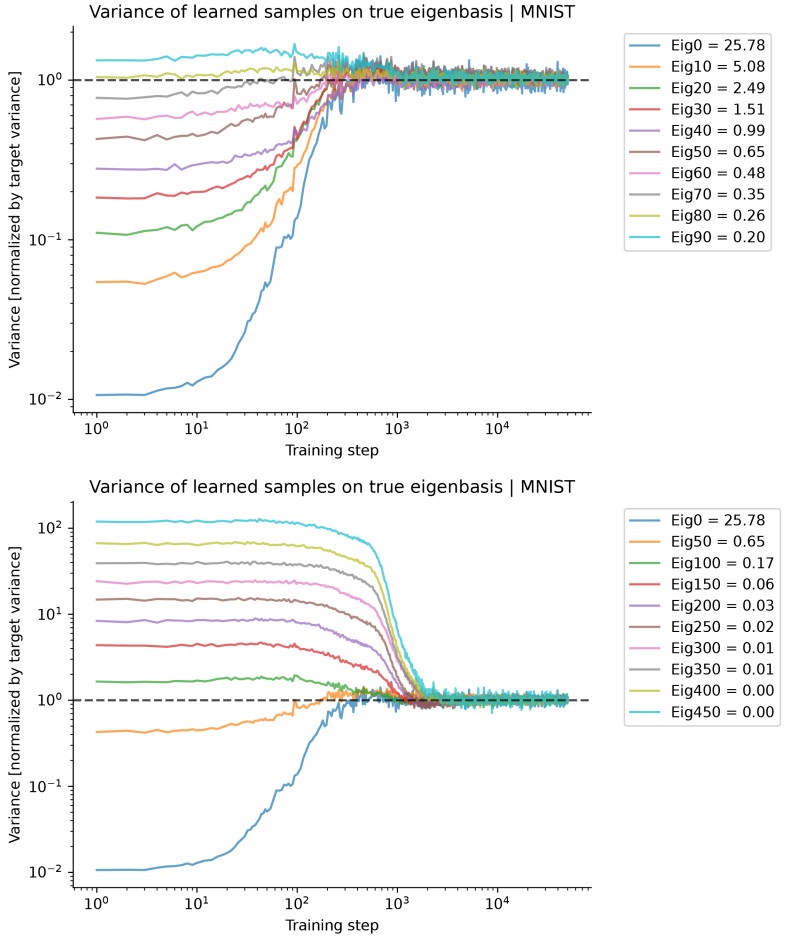

Figure 17: **Spectral Bias in Whole Image of CNN learning | MNIST** Training dynamics of sample (whole image) variance along eigenbasis of training set, normalized by target variance. **Upper** 0-100 eigen modes, **Lower** 0-500 eigenmodes.

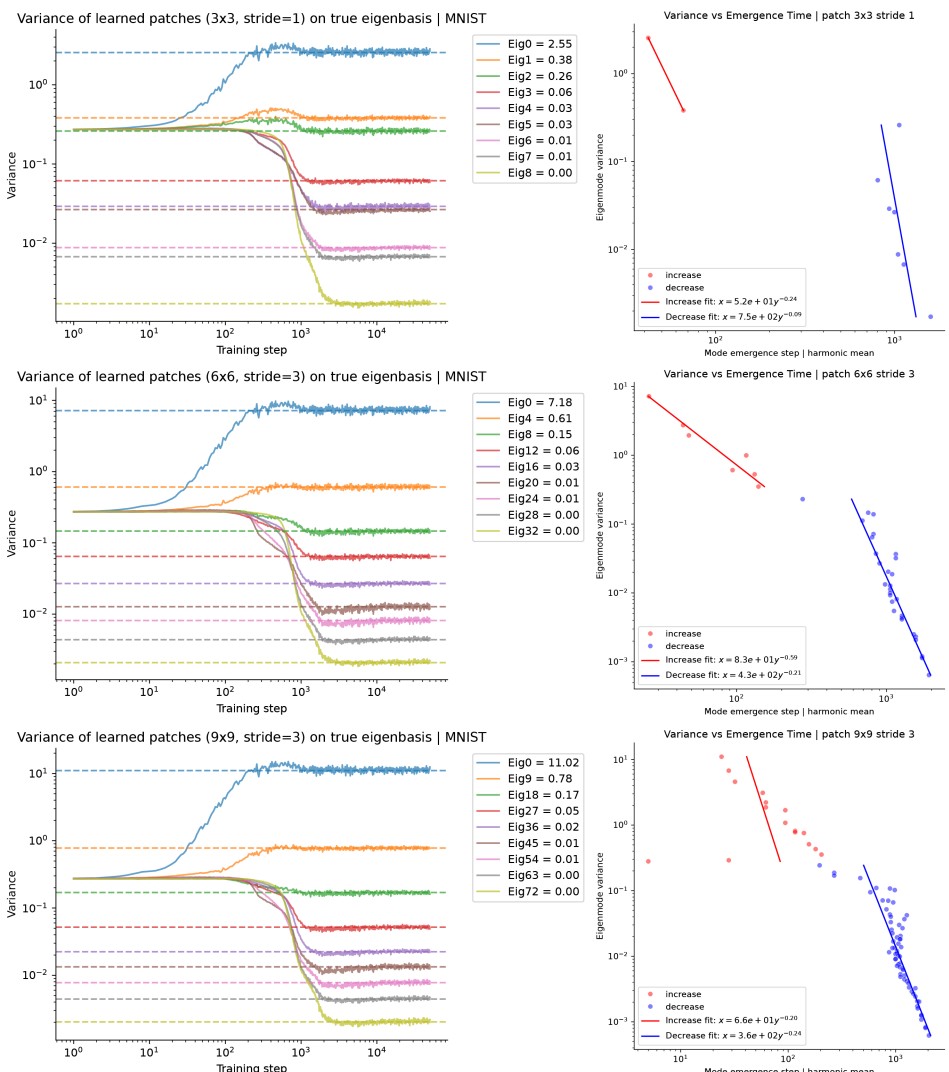

Figure 18: **Spectral Bias in CNN-Based Diffusion Learning: Variance Dynamics in Image Patches | MNIST (32 pixel resolution). Left,** Raw variance of generated patches along true eigenbases during training. **Right,** Scaling relationship between the target variance of eigenmode versus mode emergence time (harmonic mean criterion). Each row corresponds to a different patch size and stride used for extracting patches from images.

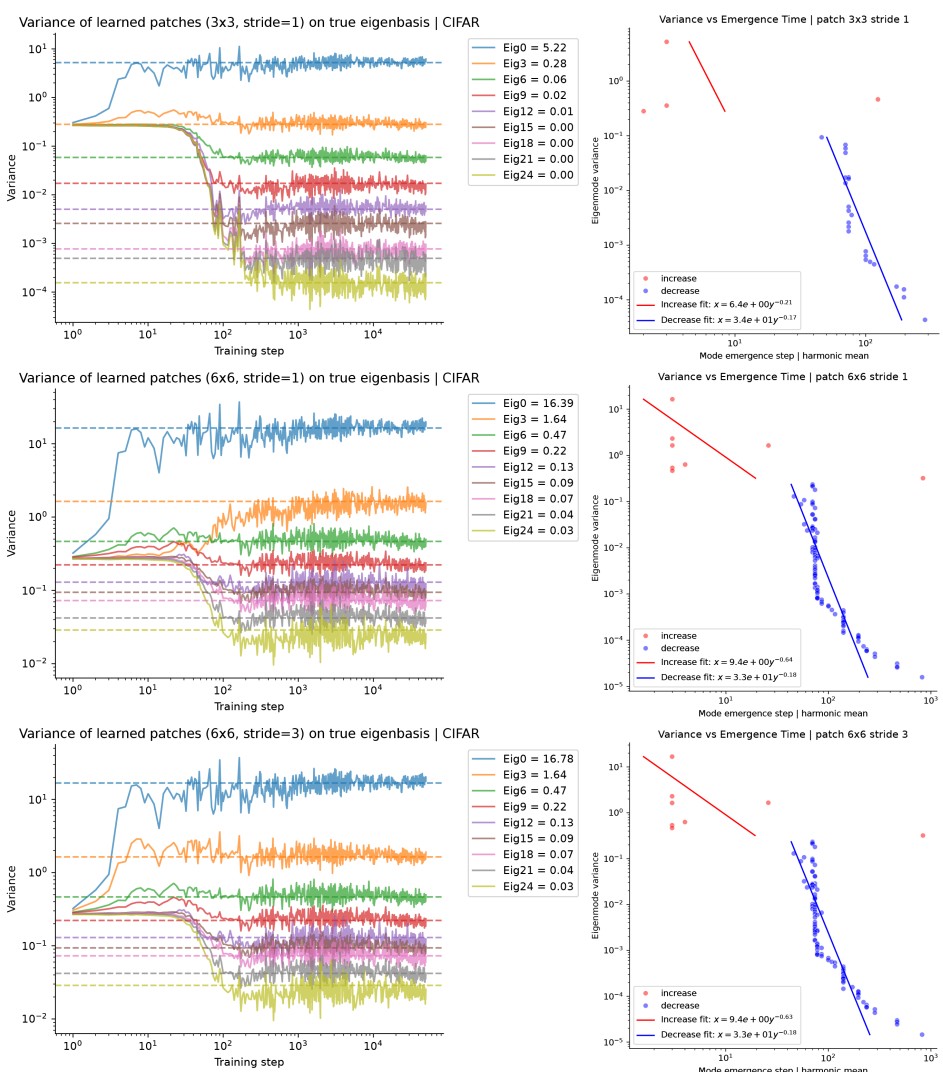

Figure 19: **Spectral Bias in CNN-Based Diffusion Learning: Variance Dynamics in Image Patches | CIFAR10 (32 pixel resolution). Left,** Raw variance of generated patches along true eigenbases during training. **Right,** Scaling relationship between the target variance of eigenmode versus mode emergence time (harmonic mean criterion). Each row corresponds to a different patch size and stride used for extracting patches from images.

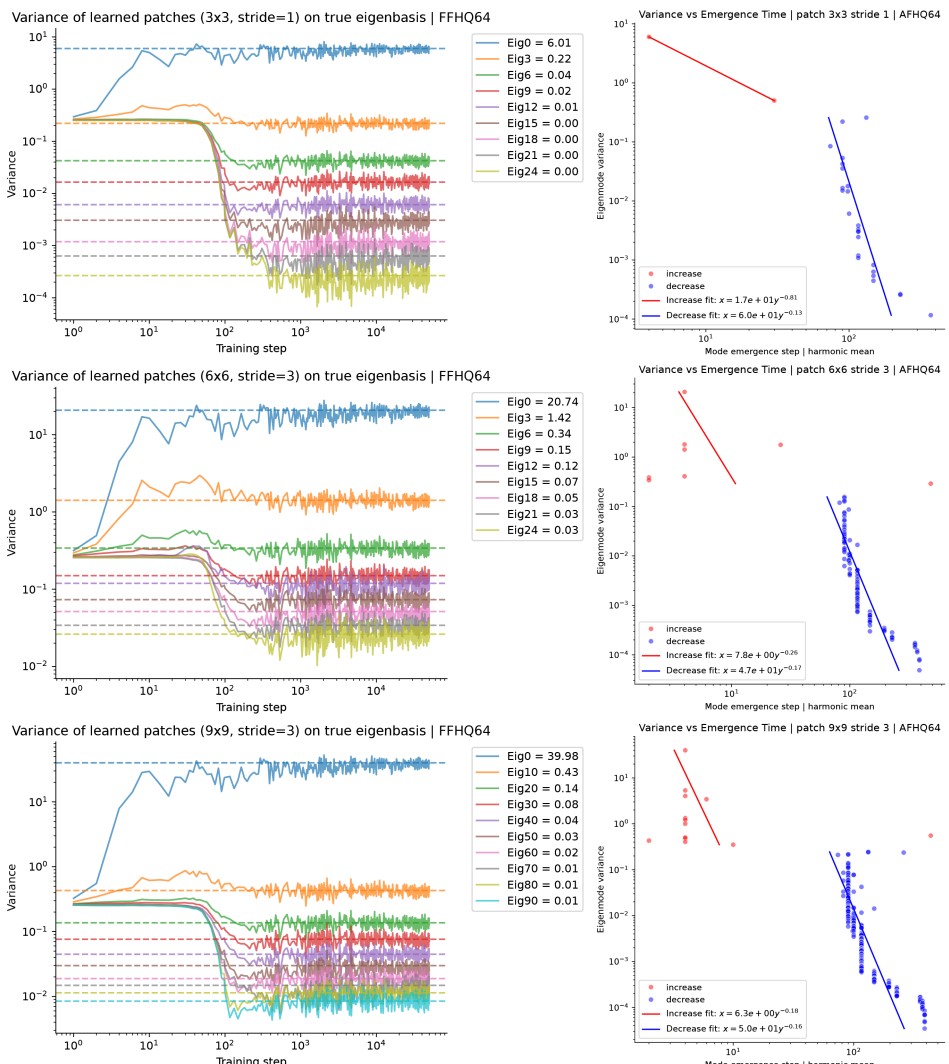

Figure 20: **Spectral Bias in CNN-Based Diffusion Learning: Variance Dynamics in Image Patches | FFHQ (64 pixel resolution). Left,** Raw variance of generated patches along true eigenbases during training. **Right,** Scaling relationship between the target variance of eigenmode versus mode emergence time (harmonic mean criterion). Each row corresponds to a different patch size and stride used for extracting patches from images.

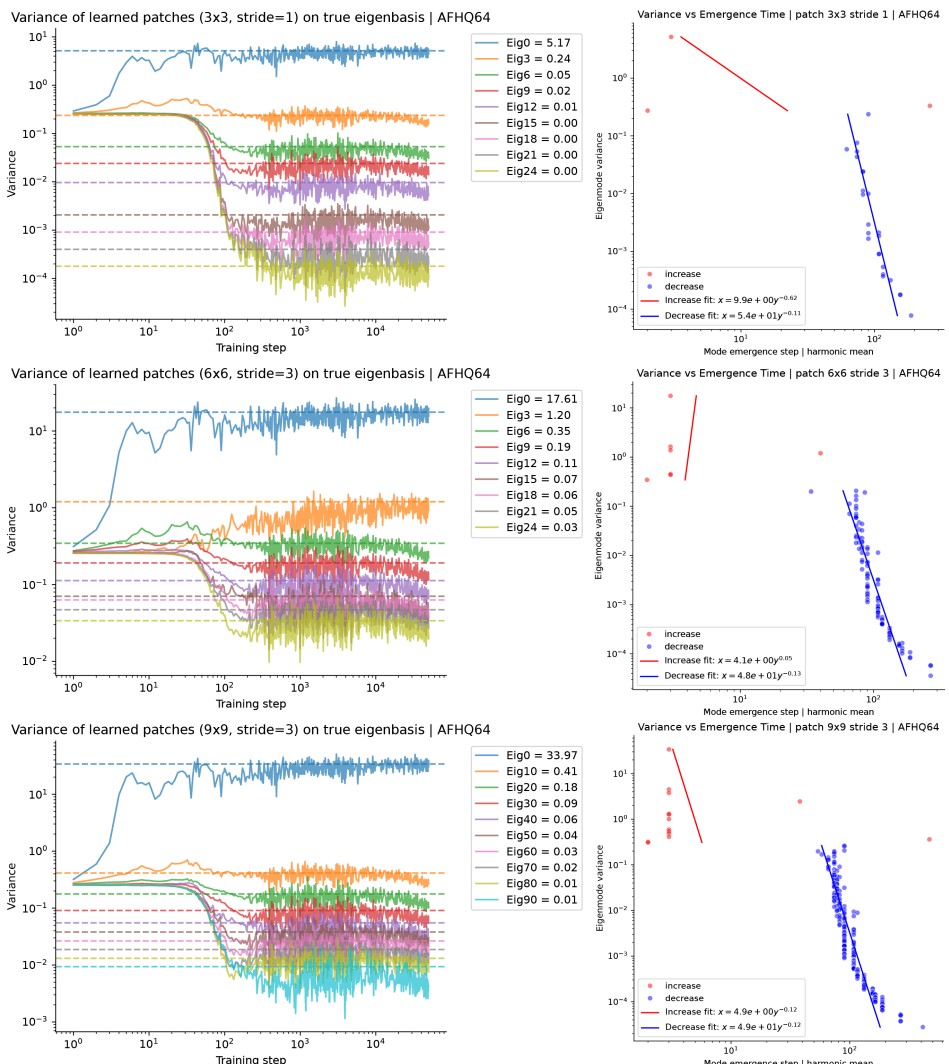

Figure 21: **Spectral Bias in CNN-Based Diffusion Learning: Variance Dynamics in Image Patches | AFHQv2 (64 pixel resolution). Left,** Raw variance of generated patches along true eigenbases during training. **Right,** Scaling relationship between the target variance of eigenmode versus mode emergence time (harmonic mean criterion). Each row corresponds to a different patch size and stride used for extracting patches from images.

### B.2.4 CNN architecture ablation experiments

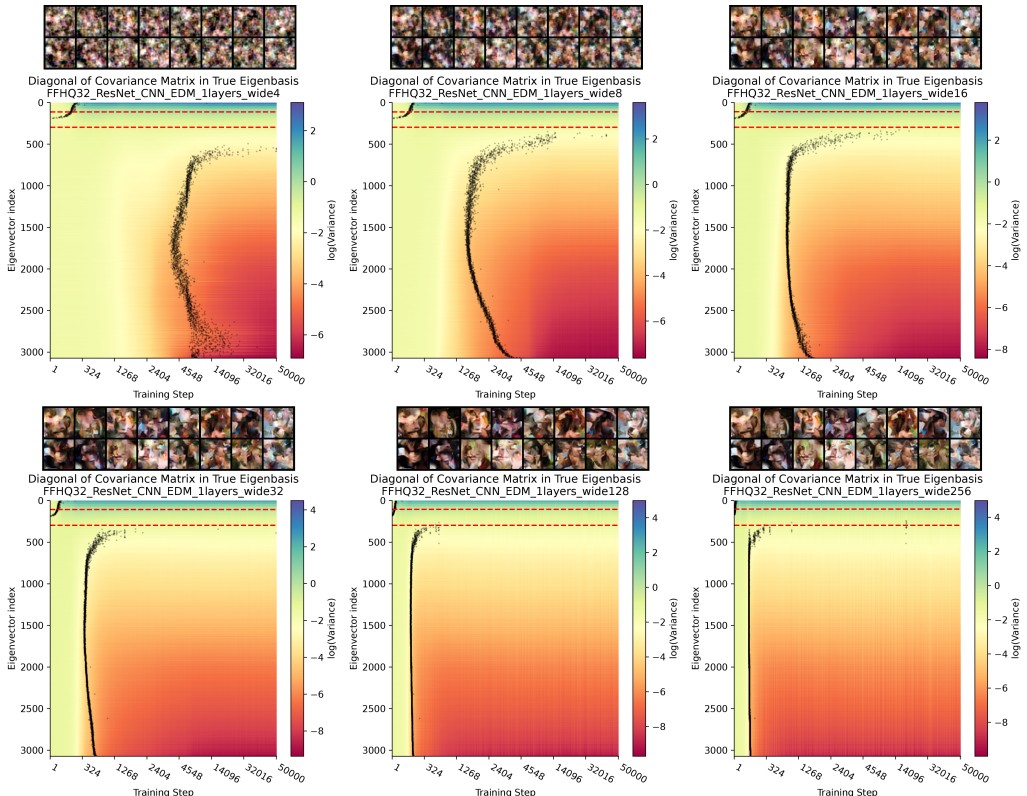

Figure 22: **Spectral Learning Dynamics of CNN-ResNet (FFHQ32) with architectural variation - 1 layer, channel numebr variation,** $ch = 4, 8, 16, 32, 128, 256$. (same analysis procedure and format as main Fig. 3C.) Each panel describes one architecture, **Top** Sample final images **Bottom** Heatmap of variance trajectories along all eigenmodes, with dots marking mode emergence times $\tau^*$ (first-passage time at the geometric mean of initial and final variances). The gray zone (0.5–2× target variance) indicates modes starting too close to their target, causing unreliable $\tau^*$ estimates.

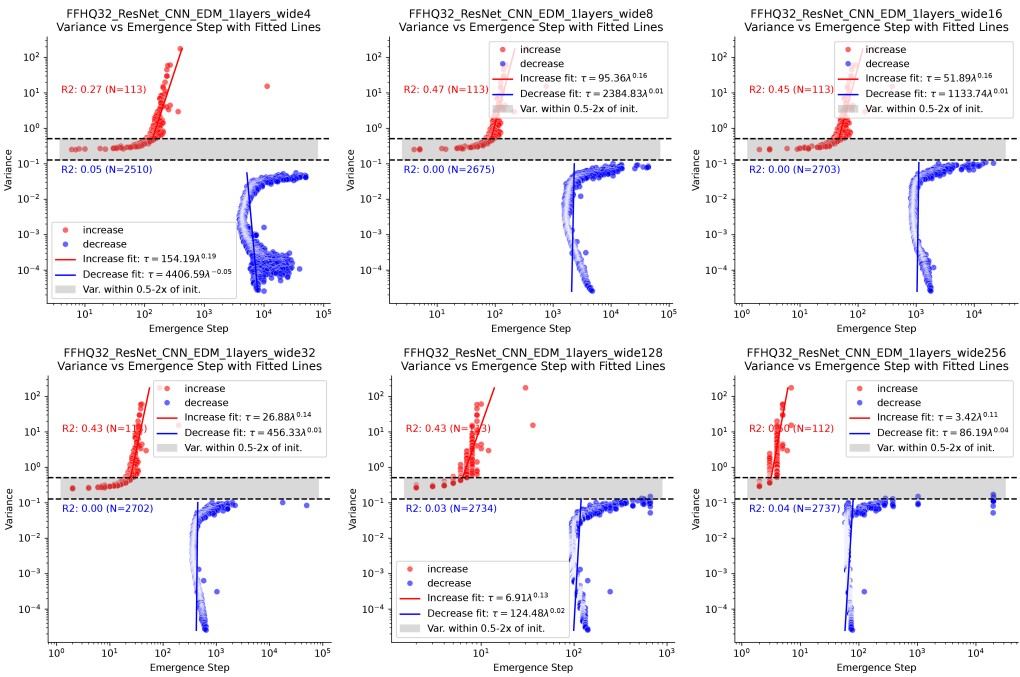

Figure 23: **Spectral Learning Dynamics of CNN-ResNet (FFHQ32) with architectural variation - 1 layer, channel numebr variation,** $ch = 4, 8, 16, 32, 128, 256$**.** (same analysis procedure and format as main Fig. 3D.) Each panel describes one architecture. Power-law scaling of $\tau^*$ versus target variance $\lambda_k$, excluding gray-zone eigenmodes for stability.

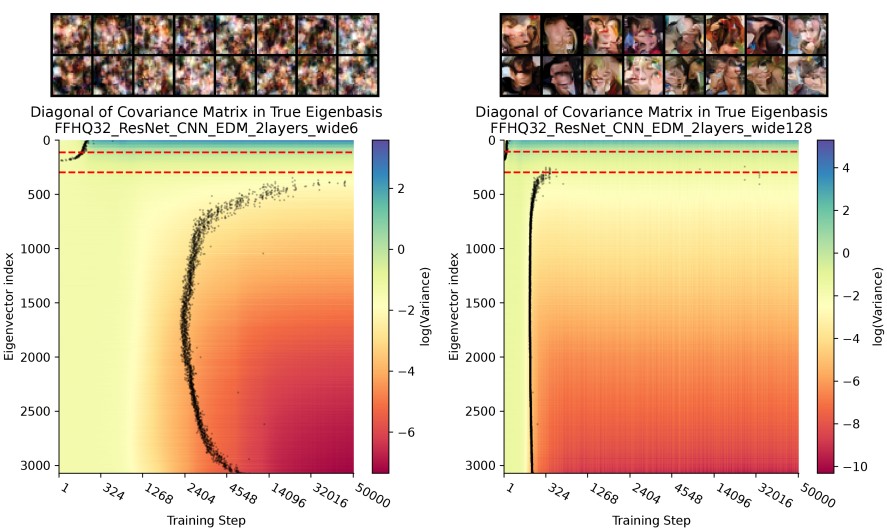

Figure 24: **Spectral Learning Dynamics of CNN-ResNet (FFHQ32) with architectural variation - 2 layer, channel numebr variation** $ch = 6, 128$**.** (same analysis procedure and format as Fig. 22)

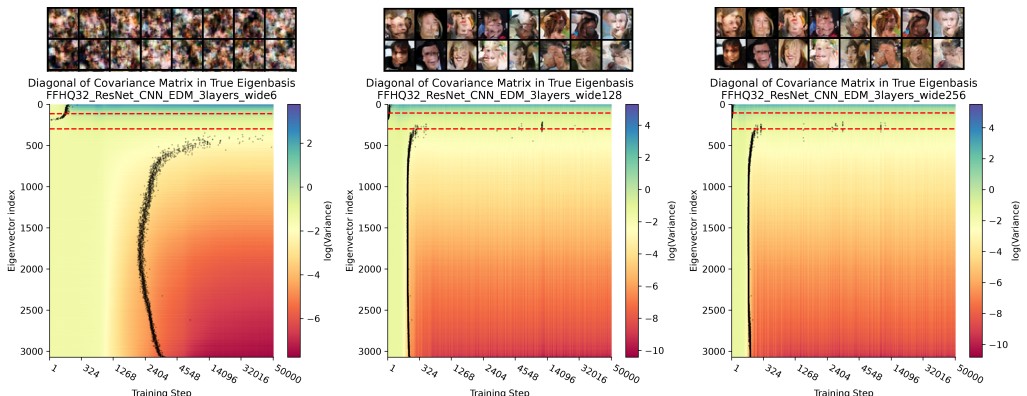

Figure 25: **Spectral Learning Dynamics of CNN-ResNet (FFHQ32) with architectural variation - 3 layer, channel numebr variation** $ch = 6, 128, 256$**.** (same analysis procedure and format as Fig. 22)

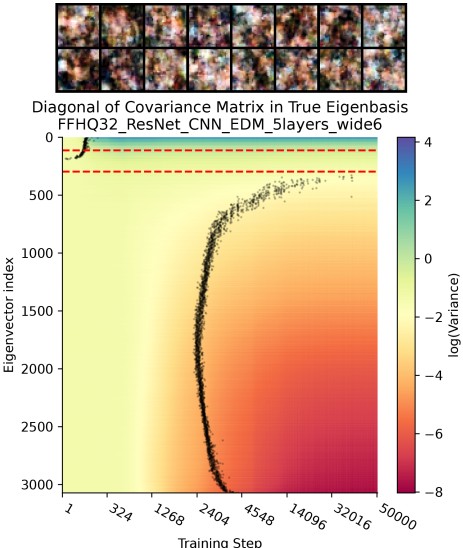

Figure 26: **Spectral Learning Dynamics of CNN-ResNet (FFHQ32) with architectural variation - 5 layer, channel numebr variation** $ch = 6$**.** (same analysis procedure and format as Fig. 22)

# C Detailed Derivation: General analysis

## C.1 General Property of Linear Regression

### C.1.1 Gaussian equivalence

**Lemma C.1.** *For a general linear regression problem, where* $\mathbf{x}, \mathbf{y}$ *come from an arbitrary joint distribution* $p(\mathbf{x}, \mathbf{y})$ *with finite moments,*

$$\mathcal{L} = \mathbb{E}_{\mathbf{x},\mathbf{y}} \|\mathbf{W}\mathbf{x} + \mathbf{b} - \mathbf{y}\|^2$$

*then its optimal solution and gradient only depend on the first two moments of* $\mathbf{x}, \mathbf{y}$.

*Proof.* Let the error be $\mathbf{e} = \mathbf{W}\mathbf{x} + \mathbf{b} - \mathbf{y}$, then

$$
\begin{aligned}
\nabla_{\mathbf{W}}\mathcal{L} &= \frac{\partial}{\partial \mathbf{W}} \mathbb{E}_{\mathbf{x},\mathbf{y}} \left[ \mathbf{e}^\top \mathbf{e} \right] \\
&= 2\mathbb{E} \left[ \mathbf{e}\,\mathbf{x}^\top \right] \\
&= 2\mathbb{E} \left[ (\mathbf{W}\mathbf{x} + \mathbf{b} - \mathbf{y})\mathbf{x}^\top \right] \\
&= 2 \left( \mathbf{W}\mathbb{E} \left[ \mathbf{x}\mathbf{x}^\top \right] + \mathbf{b}\mathbb{E} \left[ \mathbf{x}^\top \right] - \mathbb{E} \left[ \mathbf{y}\mathbf{x}^\top \right] \right) \\
&= 2 \left( \mathbf{W}(\Sigma_{xx} + \mu_x \mu_x^T) + \mathbf{b}\mu_x^T - (\Sigma_{yx} + \mu_y \mu_x^T) \right) \\
&= 2 \left( \mathbf{W}(\Sigma_{xx} + \mu_x \mu_x^T) + (\mathbf{b} - \mu_y)\mu_x^T - \Sigma_{yx} \right) \\
&= 2(\mathbf{W}\Sigma_{xx} - \Sigma_{yx}) + 2(\mathbf{W}\mu_x + \mathbf{b} - \mu_y)\mu_x^T \\
&= 2(\mathbf{W}\Sigma_{xx} - \Sigma_{yx}) + \nabla_{\mathbf{b}}\mathcal{L}\mu_x^T
\end{aligned}
$$

$$
\begin{aligned}
\nabla_{\mathbf{b}}\mathcal{L} &= \frac{\partial}{\partial \mathbf{b}} \mathbb{E} \left[ \mathbf{e}^\top \mathbf{e} \right] \\
&= 2\mathbb{E} \left[ \mathbf{e} \right] \\
&= 2\mathbb{E} \left[ \mathbf{W}\mathbf{x} + \mathbf{b} - \mathbf{y} \right] \\
&= 2(\mathbf{W}\mu_x + \mathbf{b} - \mu_y)
\end{aligned}
$$

We used the fact that

$$
\begin{aligned}
\mathbb{E} \left[ \mathbf{x} \right] &= \mu_x \\
\mathbb{E} \left[ \mathbf{y} \right] &= \mu_y \\
\mathbb{E} \left[ \mathbf{y}\mathbf{x}^\top \right] &= \Sigma_{yx} + \mu_y \mu_x^\top \\
\mathbb{E} \left[ \mathbf{x}\mathbf{x}^\top \right] &= \Sigma_{xx} + \mu_x \mu_x^\top
\end{aligned}
$$

Setting the gradient to zero, we get optimal values

$$\mathbf{W}^* = \Sigma_{yx}\Sigma_{xx}^{-1} \tag{11}$$
$$\mathbf{b}^* = \mu_y - \mathbf{W}^*\mu_x \tag{12}$$

The gradient flow dynamics read

$$\frac{d}{d\tau}\mathbf{W} = -2\eta(\mathbf{W}\Sigma_{xx} - \Sigma_{yx}) - 2\eta\nabla_{\mathbf{b}}\mathcal{L}\mu_x^T \tag{13}$$

$$\frac{d}{d\tau}\mathbf{b} = -2\eta(\mathbf{W}\mu_x + \mathbf{b} - \mu_y) \tag{14}$$

The $\Sigma_{xx}$ determines the gradient flow dynamics and convergence rate of $\mathbf{W}$, while $\Sigma_{xx}^{-1}\Sigma_{yx}$ determines the target level or optimal solution of the regression. $\qquad\square$

*Remark* C.2. For all the loss variants with linear denoiser, the loss is of this class. We can write down the gradient structure and optimal values of $\mathbf{W}$ and $\mathbf{b}$ by plugging in the mean and covariance of input $\mathbf{x}$ and predicting target $\mathbf{y}$.

**Lemma C.3.** *For two independent random variables $X, Y \in \mathbb{R}^d$, we have*

$$Cov(aX + bY, cX + dY) = ac\Sigma_X + bd\Sigma_Y \tag{15}$$

*Proof.* This can be proved using bilinearity of covariance and independence which entails $\Sigma_{XY} = 0$.

$$
\begin{aligned}
\text{Cov}(aX + bY, cX + dY) &= a\text{Cov}(X, cX + dY) + b\text{Cov}(Y, cX + dY) \\
&= ac\text{Cov}(X, X) + ad\text{Cov}(X, Y) + bc\text{Cov}(Y, X) + bd\text{Cov}(Y, Y) \\
&= ac\,\Sigma_X + bd\,\Sigma_Y
\end{aligned}
$$

$\square$

*Remark* C.4. For diffusion training loss, the clean image samples and the noise patterns are designed to be sampled independently. Thus we can use (15) to calculate the covariance of input and outputs.

Using these two lemmas, the gradient structure and learning dynamics of various loss functions can be easily read off.

## C.2 General analysis of the gradient learning of linear predictor

### C.2.1 Denoising as Ridge Regression

**Lemma C.5** (Diffusion learning as Ridge regression)**.** *For linear denoisers $\mathbf{D}(\mathbf{x}, \sigma) = \mathbf{W}\mathbf{x} + \mathbf{b}$, at the full batch limit, the denoising score matching loss* (2) *is equivalent to the following Ridge regression loss.*

$$\mathcal{L}_\sigma = \mathbb{E}_{\mathbf{x}}\|\mathbf{W}\mathbf{x} + \mathbf{b} - \mathbf{x}\|^2 + \sigma^2\|\mathbf{W}\|^2 \tag{16}$$

*Proof.*

$$
\begin{aligned}
\mathcal{L}_\sigma &= \mathbb{E}_{\mathbf{x}\sim p_0, \mathbf{z}\sim\mathcal{N}(0,\mathbf{I})}\|\mathbf{D}_\theta(\mathbf{x} + \sigma\mathbf{z}; \sigma) - \mathbf{x}\|^2 \\
&= \mathbb{E}_{\mathbf{x}\sim p_0, \mathbf{z}\sim\mathcal{N}(0,\mathbf{I})}\|\mathbf{W}(\mathbf{x} + \sigma\mathbf{z}) + \mathbf{b} - \mathbf{x}\|^2 \\
&= \mathbb{E}_{\mathbf{x}\sim p_0, \mathbf{z}\sim\mathcal{N}(0,\mathbf{I})}\text{Tr}\big[\big(\mathbf{W}(\mathbf{x} + \sigma\mathbf{z}) + \mathbf{b} - \mathbf{x}\big)\big(\mathbf{W}(\mathbf{x} + \sigma\mathbf{z}) + \mathbf{b} - \mathbf{x}\big)^\top\big] \\
&= \mathbb{E}_{\mathbf{x}\sim p_0, \mathbf{z}\sim\mathcal{N}(0,\mathbf{I})}\text{Tr}\big[\big(\mathbf{W}(\mathbf{x} + \sigma\mathbf{z}) + \mathbf{b} - \mathbf{x}\big)\big((\mathbf{x} + \sigma\mathbf{z})^\top\mathbf{W}^\top + \mathbf{b}^\top - \mathbf{x}^\top\big)\big] \\
&= \mathbb{E}_{\mathbf{x}\sim p_0, \mathbf{z}\sim\mathcal{N}(0,\mathbf{I})}\text{Tr}\big[\big(\mathbf{W}\mathbf{x} + \mathbf{b} - \mathbf{x} + \sigma\mathbf{W}\mathbf{z}\big)\big(\mathbf{x}^\top\mathbf{W}^\top + \mathbf{b}^\top - \mathbf{x}^\top + \sigma\mathbf{z}^\top\mathbf{W}^\top\big)\big] \\
&= \mathbb{E}_{\mathbf{x}\sim p_0, \mathbf{z}\sim\mathcal{N}(0,\mathbf{I})}\text{Tr}\big[\big(\mathbf{W}\mathbf{x} + \mathbf{b} - \mathbf{x}\big)\big(\mathbf{W}\mathbf{x} + \mathbf{b} - \mathbf{x}\big)^\top + 2\sigma\mathbf{W}\mathbf{z}\big(\mathbf{W}\mathbf{x} + \mathbf{b} - \mathbf{x}\big)^\top + \sigma^2\mathbf{W}\mathbf{z}\mathbf{z}^\top\mathbf{W}^\top\big] \\
&= \mathbb{E}_{\mathbf{x}\sim p_0}\|\mathbf{W}\mathbf{x} + \mathbf{b} - \mathbf{x}\|^2 + \mathbb{E}_{\mathbf{x}\sim p_0, \mathbf{z}\sim\mathcal{N}(0,\mathbf{I})}\text{Tr}\big[2\sigma\mathbf{W}\mathbf{z}\big(\mathbf{W}\mathbf{x} + \mathbf{b} - \mathbf{x}\big)^\top + \sigma^2\mathbf{W}\mathbf{z}\mathbf{z}^\top\mathbf{W}^\top\big] \\
&= \mathbb{E}_{\mathbf{x}\sim p_0}\|\mathbf{W}\mathbf{x} + \mathbf{b} - \mathbf{x}\|^2 + \sigma^2\|\mathbf{W}\|^2
\end{aligned}
$$

This derivation depends on the linearity of the denoiser and the white Gaussian nature of noise $\mathbf{z} \sim \mathcal{N}(0, \mathbf{I})$. $\square$

*Remark* C.6. This equivalence reveals that for diffusion models the additive white noise can be viewed as L2 regularization on the weights of the auto-encoding regression problem, where $\sigma^2$ functions as regularization strength $\lambda$. Thus per classic analysis of Ridge regression, we can see that at higher noise scales, fewer modes of the data will be "resolved".

Further, if the noise is zero-mean but not white, the final term $\text{Tr}\big[\sigma^2\mathbf{W}\mathbf{z}\mathbf{z}^\top\mathbf{W}^\top\big]$ will impose other types of weighted regularization on the weights.

## C.3 General Analysis of the Denoising Score Matching Objective

To simplify, we first consider a fixed $\sigma$, ignoring the nonlinear dependency on it. Consider our score, denoiser function approximators as a linear function $\mathbf{D}(\mathbf{x}; \sigma) = \mathbf{b} + \mathbf{W}\mathbf{x}$. Expanding the per-sample DSM loss, we get

$$
\begin{aligned}
&\|\mathbf{D}(\mathbf{x}_0 + \sigma\mathbf{z}; \sigma) - \mathbf{x}_0\|^2 \\
&= \|\mathbf{b} + \mathbf{W}(\mathbf{x}_0 + \sigma\mathbf{z}) - \mathbf{x}_0\|^2 \\
&= \|\mathbf{b} + \mathbf{W}\sigma\mathbf{z} + (\mathbf{W} - \mathbf{I})\mathbf{x}_0\|^2 \\
&= (\mathbf{b} + \mathbf{W}\sigma\mathbf{z})^T(\mathbf{b} + \mathbf{W}\sigma\mathbf{z}) + \mathbf{x}_0^T(\mathbf{W} - \mathbf{I})^T(\mathbf{W} - \mathbf{I})\mathbf{x}_0 + 2(\mathbf{b} + \mathbf{W}\sigma\mathbf{z})^T(\mathbf{W} - \mathbf{I})\mathbf{x}_0 \\
&= \mathbf{b}^T\mathbf{b} + 2\sigma\mathbf{b}^T\mathbf{W}\mathbf{z} + \sigma^2\mathbf{z}^T\mathbf{W}^T\mathbf{W}\mathbf{z} + \mathbf{x}_0^T(\mathbf{W} - \mathbf{I})^T(\mathbf{W} - \mathbf{I})\mathbf{x}_0 \\
&\qquad + 2\mathbf{b}^T(\mathbf{W} - \mathbf{I})\mathbf{x}_0 + 2\sigma\mathbf{z}^T\mathbf{W}^T(\mathbf{W} - \mathbf{I})\mathbf{x}_0 \\
&= \mathbf{b}^T\mathbf{b} + 2\sigma\mathbf{b}^T\mathbf{W}\mathbf{z} + \sigma^2\text{Tr}[\mathbf{W}^T\mathbf{W}\mathbf{z}\mathbf{z}^T] + \text{Tr}[(\mathbf{W} - \mathbf{I})^T(\mathbf{W} - \mathbf{I})\mathbf{x}_0\mathbf{x}_0^T] \\
&\qquad + 2\mathbf{b}^T(\mathbf{W} - \mathbf{I})\mathbf{x}_0 + 2\sigma\text{Tr}[\mathbf{W}^T(\mathbf{W} - \mathbf{I})\mathbf{x}_0\mathbf{z}^T] \\
&= \|\mathbf{b} - \mathbf{x}_0\|^2 + \|\mathbf{W}(\mathbf{x}_0 + \sigma\mathbf{z})\|^2 + 2\mathbf{W}(\mathbf{x}_0 + \sigma\mathbf{z})(\mathbf{b} - \mathbf{x}_0)^T
\end{aligned}
$$

**Full batch limit**   Here we take the full-batch expectation over $\mathbf{z}, \mathbf{x}_0$, where $\mathbf{z} \sim \mathcal{N}(0, \mathbf{I}), \mathbf{x}_0 \sim p(\mathbf{x}_0)$. Their moments are the following.

$$
\begin{aligned}
\mathbb{E}[\mathbf{z}] &= 0 \\
\mathbb{E}[\mathbf{z}\mathbf{z}^T] &= \mathbf{I} \\
\mathbb{E}[\mathbf{x}_0] &= \mu \\
\mathbb{E}[\mathbf{x}_0\mathbf{x}_0^T] &= \mu\mu^T + \Sigma \\
\mathbb{E}[\mathbf{x}_0\mathbf{z}^T] &= 0
\end{aligned}
$$

Note that the data do not need to be Gaussian, as long as the expectations or the first two moments are as computed above, the results should be the same. In a sense, if our function approximator is linear, then we just care about the Gaussian part of data.

$$
\begin{aligned}
\mathcal{L} &= \mathbb{E}_{\mathbf{x}_0 \sim p(\mathbf{x}_0), \mathbf{z} \sim \mathcal{N}(0, \mathbf{I})} \|\mathbf{D}(\mathbf{x}_0 + \sigma\mathbf{z}; \sigma) - \mathbf{x}_0\|^2 \\
&= \mathbf{b}^T\mathbf{b} + \sigma^2\text{Tr}[\mathbf{W}^T\mathbf{W}] + \text{Tr}[(\mathbf{W} - \mathbf{I})^T(\mathbf{W} - \mathbf{I})(\mu\mu^T + \Sigma)] + 2\mathbf{b}^T(\mathbf{W} - \mathbf{I})\mu
\end{aligned}
\tag{17}
$$

We can simplify the objective by completing the square,

$$
\begin{aligned}
\mathcal{L} &= \mathbb{E}_{\mathbf{x}_0 \sim p(\mathbf{x}_0), \mathbf{z} \sim \mathcal{N}(0, \mathbf{I})} \|\mathbf{D}(\mathbf{x}_0 + \sigma\mathbf{z}; \sigma) - \mathbf{x}_0\|^2 \\
&= \|\mathbf{b} + (\mathbf{W} - \mathbf{I})\mu\|_2^2 - \mu^T(\mathbf{W} - \mathbf{I})^T(\mathbf{W} - \mathbf{I})\mu + \sigma^2\text{Tr}[\mathbf{W}^T\mathbf{W}] + \text{Tr}[(\mathbf{W} - \mathbf{I})^T(\mathbf{W} - \mathbf{I})(\mu\mu^T + \Sigma)] \\
&= \|\mathbf{b} - (\mathbf{I} - \mathbf{W})\mu\|_2^2 + \sigma^2\text{Tr}[\mathbf{W}^T\mathbf{W}] + \text{Tr}[(\mathbf{W} - \mathbf{I})^T(\mathbf{W} - \mathbf{I})\Sigma] \tag{18} \\
&= \|\mathbf{b} - (\mathbf{I} - \mathbf{W})\mu\|_2^2 + \text{Tr}[\mathbf{W}^T\mathbf{W}(\sigma^2\mathbf{I} + \Sigma)] - 2\text{Tr}[\mathbf{W}^T\Sigma] + \text{Tr}[\Sigma] \tag{19}
\end{aligned}
$$

**Gradient**   The gradients from loss to the weight and bias read

$$
\nabla_{\mathbf{b}}\mathcal{L} = 2(\mathbf{b} - (\mathbf{I} - \mathbf{W})\mu) \tag{20}
$$

$$
\nabla_{\mathbf{W}}\mathcal{L} = -2\Sigma + 2\mathbf{W}(\sigma^2\mathbf{I} + \Sigma) + [2\mathbf{W}\mu\mu^T + 2(\mathbf{b} - \mu)\mu^T] \tag{21}
$$

**Global optimum**   Examining the quadratic loss, and setting the gradient to zero, we can see the optimal parameters are the following,

$$
\mathbf{b}^* = (\mathbf{I} - \mathbf{W}^*)\mu \tag{22}
$$

$$
\mathbf{W}^* = \Sigma(\sigma^2\mathbf{I} + \Sigma)^{-1} \tag{23}
$$

which recovers the Gaussian denoiser.

**Gradient flow dynamics** Next, let's examine the learning dynamics by gradient descent. We consider the continuous-time limit, i.e. gradient flow. Denoting the learning rate $\eta$ and continuous training time $\tau$, their gradient flow dynamics are

$$\frac{d}{d\tau}\mathbf{b} = -\eta\nabla_{\mathbf{b}}\mathcal{L} \tag{24}$$

$$\frac{d}{d\tau}\mathbf{W} = -\eta\nabla_{\mathbf{W}}\mathcal{L} \tag{25}$$

## C.4 Gradient Structure of Other Variants of Loss

Here we generalize our analysis to many popular training losses of diffusion models, including flow matching. We provide a detailed table for different variants and the learning dynamics of the linear denoiser.

Table 3: **Variants of the diffusion training loss and their linear solution and gradient structure.** Here $\Sigma_{xx}$ denotes the covariance of the input to the linear predictor; $\Sigma_{yx}$ is the covariance between target and input, not to be confused with the covariance of training sample $\mathbf{x}_0$. $w_k^*$ denotes the optimal weight projected onto principal component $\mathbf{u}_k$, $w_k^* := \mathbf{u}_k^T\mathbf{W}^*\mathbf{u}_k$; $1/\tau^*$ represents the convergence speed of a target mode. We abbreviated away the expectation over data and noise in our loss

| | EDM | X-pred | EPS-pred | V-pred | Flow matching |
|---|---|---|---|---|---|
| Loss | $\|\mathbf{D}_\theta(\mathbf{x}+\sigma\epsilon;\sigma)-\mathbf{x}\|^2$ | $\|\mathbf{F}_\theta(\alpha_t\mathbf{x}+\sigma_t\epsilon;t)-\mathbf{x}\|^2$ | $\|\mathbf{F}_\theta(\alpha_t\mathbf{x}+\sigma_t\epsilon;t)-\epsilon\|^2$ | $\|\mathbf{F}_\theta(\alpha_t\mathbf{x}+\sigma_t\epsilon;t)-(\alpha_t\epsilon-\sigma_t\mathbf{x})\|^2$ | $\|\mathbf{u}_\theta((1-t)\mathbf{x}_0+t\mathbf{x}_1,t)-(\mathbf{x}_1-\mathbf{x}_0)\|^2$ |
| $\Sigma_{xx}$ | $\Sigma+\sigma^2 I$ | $\alpha_t^2\Sigma+\sigma_t^2 I$ | $\alpha_t^2\Sigma+\sigma_t^2 I$ | $\alpha_t^2\Sigma+\sigma_t^2 I$ | $t^2\Sigma+(1-t)^2 I$ |
| $\Sigma_{yx}$ | $\Sigma$ | $\alpha_t\Sigma$ | $\sigma_t I$ | $\alpha_t\sigma_t(I-\Sigma)$ | $t\Sigma-(1-t)I$ |
| $\mathbf{W}^*$ | $(\Sigma+\sigma^2 I)^{-1}\Sigma$ | $\alpha_t(\alpha_t^2\Sigma+\sigma_t^2 I)^{-1}\Sigma$ | $\sigma_t(\alpha_t^2\Sigma+\sigma_t^2 I)^{-1}$ | $\alpha_t\sigma_t(\alpha_t^2\Sigma+\sigma_t^2 I)^{-1}(I-\Sigma)$ | $(t^2\Sigma+(1-t)^2 I)^{-1}(t\Sigma-(1-t)I)$ |
| $\mathbf{b}^*$ | $\sigma^2(\Sigma+\sigma^2 I)^{-1}\mu$ | $\sigma_t^2(\alpha_t^2\Sigma+\sigma_t^2 I)^{-1}\mu$ | $-\alpha_t\sigma_t(\alpha_t^2\Sigma+\sigma_t^2 I)^{-1}\mu$ | $-\sigma_t(\alpha_t^2+\sigma_t^2)(\alpha_t^2\Sigma+\sigma_t^2 I)^{-1}\mu$ | $(1-t)(t^2\Sigma+(1-t)^2 I)^{-1}\mu$ |
| $w_k^*$ | $\frac{\lambda_k}{\lambda_k+\sigma^2}$ | $\frac{\alpha_t\lambda_k}{\alpha_t^2\lambda_k+\sigma_t^2}$ | $\frac{\sigma_t}{\alpha_t^2\lambda_k+\sigma_t^2}$ | $\frac{\alpha_t\sigma_t(1-\lambda_k)}{\alpha_t^2\lambda_k+\sigma_t^2}$ | $\frac{t\lambda_k-(1-t)}{t^2\lambda_k+(1-t)^2}$ |
| $1/\tau^*$ | $\lambda_k+\sigma^2$ | $\alpha_t^2\lambda_k+\sigma_t^2$ | $\alpha_t^2\lambda_k+\sigma_t^2$ | $\alpha_t^2\lambda_k+\sigma_t^2$ | $t^2\lambda_k+(1-t)^2$ |

**Denoiser / clean image / $\mathrm{x}0$ prediction (EDM) loss**

$$\|\mathbf{D}_\theta(\mathbf{x}+\sigma\epsilon;\sigma)-\mathbf{x}\|^2$$

Moments of input-output

$$\mu_x = \mu, \ \mu_y = \mu$$
$$\Sigma_{xx} = \mathrm{Cov}(\mathbf{x}+\sigma\epsilon, \mathbf{x}+\sigma\epsilon) = \Sigma + \sigma^2 I$$
$$\Sigma_{yx} = \mathrm{Cov}(\mathbf{x}, \mathbf{x}+\sigma\epsilon) = \Sigma$$

Optimum

$$\mathbf{W}^* = \Sigma_{xx}^{-1}\Sigma_{yx} = (\Sigma+\sigma^2 I)^{-1}\Sigma$$
$$\mathbf{b}^* = \mu_y - \mathbf{W}^*\mu_x$$
$$= (I - \mathbf{W}^*)\mu$$
$$= \sigma^2(\Sigma+\sigma^2 I)^{-1}\mu$$

**Noise / eps prediction (EDM) loss**

$$\|\mathbf{F}_\theta(\mathbf{x}+\sigma\epsilon;\sigma)-\epsilon\|^2$$

Moments of input-output

$$\mu_x = \mu, \ \mu_y = 0$$
$$\Sigma_{xx} = \mathrm{Cov}(\mathbf{x}+\sigma\epsilon, \mathbf{x}+\sigma\epsilon) = \Sigma + \sigma^2 I$$
$$\Sigma_{yx} = \mathrm{Cov}(\epsilon, \mathbf{x}+\sigma\epsilon) = \sigma I$$

Optimum

$$\mathbf{W}^* = \Sigma_{xx}^{-1}\Sigma_{yx} = \sigma(\Sigma+\sigma^2 I)^{-1}$$
$$\mathbf{b}^* = \mu_y - \mathbf{W}^*\mu_x$$
$$= -\mathbf{W}^*\mu$$
$$= -\sigma(\Sigma+\sigma^2 I)^{-1}\mu$$

**Denoiser / clean image / $\mathrm{x}0$ prediction loss (variance preserving)**

$$\|\mathbf{D}_\theta(\alpha_t\mathbf{x}+\sigma_t\epsilon;\sigma)-\mathbf{x}\|^2$$

Moments of input-output

$$\mu_x = \alpha_t\mu, \ \mu_y = \mu$$
$$\Sigma_{xx} = \mathrm{Cov}(\alpha_t\mathbf{x}+\sigma_t\epsilon, \alpha_t\mathbf{x}+\sigma_t\epsilon) = \alpha_t^2\Sigma + \sigma_t^2 I$$
$$\Sigma_{yx} = \mathrm{Cov}(\mathbf{x}, \alpha_t\mathbf{x}+\sigma_t\epsilon) = \alpha_t\Sigma$$

Optimum

$$
\begin{aligned}
\mathbf{W}^* &= \Sigma_{xx}^{-1}\Sigma_{yx} = \alpha_t(\alpha_t^2\Sigma + \sigma_t^2 I)^{-1}\Sigma \\
\mathbf{b}^* &= \mu_y - \mathbf{W}^*\mu_x \\
&= \mu - \alpha_t\mathbf{W}^*\mu \\
&= (I - \alpha_t\mathbf{W}^*)\mu \\
&= (I - \alpha_t^2\Sigma(\alpha_t^2\Sigma + \sigma_t^2 I)^{-1})\mu \\
&= \sigma_t^2(\alpha_t^2\Sigma + \sigma_t^2 I)^{-1}\mu
\end{aligned}
$$

**Noise / eps prediction loss (variance preserving)**

$$
\|\mathbf{F}_\theta(\alpha_t\mathbf{x} + \sigma_t\epsilon; t) - \epsilon\|^2
$$

Moments of input-output

$$
\begin{aligned}
\mu_x &= \alpha_t\mu, \ \mu_y = 0 \\
\Sigma_{xx} &= \mathrm{Cov}(\alpha_t\mathbf{x} + \sigma_t\epsilon, \alpha_t\mathbf{x} + \sigma_t\epsilon) = \alpha_t^2\Sigma + \sigma_t^2 I \\
\Sigma_{yx} &= \mathrm{Cov}(\epsilon, \alpha_t\mathbf{x} + \sigma_t\epsilon) = \sigma_t I
\end{aligned}
$$

Optimum

$$
\begin{aligned}
\mathbf{W}^* &= \Sigma_{xx}^{-1}\Sigma_{yx} = \sigma_t(\alpha_t^2\Sigma + \sigma_t^2 I)^{-1} \\
\mathbf{b}^* &= \mu_y - \mathbf{W}^*\mu_x \\
&= -\alpha_t\mathbf{W}^*\mu \\
&= -\alpha_t\sigma_t(\alpha_t^2\Sigma + \sigma_t^2 I)^{-1}\mu
\end{aligned}
$$

**Velocity / V-prediction loss (variance preserving)**

$$
\|\mathbf{F}_\theta(\alpha_t\mathbf{x} + \sigma_t\epsilon; t) - (\alpha_t\epsilon - \sigma_t\mathbf{x})\|^2
$$

Moments of input-output

$$
\begin{aligned}
\mu_x &= \alpha_t\mu, \ \mu_y = -\sigma_t\mu \\
\Sigma_{xx} &= \mathrm{Cov}(\alpha_t\mathbf{x} + \sigma_t\epsilon, \alpha_t\mathbf{x} + \sigma_t\epsilon) = \alpha_t^2\Sigma + \sigma_t^2 I \\
\Sigma_{yx} &= \mathrm{Cov}(\alpha_t\epsilon - \sigma_t\mathbf{x}, \alpha_t\mathbf{x} + \sigma_t\epsilon) = -\alpha_t\sigma_t\Sigma + \alpha_t\sigma_t I = \alpha_t\sigma_t(I - \Sigma)
\end{aligned}
$$

Optimum

$$
\begin{aligned}
\mathbf{W}^* &= \Sigma_{xx}^{-1}\Sigma_{yx} \\
&= \alpha_t\sigma_t(\alpha_t^2\Sigma + \sigma_t^2 I)^{-1}(I - \Sigma) \\
\mathbf{b}^* &= \mu_y - \mathbf{W}^*\mu_x \\
&= -\sigma_t\mu - \alpha_t\mathbf{W}^*\mu \\
&= -(\sigma_t I + \alpha_t\mathbf{W}^*)\mu \\
&= -\left(\sigma_t I + \alpha_t^2\sigma_t(\alpha_t^2\Sigma + \sigma_t^2 I)^{-1}(I - \Sigma)\right)\mu \\
&= -\sigma_t\left(I + \alpha_t^2(\alpha_t^2\Sigma + \sigma_t^2 I)^{-1}(I - \Sigma)\right)\mu \\
&= -\sigma_t\left(\alpha_t^2\Sigma + \sigma_t^2 I + \alpha_t^2(I - \Sigma)\right)(\alpha_t^2\Sigma + \sigma_t^2 I)^{-1}\mu \\
&= -\sigma_t(\alpha_t^2 + \sigma_t^2)(\alpha_t^2\Sigma + \sigma_t^2 I)^{-1}\mu
\end{aligned}
$$

**Flow matching loss**

$$
\mathcal{L} = \mathbb{E}_{\mathbf{x}_0\sim\mathcal{N}(0,I),\ \mathbf{x}_1\sim p_1}\|\mathbf{u}_\theta\big((1-t)\mathbf{x}_0 + t\mathbf{x}_1, t\big) - (\mathbf{x}_1 - \mathbf{x}_0)\|^2
$$

Moments of input-output

$$\mu_x = t\mu, \ \mu_y = \mu$$
$$\Sigma_{xx} = \mathrm{Cov}((1-t)\mathbf{x}_0 + t\mathbf{x}_1, (1-t)\mathbf{x}_0 + t\mathbf{x}_1) = t^2\Sigma + (1-t)^2 I$$
$$\Sigma_{yx} = \mathrm{Cov}(\mathbf{x}_1 - \mathbf{x}_0, (1-t)\mathbf{x}_0 + t\mathbf{x}_1) = t\Sigma - (1-t)I$$

Optimum

$$\begin{aligned}
\mathbf{W}^* &= \Sigma_{xx}^{-1}\Sigma_{yx} \\
&= (t^2\Sigma + (1-t)^2 I)^{-1}(t\Sigma - (1-t)I) \\
\mathbf{b}^* &= \mu_y - \mathbf{W}^*\mu_x \\
&= \mu - \mathbf{W}^* t\mu \\
&= (I - t\mathbf{W}^*)\mu \\
&= \left(I - t(t^2\Sigma + (1-t)^2 I)^{-1}(t\Sigma - (1-t)I)\right)\mu \\
&= \left(t^2\Sigma + (1-t)^2 I - t(t\Sigma - (1-t)I)\right)(t^2\Sigma + (1-t)^2 I)^{-1}\mu \\
&= \left((1-t)^2 I + t(1-t)I\right)(t^2\Sigma + (1-t)^2 I)^{-1}\mu \\
&= (1-t)(t^2\Sigma + (1-t)^2 I)^{-1}\mu
\end{aligned}$$

## C.5 General Analysis of the Sampling ODE

The diffusion sampling process per probability flow ODE is the following

$$\frac{d\mathbf{x}}{d\sigma} = -\sigma\mathbf{s}(\mathbf{x}, \sigma) \tag{26}$$

$$= -\frac{\mathbf{D}(\mathbf{x}, \sigma) - \mathbf{x}}{\sigma}$$

given the relation between score and denoiser (Tweedie's formula).

$$\mathbf{s}(\mathbf{x}, \sigma) = \frac{\mathbf{D}(\mathbf{x}, \sigma) - \mathbf{x}}{\sigma^2}$$

Intuitively, when the denoiser is a linear function of $\mathbf{x}$, then the sampling ODE is also a linear (time-varying) dynamic system with respect to $\mathbf{x}$.

When all $\mathbf{W}_\sigma$ are jointly diagonalizable by a shared set of orthonormal bases $\{\mathbf{u}_k\}$ (e.g., the PC basis of data, or Fourier basis), i.e., commute, we can solve the sampling dynamics mode by mode by projecting onto such basis.

$$\frac{d}{d\sigma}\mathbf{x} = -\frac{\mathbf{W}_\sigma\mathbf{x} + \mathbf{b}_\sigma - \mathbf{x}}{\sigma} \tag{27}$$

$$\frac{d}{d\sigma}\mathbf{u}_k^T\mathbf{x} = -\frac{1}{\sigma}(\mathbf{u}_k^T\mathbf{W}_\sigma\mathbf{x} + \mathbf{u}_k^T\mathbf{b}_\sigma - \mathbf{u}_k^T\mathbf{x}) \tag{28}$$

Representing $\mathbf{W}_\sigma, \mathbf{b}_\sigma, \mathbf{x}(\sigma)$ on the shared eigenbasis,

$$\mathbf{W}_\sigma = \sum_k \psi_k(\sigma)\mathbf{u}_k\mathbf{u}_k^T \tag{29}$$

$$\mathbf{b}_\sigma = \sum_k b_k(\sigma)\mathbf{u}_k \tag{30}$$

$$\mathbf{x}(\sigma) = \sum_k c_k(\sigma)\mathbf{u}_k \tag{31}$$

We can project the sampling ODE onto eigenbasis, i.e.

$$\frac{d}{d\sigma}\mathbf{u}_k^T\mathbf{x} = -\frac{1}{\sigma}\big((\psi_k(\sigma) - 1)\mathbf{u}_k^T\mathbf{x} + b_k(\sigma)\big)$$

$$\frac{d}{d\sigma}c_k(\sigma) = -\frac{1}{\sigma}\big((\psi_k(\sigma) - 1)c_k(\sigma) + b_k(\sigma)\big)$$

$$\frac{d}{d\sigma}c_k(\sigma) + \big(\frac{\psi_k(\sigma) - 1}{\sigma}\big)c_k(\sigma) = -\frac{1}{\sigma}b_k(\sigma)$$

This can be solved via the general solution of a first-order linear ODE: given

$$\frac{dy}{dx} + P(x)y = Q(x)$$

the general solution reads

$$y = e^{-\int^x P(\lambda)\,d\lambda}\left[\int^x e^{\int^\lambda P(\varepsilon)\,d\varepsilon}Q(\lambda)\,d\lambda + C\right]$$

In our case,

$$c_k(\sigma) = e^{-\int^\sigma \frac{\psi_k(\lambda)-1}{\lambda}\,d\lambda}\left[\int^\sigma -\frac{b_k(\lambda)}{\lambda}e^{\int^\lambda \frac{\psi_k(\varepsilon)-1}{\varepsilon}\,d\varepsilon}\,d\lambda + C\right]$$

$$= e^{-\int_{\sigma_T}^\sigma \frac{\psi_k(\lambda)-1}{\lambda}\,d\lambda}\left[c_k(\sigma_T) + \int_{\sigma_T}^\sigma -\frac{b_k(\lambda)}{\lambda}e^{\int^\lambda \frac{\psi_k(\varepsilon)-1}{\varepsilon}\,d\varepsilon}\,d\lambda\right]$$

$$= e^{-\int_{\sigma_T}^\sigma \frac{\psi_k(\lambda)-1}{\lambda}\,d\lambda}c_k(\sigma_T) + \int_{\sigma_T}^\sigma -\frac{b_k(\lambda)}{\lambda}e^{-\int_\lambda^\sigma \frac{\psi_k(\varepsilon)-1}{\varepsilon}\,d\varepsilon}\,d\lambda$$

Rewriting the solution to expose the linear dependency on initial values, the solution reads,

$$c_k(\sigma) = A_k(\sigma; \sigma_T)c_k(\sigma_T) + B_k(\sigma; \sigma_T) \tag{32}$$

$$A_k(\sigma; \sigma_T) = e^{-\int_{\sigma_T}^{\sigma} \frac{\psi_k(\lambda)-1}{\lambda} d\lambda} \tag{33}$$

$$B_k(\sigma; \sigma_T) = \int_{\sigma_T}^{\sigma} -\frac{b_k(\lambda)}{\lambda} e^{-\int_{\lambda}^{\sigma} \frac{\psi_k(\varepsilon)-1}{\varepsilon} d\varepsilon} d\lambda \tag{34}$$

Consider the function

$$\Phi_k(\sigma) := \exp\left(-\int_0^{\sigma} \frac{\psi_k(\lambda)-1}{\lambda} d\lambda\right) \tag{35}$$

Then the integration functions can be expressed as

$$A_k(\sigma; \sigma_T) = \Phi_k(\sigma)/\Phi_k(\sigma_T) \tag{36}$$

$$B_k(\sigma; \sigma_T) = \int_{\sigma_T}^{\sigma} -\frac{b_k(\lambda)}{\lambda} \Phi_k(\sigma)/\Phi_k(\lambda) \, d\lambda \tag{37}$$

By initial noise distribution, $c_k(\sigma_T) \sim \mathcal{N}(0, \sigma_T^2)$, the variance of $c_k(\sigma)$ at any sampling time $\sigma$ can be written down

$$\mathrm{Var}[c_k(\sigma)] = \sigma_T^2 \exp\left(-2\int_{\sigma_T}^{\sigma} \frac{\psi_k(\lambda)-1}{\lambda}\right) \tag{38}$$

$$= \sigma_T^2 \left(\frac{\Phi_k(\sigma)}{\Phi_k(\sigma_T)}\right)^2 \tag{39}$$

Since $\mathbb{E}[c_k(\sigma_T)] = 0$,

$$\mathbb{E}[c_k(\sigma)] = e^{-\int_{\sigma_T}^{\sigma} \frac{\psi_k(\lambda)-1}{\lambda} d\lambda} \left[\int_{\sigma_T}^{\sigma} -\frac{b_k(\lambda)}{\lambda} e^{\int_{\sigma_T}^{\lambda} \frac{\psi_k(\varepsilon)-1}{\varepsilon} d\varepsilon} d\lambda\right] \tag{40}$$

$$= -\int_{\sigma_T}^{\sigma} \frac{b_k(\lambda)}{\lambda} e^{-\int_{\lambda}^{\sigma} \frac{\psi_k(\varepsilon)-1}{\varepsilon} d\varepsilon} d\lambda \tag{41}$$

$$= -\int_{\sigma_T}^{\sigma} \frac{b_k(\lambda)}{\lambda} \frac{\Phi_k(\sigma)}{\Phi_k(\lambda)} d\lambda \tag{42}$$

$$= B_k(\sigma; \sigma_T) \tag{43}$$

Since $\mathbf{x}_\sigma$ is a linear transformation of a Gaussian random variable, the distribution of $\mathbf{x}_\sigma$ at any sampling time $\sigma$ is also Gaussian $\mathbf{x}_\sigma \sim \mathcal{N}(\tilde{\mu}, \tilde{\Sigma})$ with the following mean and covariance.

$$\tilde{\mu} = \sum_k B_k(\sigma; \sigma_T)\mathbf{u}_k \tag{44}$$

$$\tilde{\Sigma} = \sum_k \sigma_T^2 \left(\frac{\Phi_k(\sigma)}{\Phi_k(\sigma_T)}\right)^2 \mathbf{u}_k \tag{45}$$

## C.6    KL Divergence Computation

KL divergence between two multivariate Gaussian distributions is,

$$
\begin{aligned}
&KL(\mathcal{N}(\mu_1, \boldsymbol{\Sigma}_1) \,||\, \mathcal{N}(\mu_2, \boldsymbol{\Sigma}_2)) \\
&= \int \left[ \frac{1}{2} \log \frac{|\boldsymbol{\Sigma}_2|}{|\boldsymbol{\Sigma}_1|} - \frac{1}{2}(x - \mu_1)^T \boldsymbol{\Sigma}_1^{-1}(x - \mu_1) + \frac{1}{2}(x - \mu_2)^T \boldsymbol{\Sigma}_2^{-1}(x - \mu_2) \right] \times p(x) dx \\
&= \frac{1}{2} \log \frac{|\boldsymbol{\Sigma}_2|}{|\boldsymbol{\Sigma}_1|} - \frac{1}{2}\text{tr}\left\{ E[(x - \mu_1)(x - \mu_1)^T]\, \boldsymbol{\Sigma}_1^{-1} \right\} + \frac{1}{2}E[(x - \mu_2)^T \boldsymbol{\Sigma}_2^{-1}(x - \mu_2)] \\
&= \frac{1}{2} \log \frac{|\boldsymbol{\Sigma}_2|}{|\boldsymbol{\Sigma}_1|} - \frac{1}{2}\text{tr}\left\{ \mathbf{I}_d \right\} + \frac{1}{2}(\mu_1 - \mu_2)^T \boldsymbol{\Sigma}_2^{-1}(\mu_1 - \mu_2) + \frac{1}{2}\text{tr}\{\boldsymbol{\Sigma}_2^{-1}\boldsymbol{\Sigma}_1\} \\
&= \frac{1}{2} \left[ \log \frac{|\boldsymbol{\Sigma}_2|}{|\boldsymbol{\Sigma}_1|} - d + \text{tr}\{\boldsymbol{\Sigma}_2^{-1}\boldsymbol{\Sigma}_1\} + (\mu_2 - \mu_1)^T \boldsymbol{\Sigma}_2^{-1}(\mu_2 - \mu_1) \right]. \tag{46}
\end{aligned}
$$

This formula further simplifies when $\boldsymbol{\Sigma}_2$ and $\boldsymbol{\Sigma}_1$ share the same eigenbasis. We can write their eigendecomposition as $\boldsymbol{\Sigma}_2 = U\Lambda_2 U^T$, $\boldsymbol{\Sigma}_1 = U\Lambda_1 U^T$,

$$
\begin{aligned}
KL(\mathcal{N}(\mu_1, \boldsymbol{\Sigma}_1) \,||\, \mathcal{N}(\mu_2, \boldsymbol{\Sigma}_2)) &= \frac{1}{2} \left[ \log \frac{|\Lambda_2|}{|\Lambda_1|} - d + \text{tr}\{\Lambda_2^{-1}\Lambda_1\} + (\mu_2 - \mu_1)^T U\Lambda_2^{-1}U^T(\mu_2 - \mu_1) \right] \\
&= \frac{1}{2} \left[ \log \frac{\prod_k \lambda_{2,k}}{\prod_k \lambda_{1,k}} - d + \sum_k \frac{\lambda_{1,k}}{\lambda_{2,k}} + (\mu_2 - \mu_1)^T U\Lambda_2^{-1}U^T(\mu_2 - \mu_1) \right] \\
&= \frac{1}{2} \left[ \sum_k \log \frac{\lambda_{2,k}}{\lambda_{1,k}} - d + \sum_k \frac{\lambda_{1,k}}{\lambda_{2,k}} + (\mu_2 - \mu_1)^T U\Lambda_2^{-1}U^T(\mu_2 - \mu_1) \right]
\end{aligned}
$$

In more explicit form

$$
\begin{aligned}
KL(\mathcal{N}(\mu_1, \boldsymbol{\Sigma}_1) \,||\, \mathcal{N}(\mu_2, \boldsymbol{\Sigma}_2)) &= \frac{1}{2} \left[ \sum_k \log \frac{\lambda_{2,k}}{\lambda_{1,k}} - d + \sum_k \frac{\lambda_{1,k}}{\lambda_{2,k}} + (\mu_2 - \mu_1)^T \sum_k \frac{\mathbf{u}_k \mathbf{u}_k^T}{\lambda_{2,k}}(\mu_2 - \mu_1) \right] \\
&= \frac{1}{2} \sum_k \left[ \log \frac{\lambda_{2,k}}{\lambda_{1,k}} + \frac{\lambda_{1,k}}{\lambda_{2,k}} - 1 + \frac{\left(\mathbf{u}_k^T(\mu_2 - \mu_1)\right)^2}{\lambda_{2,k}} \right] \tag{47}
\end{aligned}
$$

If they share the same mean $\mu_1 = \mu_2$ it simplifies even further

$$
KL = \frac{1}{2} \left[ \sum_k \frac{\lambda_{1,k}}{\lambda_{2,k}} - \sum_k \log \frac{\lambda_{1,k}}{\lambda_{2,k}} - d \right] \tag{48}
$$

which has unique minimizer when $\frac{\lambda_{1,k}}{\lambda_{2,k}} = 1$.

Thus, we can compute the contribution to KL mode by mode. We denote the contribution from mode $k$ as

$$
KL_k = \frac{1}{2}\left(\frac{\lambda_{1,k}}{\lambda_{2,k}} - \log \frac{\lambda_{1,k}}{\lambda_{2,k}} - 1\right) \tag{49}
$$

Thus for Gaussian data that share the same mean, the KL divergence can be reduced to the ratio of generated and true variance along each principal axis of data.

# D  Detailed Derivations for the One-Layer Linear Model

## D.1  Zero-mean data: Exponential converging training dynamics

If $\mu = 0$, then the problem reduces to **learning the covariance of data.** The gradient to weights and bias decouples [1]

$$\nabla_{\mathbf{b}}\mathcal{L} = 2\mathbf{b} \tag{50}$$

$$\nabla_{\mathbf{W}}\mathcal{L} = -2\boldsymbol{\Sigma} + 2\mathbf{W}(\sigma^2\mathbf{I} + \boldsymbol{\Sigma}) \tag{51}$$

The solution of $\mathbf{b}$ is an exponential decay

$$\frac{d}{d\tau}\mathbf{b} = -2\eta\mathbf{b} \tag{52}$$

$$\mathbf{b}(\tau) = \mathbf{b}^0 \exp(-2\eta\tau) \tag{53}$$

The solution of $\mathbf{W}$ can be obtained by projecting onto eigenbasis of $\boldsymbol{\Sigma}$

$$\nabla_{\mathbf{W}}\mathcal{L} \cdot \mathbf{u}_k = -2\boldsymbol{\Sigma}\mathbf{u}_k + 2\mathbf{W}(\sigma^2\mathbf{I} + \boldsymbol{\Sigma})\mathbf{u}_k \tag{54}$$

$$= -2\lambda_k\mathbf{u}_k + 2(\sigma^2 + \lambda_k)\mathbf{W}\mathbf{u}_k \tag{55}$$

Thus,

$$\frac{d}{d\tau}(\mathbf{W}\mathbf{u}_k) = -\eta[-2\lambda_k\mathbf{u}_k + 2(\sigma^2 + \lambda_k)(\mathbf{W}\mathbf{u}_k)]$$

Define variable $\mathbf{v}_k = \mathbf{W}\mathbf{u}_k$

$$\frac{d}{d\tau}\mathbf{v}_k = 2\eta\lambda_k\mathbf{u}_k - 2\eta(\sigma^2 + \lambda_k)\mathbf{v}_k$$

$$= 2\eta(\sigma^2 + \lambda_k)\left(\frac{\lambda_k}{(\sigma^2 + \lambda_k)}\mathbf{u}_k - \mathbf{v}_k\right)$$

$$\mathbf{v}_k(\tau) = \frac{\lambda_k}{(\sigma^2 + \lambda_k)}\mathbf{u}_k + \left(\mathbf{v}_k^0 - \frac{\lambda_k}{(\sigma^2 + \lambda_k)}\mathbf{u}_k\right)\exp\left(-2\eta(\sigma^2 + \lambda_k)\tau\right)$$

Since

$$[\mathbf{v}_k..] = \mathbf{W}[\mathbf{u}_k...]$$

$$VU^T = \mathbf{W}$$

$$\mathbf{W} = \sum_k \mathbf{v}_k\mathbf{u}_k^T$$

The full solution of $\mathbf{W}$ reads

$$\mathbf{W}(\tau) = \sum_k \mathbf{v}_k(\tau)\mathbf{u}_k^T$$

$$= \sum_k \frac{\lambda_k}{(\sigma^2 + \lambda_k)}\mathbf{u}_k\mathbf{u}_k^T + \sum_k \left(\mathbf{v}_k^0 - \frac{\lambda_k}{(\sigma^2 + \lambda_k)}\mathbf{u}_k\right)\mathbf{u}_k^T e^{-2\eta(\sigma^2+\lambda_k)\tau}$$

$$= \sum_k \frac{\lambda_k}{(\sigma^2 + \lambda_k)}\mathbf{u}_k\mathbf{u}_k^T(1 - e^{-2\eta(\sigma^2+\lambda_k)\tau}) + \sum_k \mathbf{v}_k^0\mathbf{u}_k^T e^{-2\eta(\sigma^2+\lambda_k)\tau}$$

$$= \sum_k \frac{\lambda_k}{(\sigma^2 + \lambda_k)}\mathbf{u}_k\mathbf{u}_k^T(1 - e^{-2\eta(\sigma^2+\lambda_k)\tau}) + \mathbf{W}(0)\sum_k \mathbf{u}_k\mathbf{u}_k^T e^{-2\eta(\sigma^2+\lambda_k)\tau}$$

$$= \mathbf{W}^* + \sum_k \left(\mathbf{v}_k^0 - \frac{\lambda_k}{(\sigma^2 + \lambda_k)}\mathbf{u}_k\right)\mathbf{u}_k^T e^{-2\eta(\sigma^2+\lambda_k)\tau}$$

$$= \mathbf{W}^* + \sum_k \left(\mathbf{W}(0)\mathbf{u}_k - \frac{\lambda_k}{(\sigma^2 + \lambda_k)}\mathbf{u}_k\right)\mathbf{u}_k^T e^{-2\eta(\sigma^2+\lambda_k)\tau} \tag{56}$$

$$= \mathbf{W}^* + \sum_k \left(\mathbf{W}(0) - \mathbf{W}^*\right)\mathbf{u}_k\mathbf{u}_k^T e^{-2\eta(\sigma^2+\lambda_k)\tau} \tag{57}$$

where $\mathbf{v}_k^0 := \mathbf{W}(0)\mathbf{u}_k$.

---

[1]For notational clarity, we derive the gradient flow at a fixed noise scale $\sigma$, and omit the subscript and/or argument $\sigma$ from $\mathbf{W}$, $\mathbf{b}$, and $\mathcal{L}$.

**Remarks**

- The weight matrix $\mathbf{W}$ converges to the final target mode by mode.
- The deviation along each eigen dimension decays at different rates depending on the eigenvalue.
- The deviation on eigen mode $\mathbf{u}_k$ has the time constant $(2\eta(\sigma^2 + \lambda_k))^{-1}$ i.e. the larger eigen dimensions will be learned faster.
- While the "non-resolved" dimensions will learn at the same speed $\sim (2\eta\sigma^2)^{-1}$
- Comparing across noise scale $\sigma$, the larger noise scales will be learned faster.

**Score estimation error dynamics**     Consider a target quantity of interest, i.e. difference of the score approximator from the true score.

First, under the $\mu = 0$ assumption, we have

$$E_s = \mathbb{E}_{\mathbf{x}} \|s(\mathbf{x}) - s^*(\mathbf{x})\|^2 = \frac{1}{\sigma^4}\left[ \|b - b^*\|^2 + \mathrm{Tr}[(\mathbf{W} - \mathbf{W}^*)^T(\mathbf{W} - \mathbf{W}^*)(\mathbf{\Sigma} + \sigma^2\mathbf{I})] \right] \quad (58)$$

The deviations can be expressed as

$$b - b^* = b^0 \exp(-2\eta\tau)$$

$$\mathbf{W} - \mathbf{W}^* = \sum_k \left( \mathbf{v}_k^0 - \frac{\lambda_k}{(\sigma^2 + \lambda_k)}\mathbf{u}_k \right)\mathbf{u}_k^T e^{-2\eta(\sigma^2 + \lambda_k)\tau}$$

with the initial projection $\mathbf{v}_k^0 := \mathbf{W}(0)\mathbf{u}_k$

$$E_s = \frac{1}{\sigma^4}\left[ \|b^0\|^2 \exp(-4\eta\tau) + \sum_k (\sigma^2 + \lambda_k)\left\|\mathbf{v}_k^0 - \frac{\lambda_k}{(\sigma^2 + \lambda_k)}\mathbf{u}_k\right\|^2 e^{-4\eta(\sigma^2 + \lambda_k)\tau} \right] \quad (59)$$

$$= \frac{1}{\sigma^4}\left[ \|\delta_b\|^2 \exp(-4\eta\tau) + \sum_k (\sigma^2 + \lambda_k)\|\delta_{\mathbf{k}}\|^2 e^{-4\eta(\sigma^2 + \lambda_k)\tau} \right] \quad (60)$$

$$\delta_{\mathbf{k}} := \mathbf{W}(0)\mathbf{u}_k - \frac{\lambda_k}{(\sigma^2 + \lambda_k)}\mathbf{u}_k$$

$$\delta_b := b^0$$

This provides us with the exact formula for error decay during training.

**Denoiser estimation error dynamics**

$$\begin{aligned}
E_D &= \mathbb{E}_{\mathbf{x}}\|D(\mathbf{x}) - D^*(\mathbf{x})\|^2 \\
&= \sigma^4 E_s \\
&= (\delta_b)^2 \exp(-4\eta\tau) + \sum_k (\sigma^2 + \lambda_k)\|\delta_{\mathbf{k}}\|^2 e^{-4\eta(\sigma^2 + \lambda_k)\tau} \quad (61)
\end{aligned}$$

**Training loss dynamics**     Under $\mu = 0$ assumption, the training loss is basically the true denoiser estimation error plus a constant term $\sigma^2\mathrm{Tr}[\mathbf{\Sigma}(\sigma^2\mathbf{I} + \mathbf{\Sigma})^{-1}]$ determined by data covariance (trace of resolvent).

$$\begin{aligned}
\mathcal{L}_{\mu=0} &= \|b - (\mathbf{I} - \mathbf{W})\mu\|_2^2 + \mathrm{Tr}[(\mathbf{W} - \mathbf{W}^*)(\sigma^2\mathbf{I} + \mathbf{\Sigma})(\mathbf{W} - \mathbf{W}^*)^T] + \sigma^2\mathrm{Tr}[\mathbf{\Sigma}(\sigma^2\mathbf{I} + \mathbf{\Sigma})^{-1}] \\
(\mu = 0) &= \|b\|_2^2 + \mathrm{Tr}[(\mathbf{W} - \mathbf{W}^*)^T(\mathbf{W} - \mathbf{W}^*)(\sigma^2\mathbf{I} + \mathbf{\Sigma})] + \sigma^2\mathrm{Tr}[\mathbf{\Sigma}(\sigma^2\mathbf{I} + \mathbf{\Sigma})^{-1}] \quad (62) \\
&= E_D + \sigma^2\mathrm{Tr}[\mathbf{\Sigma}(\sigma^2\mathbf{I} + \mathbf{\Sigma})^{-1}] \quad (63)
\end{aligned}$$

### D.1.1   Discrete time Gradient descent dynamics

When the dynamics is discrete-time gradient descent instead of gradient flow, we have,

$$\nabla_{\mathbf{b}}\mathcal{L} = 2\mathbf{b} \quad (64)$$

$$\nabla_{\mathbf{W}}\mathcal{L} = -2\mathbf{\Sigma} + 2\mathbf{W}(\sigma^2\mathbf{I} + \mathbf{\Sigma}) \quad (65)$$

The GD update equation reads, with $\eta$ the learning rate

$$\mathbf{b}_{t+1} - \mathbf{b}_t = -\eta \nabla_{\mathbf{b}_t} \mathcal{L} \tag{66}$$

$$\mathbf{W}_{t+1} - \mathbf{W}_t = -\eta \nabla_{\mathbf{W}_t} \mathcal{L} \tag{67}$$

$$\mathbf{b}_{t+1} = (1 - 2\eta)\mathbf{b}_t \tag{68}$$

$$\mathbf{W}_{t+1} = \mathbf{W}_t - 2\eta(-\mathbf{\Sigma} + \mathbf{W}_t(\sigma^2\mathbf{I} + \mathbf{\Sigma})) \tag{69}$$

$$= 2\eta\mathbf{\Sigma} + \mathbf{W}_t(\mathbf{I} - 2\eta\sigma^2\mathbf{I} - 2\eta\mathbf{\Sigma}) \tag{70}$$

$$= 2\eta\mathbf{\Sigma} + \mathbf{W}_t\big((1 - 2\eta\sigma^2)\mathbf{I} - 2\eta\mathbf{\Sigma}\big) \tag{71}$$

For the weight dynamics, we have

$$\mathbf{W}_{t+1}\mathbf{u}_k = 2\eta\mathbf{\Sigma}\mathbf{u}_k + \mathbf{W}_t\big((1 - 2\eta\sigma^2)\mathbf{I} - 2\eta\mathbf{\Sigma}\big)\mathbf{u}_k \tag{72}$$

$$= 2\eta\lambda_k\mathbf{u}_k + \mathbf{W}_t\mathbf{u}_k\big(1 - 2\eta\sigma^2 - 2\eta\lambda_k\big) \tag{73}$$

This iteration is exponentially converging to the fixed point $\frac{\lambda_k}{\sigma^2 + \lambda_k}$.

$$\mathbf{W}_{t+1}\mathbf{u}_k - \frac{\lambda_k}{\sigma^2 + \lambda_k}\mathbf{u}_k = (2\eta\lambda_k - \frac{\lambda_k}{\sigma^2 + \lambda_k})\mathbf{u}_k + \mathbf{W}_t\mathbf{u}_k\big(1 - 2\eta\sigma^2 - 2\eta\lambda_k\big) \tag{74}$$

$$= (2\eta(\sigma^2 + \lambda_k) - 1)\frac{\lambda_k}{\sigma^2 + \lambda_k}\mathbf{u}_k + \mathbf{W}_t\mathbf{u}_k\big(1 - 2\eta\sigma^2 - 2\eta\lambda_k\big) \tag{75}$$

$$= (\mathbf{W}_t\mathbf{u}_k - \frac{\lambda_k}{\sigma^2 + \lambda_k}\mathbf{u}_k)\big(1 - 2\eta\sigma^2 - 2\eta\lambda_k\big) \tag{76}$$

Thus

$$\mathbf{W}_t\mathbf{u}_k = \frac{\lambda_k}{\sigma^2 + \lambda_k}\mathbf{u}_k + (\mathbf{W}_0\mathbf{u}_k - \frac{\lambda_k}{\sigma^2 + \lambda_k}\mathbf{u}_k)\big(1 - 2\eta\sigma^2 - 2\eta\lambda_k\big)^t \tag{77}$$

$$\mathbf{b}_t = \mathbf{b}_0(1 - 2\eta)^t \tag{78}$$

$$\mathbf{W}_t = \sum_k \frac{\lambda_k}{\sigma^2 + \lambda_k}\mathbf{u}_k\mathbf{u}_k^T + (\mathbf{W}_0 - \frac{\lambda_k}{\sigma^2 + \lambda_k}\mathbf{u}_k)\mathbf{u}_k^T\big(1 - 2\eta\sigma^2 - 2\eta\lambda_k\big)^t \tag{79}$$

So, there is no significant change from the continuous-time version.

### D.1.2 Special parametrization: residual connection

Consider a special parametrization of weights

$$\mathbf{W} = c_{skip}\mathbf{I} + c_{out}\mathbf{W}' \tag{80}$$

It's easy to derive the dynamics of the new variables via chain rule,

$$\frac{\partial \mathbf{W}}{\partial \mathbf{W}'} = c_{out}, \tag{81}$$

$$\frac{\partial \mathcal{L}}{\partial \mathbf{W}'} = c_{out}\frac{\partial \mathcal{L}}{\partial \mathbf{W}}. \tag{82}$$

With the original gradient

$$\nabla_{\mathbf{b}}\mathcal{L} = 2(\mathbf{b} - (\mathbf{I} - \mathbf{W})\mu) \tag{83}$$

$$\nabla_{\mathbf{W}}\mathcal{L} = -2\mathbf{\Sigma} + 2\mathbf{W}(\sigma^2\mathbf{I} + \mathbf{\Sigma}) + [2\mathbf{W}\mu\mu^T + 2(\mathbf{b} - \mu)\mu^T] \tag{84}$$

and zero mean case

$$\nabla_{\mathbf{b}}\mathcal{L} = 2\mathbf{b} \tag{85}$$

$$\nabla_{\mathbf{W}}\mathcal{L} = -2\mathbf{\Sigma} + 2\mathbf{W}(\sigma^2\mathbf{I} + \mathbf{\Sigma}) \tag{86}$$

the gradient to new parameters

$$\nabla_{\mathbf{W}'}\mathcal{L} = c_{out}\nabla_{\mathbf{W}}\mathcal{L} = 2c_{out}(-\mathbf{\Sigma} + (c_{skip}\mathbf{I} + c_{out}\mathbf{W}')(\sigma^2\mathbf{I} + \mathbf{\Sigma})) \tag{87}$$

$$= 2c_{out}(-\mathbf{\Sigma} + c_{skip}(\sigma^2\mathbf{I} + \mathbf{\Sigma}) + c_{out}\mathbf{W}'(\sigma^2\mathbf{I} + \mathbf{\Sigma})) \tag{88}$$

$$= 2c_{out}(c_{skip}(\sigma^2\mathbf{I} + \mathbf{\Sigma}) - \mathbf{\Sigma} + c_{out}\mathbf{W}'(\sigma^2\mathbf{I} + \mathbf{\Sigma})) \tag{89}$$

$$\frac{d\mathbf{W}'}{d\tau} = -\eta\nabla_{\mathbf{W}'}\mathcal{L} \tag{90}$$

$$\frac{1}{2\eta c_{out}}\frac{d\mathbf{W}'}{d\tau} = -(c_{skip}(\sigma^2\mathbf{I} + \mathbf{\Sigma}) - \mathbf{\Sigma} + c_{out}\mathbf{W}'(\sigma^2\mathbf{I} + \mathbf{\Sigma})) \tag{91}$$

$$\mathbf{W}'^* = \frac{1}{c_{out}}(\mathbf{\Sigma} - c_{skip}(\sigma^2\mathbf{I} + \mathbf{\Sigma}))(\sigma^2\mathbf{I} + \mathbf{\Sigma})^{-1} \tag{92}$$

$$= \frac{1}{c_{out}}(\mathbf{\Sigma}(\sigma^2\mathbf{I} + \mathbf{\Sigma})^{-1} - c_{skip}\mathbf{I}) \tag{93}$$

$$\mathbf{W}'(\tau)\mathbf{u}_k = \mathbf{W}'(0)\mathbf{u}_k\exp(-2\eta\tau c_{out}^2(\sigma^2 + \lambda_k)) + \tag{94}$$
$$\mathbf{u}_k\frac{\lambda_k - c_{skip}(\sigma^2 + \lambda_k)}{c_{out}(\sigma^2 + \lambda_k)}(1 - \exp(-2\eta\tau c_{out}^2(\sigma^2 + \lambda_k)))$$

The solution to the new weights reads

$$\mathbf{W}'(\tau) = (\mathbf{W}'(0) - \mathbf{W}'^*)\sum_k\mathbf{u}_k\mathbf{u}_k^T\exp(-2\eta\tau c_{out}^2(\sigma^2 + \lambda_k)) + \mathbf{W}'^* \tag{95}$$

$$\mathbf{W}(\tau) = c_{skip}\mathbf{I} + c_{out}\mathbf{W}'(\tau) \tag{96}$$

$$= c_{skip}\mathbf{I} + c_{out}(\mathbf{W}'(0) - \mathbf{W}'^*)\sum_k\mathbf{u}_k\mathbf{u}_k^T\exp(-2\eta\tau c_{out}^2(\sigma^2 + \lambda_k)) + c_{out}\mathbf{W}'^* \tag{97}$$

$$= \mathbf{\Sigma}(\sigma^2\mathbf{I} + \mathbf{\Sigma})^{-1} + c_{out}(\mathbf{W}'(0) - \mathbf{W}'^*)\sum_k\mathbf{u}_k\mathbf{u}_k^T\exp(-2\eta\tau c_{out}^2(\sigma^2 + \lambda_k)) \tag{98}$$

$$= \mathbf{W}^* + c_{out}(\mathbf{W}'(0) - \mathbf{W}'^*)\sum_k\mathbf{u}_k\mathbf{u}_k^T\exp(-2\eta\tau c_{out}^2(\sigma^2 + \lambda_k)) \tag{99}$$

$$= \mathbf{W}^* + (\mathbf{W}(0) - \mathbf{W}^*)\sum_k\mathbf{u}_k\mathbf{u}_k^T\exp(-2\eta\tau c_{out}^2(\sigma^2 + \lambda_k)) \tag{100}$$

The only difference is scaling the learning rate by a factor of $c_{out}^2$. Also potentially depending on whether we choose to initialize $\mathbf{W}(0)$ or $\mathbf{W}'(0)$ from a fixed distribution, we would get different initial values for the dynamics.

## D.2 General non-centered distribution: Interaction of mean and covariance learning

**Summary of results for non-centered case** When $\mu \neq 0$, the gradients to $\mathbf{W}$ and $\mathbf{b}$ become entangled (see Eq. 5), resulting in a coupled linear dynamic system as follows.

**Proposition D.1** (Learning dynamics of linear denoiser, non centered case)**.** *Gradient flow (Eq. GF) is equivalent to the following ODE, with redefined dynamic variables,* $\mathbf{v}_k(\tau) = \mathbf{W}(\tau)\mathbf{u}_k$, $\bar{\mathbf{b}}(\tau) = \mathbf{b}(\tau) - \mu$*. Denote overlap* $m_k := \mathbf{u}_k^{\mathsf{T}}\mu$,

$$\frac{1}{2\eta}\frac{d}{d\tau}\begin{bmatrix}\mathbf{v}_1\\\mathbf{v}_2\\\cdot\\\bar{\mathbf{b}}\end{bmatrix} = -M\begin{bmatrix}\mathbf{v}_1\\\mathbf{v}_2\\\cdot\\\bar{\mathbf{b}}\end{bmatrix} + \begin{bmatrix}\lambda_1\mathbf{u}_1\\\lambda_2\mathbf{u}_2\\\cdot\\0\end{bmatrix} \tag{101}$$

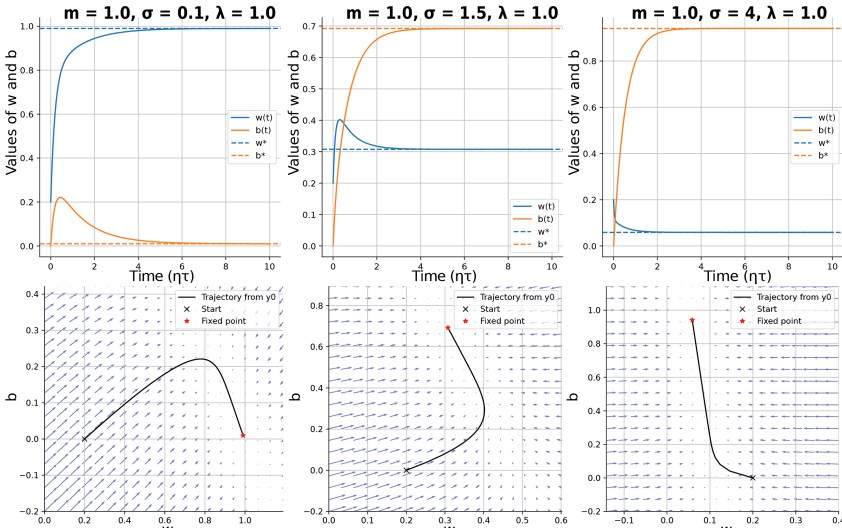

Figure 27: **Interaction of mean and covariance learning. Top** solution to the $w, b$ dynamics under different noise level $\sigma \in \{0.1, 1.5, 4\}$. **Bottom** Phase portraits corresponding to the two-d system. ($m = 1, \lambda_k = 1$)

*with a fixed dynamic matrix $M$ defined by $\otimes$ Kronecker product.*

$$M := \begin{bmatrix} \sigma^2 + \lambda_1 + m_1^2 & m_1 m_2 & . & m_1 \\ m_1 m_2 & \sigma^2 + \lambda_2 + m_2^2 & . & m_2 \\ ... & ... & ... & . \\ m_1 & m_2 & . & 1 \end{bmatrix} \otimes \mathbf{I}_d$$

$$:= \tilde{M} \otimes \mathbf{I}_d \tag{102}$$

*Remark* D.2. The dynamics matrix $\tilde{M}$ has a *rank-one plus diagonal structure*, specifically, it is the diagonal dynamics matrix in Eq. 7, perturbed by the outer product of the overlap vector $m_k$. The eigenvalues of such matrix can be efficiently solved by numerical algebra, with eigenvectors expressed by Bunch–Nielsen–Sorensen formula [56, 57]. Without a closed-form formula, we have to resort to numerical simulation and low-dimensional examples to gain further insights. We can see the coupling of $\mathbf{W}$ and $\mathbf{b}$ dynamics comes from the overlap of mean and principal component $m_k = \mathbf{u}_k^\mathsf{T} \mu$. When certain eigenmode has no overlap, $m_k = 0$, the corresponding weight projection $\mathbf{v}_k(\tau)$ will follow the same dynamics as the zero-mean case, i.e. exponentially converge to the optimal solution $\lambda_k/(\lambda_k + \sigma^2)\mathbf{u}_k$. When the overlap is non-zero $m_k \neq 0$, it will induce interaction between $\mathbf{v}_k(\tau)$ and $\mathbf{b}(\tau)$ and non-monotonic dynamics.

**Two dimensional example**   Here, we show a low-dimensional example illustrating the interaction between the bias and one eigenmode in the weight. Consider the case where distribution mean $\mu$ lies on the direction of a PC $\mathbf{u}_k$. Then only the $\mathbf{u}_k$ mode of weights interacts with the distribution mean, resulting in a two-dimensional linear system, parametrized by noise scale $\sigma$, variance of mode $\lambda$ and amount of alignment $m$. Let the dynamic variable be scalars $w, b$, $\mathbf{v}_k = w(\tau)\mathbf{u}_k, \mathbf{b} = b(\tau)\mathbf{u}_k$.

$$\frac{1}{2\eta}\frac{d}{d\tau}\begin{bmatrix} w \\ b \end{bmatrix} = -\begin{bmatrix} m^2 + \sigma^2 + \lambda & m \\ m & 1 \end{bmatrix}\begin{bmatrix} w \\ b \end{bmatrix} + \begin{bmatrix} \lambda + m^2 \\ m \end{bmatrix} \tag{103}$$

The phase diagram and dynamics depending on the noise scale are shown (Fig. 27): At larger $\sigma$ values, the dynamics of weights $w$ will be much faster than $b$, basically, $w$ gets dynamically captured by $b$, while $b$ slowly relaxes to the optimal value. At small $\sigma$ values, the dynamics timescale of $w$ and $b$ will be closer to each other, and $b$ will usually have non-monotonic transient dynamics. When $\sigma^2$ and $\lambda$ are comparable, $w$ will have non-monotonic dynamics.

### D.2.1 Derivation of non-centered case

In this full case, the dynamics of $\mathbf{b}, \mathbf{W}$ are coupled

$$\nabla_{\mathbf{b}}\mathcal{L} = 2(\mathbf{b} - \mu + \mathbf{W}\mu) \tag{104}$$

$$\nabla_{\mathbf{W}}\mathcal{L} = -2\Sigma + 2\mathbf{W}(\sigma^2\mathbf{I} + \Sigma) + 2(\mathbf{b} - \mu + \mathbf{W}\mu)\mu^T \tag{105}$$

$$= -2\Sigma + 2\mathbf{W}(\sigma^2\mathbf{I} + \Sigma) + \nabla_{\mathbf{b}}\mathcal{L} \cdot \mu^T \tag{106}$$

$$= \nabla_{\mathbf{W}}\bar{\mathcal{L}} + \nabla_{\mathbf{b}}\mathcal{L} \cdot \mu^T \tag{107}$$

$$\nabla_{\mathbf{W}}\bar{\mathcal{L}} := -2\Sigma + 2\mathbf{W}(\sigma^2\mathbf{I} + \Sigma) \tag{108}$$

The nonlinear gradient learning dynamics reads

$$\dot{\mathbf{b}} = -\eta\nabla_{\mathbf{b}}\mathcal{L} \tag{109}$$

$$\dot{\mathbf{W}} = -\eta(\nabla_{\mathbf{W}}\bar{\mathcal{L}} + \nabla_{\mathbf{b}}\mathcal{L} \cdot \mu^T) \tag{110}$$

$$\dot{\mathbf{W}} - \dot{\mathbf{b}} \cdot \mu^T = -\eta\nabla_{\mathbf{W}}\bar{\mathcal{L}} \tag{111}$$

$$= 2\eta\big[\Sigma - \mathbf{W}(\sigma^2\mathbf{I} + \Sigma)\big] \tag{112}$$

$$\dot{\mathbf{b}} = -\eta\nabla_{\mathbf{b}}\mathcal{L} \tag{113}$$

$$= -2\eta(\mathbf{b} - \mu + \mathbf{W}\mu) \tag{114}$$

Consider the projection

$$(\dot{\mathbf{W}} - \dot{\mathbf{b}} \cdot \mu^T)\mathbf{u}_k = 2\eta\big[\Sigma - \mathbf{W}(\sigma^2\mathbf{I} + \Sigma)\big]\mathbf{u}_k \tag{115}$$

$$\dot{\mathbf{W}}\mathbf{u}_k - (\mu^T\mathbf{u}_k)\dot{\mathbf{b}} = 2\eta[\lambda_k\mathbf{u}_k - (\sigma^2 + \lambda_k)\mathbf{W}\mathbf{u}_k] \tag{116}$$

$$\dot{\mathbf{b}} = -2\eta(\mathbf{b} - \mu + \sum_k \mathbf{W}\mathbf{u}_k\mathbf{u}_k^T\mu)$$

Consider the variables $\mathbf{v}_k(\tau) = \mathbf{W}(\tau)\mathbf{u}_k$, let $\bar{\mathbf{b}}(\tau) = \mathbf{b}(\tau) - \mu$ so now the dynamic variables are $\{\mathbf{v}_k, ..., \bar{\mathbf{b}}\}$

$$\dot{\mathbf{v}}_k - (\mu^T\mathbf{u}_k)\dot{\bar{\mathbf{b}}} = 2\eta[\lambda_k\mathbf{u}_k - (\sigma^2 + \lambda_k)\mathbf{v}_k] \tag{117}$$

$$\dot{\bar{\mathbf{b}}} = -2\eta(\bar{\mathbf{b}} + \sum_k (\mu^T\mathbf{u}_k)\mathbf{v}_k) \tag{118}$$

$$\dot{\mathbf{v}}_k = 2\eta[\lambda_k\mathbf{u}_k - (\sigma^2 + \lambda_k)\mathbf{v}_k] - 2\eta(\mu^T\mathbf{u}_k)[\bar{\mathbf{b}} + \sum_l (\mu^T\mathbf{u}_l)\mathbf{v}_l] \tag{119}$$

$$= 2\eta[\lambda_k\mathbf{u}_k - (\sigma^2 + \lambda_k)\mathbf{v}_k - (\mu^T\mathbf{u}_k)\bar{\mathbf{b}} - \sum_l (\mu^T\mathbf{u}_k)(\mu^T\mathbf{u}_l)\mathbf{v}_l] \tag{120}$$

$$\dot{\bar{\mathbf{b}}} = -2\eta(\bar{\mathbf{b}} + \sum_k (\mu^T\mathbf{u}_k)\mathbf{v}_k) \tag{121}$$

The whole dynamics is linear and solvable, but now the dynamics in each component $\mathbf{v}_k$ becomes entangled with other components $\mathbf{v}_l$.

$$\frac{d}{d\tau}\begin{bmatrix} \mathbf{v}_1 \\ \mathbf{v}_2 \\ \vdots \\ \bar{\mathbf{b}} \end{bmatrix} = -2\eta M \begin{bmatrix} \mathbf{v}_1 \\ \mathbf{v}_2 \\ \vdots \\ \bar{\mathbf{b}} \end{bmatrix} + 2\eta \begin{bmatrix} \lambda_1\mathbf{u}_1 \\ \lambda_2\mathbf{u}_2 \\ \vdots \\ 0 \end{bmatrix} \tag{122}$$

with the blocks in $M$ matrix defined as follows

$$M_{kk} = (\sigma^2 + \lambda_k + (\mu^T\mathbf{u}_k)^2)\mathbf{I}$$

$$M_{kl} = (\mu^T\mathbf{u}_k)(\mu^T\mathbf{u}_l)\mathbf{I}$$

$$M_{kb} = (\mu^T\mathbf{u}_k)\mathbf{I}$$

$$M_{bk} = (\mu^T\mathbf{u}_k)\mathbf{I}$$

$$M_{bb} = \mathbf{I}$$

We denote the overlap between the mean and principal components as $m_k = \mu^T \mathbf{u}_k$,

Then we can represent the dynamic matrix $M$ as a tensor product of a dense symmetric matrix $Q$ with identity $\mathbf{I}_N$. More specifically, the dense matrix $Q$ is a diagonal matrix plus the outer product of a vector. So it's real symmetric and diagonalizable.

$$M = \begin{bmatrix} \sigma^2 + \lambda_1 + m_1^2 & m_1 m_2 & . & m_1 \\ m_1 m_2 & \sigma^2 + \lambda_2 + m_2^2 & . & m_2 \\ ... & ... & ... & . \\ m_1 & m_2 & . & 1 \end{bmatrix} \otimes \mathbf{I}_N \tag{123}$$

$$= Q \otimes \mathbf{I}_N \tag{124}$$

$$Q := \begin{bmatrix} \sigma^2 + \lambda_1 + m_1^2 & m_1 m_2 & . & m_1 \\ m_1 m_2 & \sigma^2 + \lambda_2 + m_2^2 & . & m_2 \\ ... & ... & ... & . \\ m_1 & m_2 & . & 1 \end{bmatrix} \tag{125}$$

$$= D + qq^T \tag{126}$$

$$q := [m_1, m_2, ...1]^T \tag{127}$$

$$D := \mathrm{diag}(\sigma^2 + \lambda_1, \sigma^2 + \lambda_2, \sigma^2 + \lambda_3, ...\sigma^2 + \lambda_N, 0) \tag{128}$$

Note the inverse of $Q$ is analytical, but the general eigendecomposition of it is not. Since the dynamic matrix $M$ is real symmetric, the dynamics will still be separable along each eigenmode of $M$ and converge w.r.t. its own eigenvalue. The eigen decomposition of $M$ can be obtained by numerical analysis and eigenvectors from Bunch–Nielsen–Sorensen formula [56, 58]. Generally speaking, since the mean of dataset $\mu$ usually lies in the directions of higher eigenvalues, the dynamics of $\mathbf{b}$ and $\mathbf{v}_k$ in the top eigenspace will be entangled with each other.

As a take home message, the overlap of $\mu$ and spectrum of Gaussian will induce some complex dynamics of bias and weight matrix along these modes. The full dynamics of $\mathbf{W}, \mathbf{b}$ is still linear and solvable, but since the dynamic matrix is a tensor product of a Diagonal + low rank with identity, a closed form solution is generally harder, we can still obtain numerical solution of the dynamics easily.

### D.2.2 Derivation of low-dimensional interaction of mean and variance learning

To gain intuition into how the mean and variance learning happens, consider the 1d distribution case, which shares the same math as the multi dimensional case where the mean overlaps with only one eigenmode

$$\begin{aligned} \nabla_b \mathcal{L} &= 2(b - \mu + \mathbf{W}\mu) \\ &= 2b - 2(1 - w)\mu \\ &= 2\mu w + 2b - 2\mu \end{aligned} \tag{129}$$

$$\begin{aligned} \nabla_{\mathbf{W}} \mathcal{L} &= -2\mathbf{\Sigma} + 2\mathbf{W}(\sigma^2 \mathbf{I} + \mathbf{\Sigma}) + 2(b - \mu + \mathbf{W}\mu)\mu^T \\ &= -2\lambda - 2(1 - w)\mu^2 + 2w(\sigma^2 + \lambda) + 2\mu b \\ &= 2(\mu^2 + \sigma^2 + \lambda)w + 2\mu b - 2(\lambda + \mu^2) \end{aligned} \tag{130}$$

Write down the dynamic equation as matrix equation

$$\frac{d}{d\tau} \begin{bmatrix} w \\ b \end{bmatrix} = -2\eta \left( \begin{bmatrix} \mu^2 + \sigma^2 + \lambda & \mu \\ \mu & 1 \end{bmatrix} \begin{bmatrix} w \\ b \end{bmatrix} - \begin{bmatrix} \lambda + \mu^2 \\ \mu \end{bmatrix} \right)$$

Eigen equation reads

$$\det(A - \gamma \mathbf{I}) = (1 - \gamma)(\mu^2 + \sigma^2 + \lambda - \gamma) - \mu^2 \tag{131}$$

$$= \gamma^2 - \left(\lambda + \mu^2 + \sigma^2 + 1\right)\gamma + \lambda + \sigma^2 \tag{132}$$

Generally for 2x2 matrices,

$$\begin{bmatrix} a & b \\ b & c \end{bmatrix}$$

their eigenvalues are

$$\lambda_{1,2} = \frac{1}{2}\left(\pm\sqrt{(a-c)^2 + 4b^2} + a + c\right)$$

$$\mathbf{u}_{12} = \begin{bmatrix} \frac{\pm\sqrt{(a-c)^2+4b^2}+a-c}{2b} \\ 1 \end{bmatrix}$$

In our case, $a = \mu^2 + \sigma^2 + \lambda$, $b = \mu$, $c = 1$.

$$\mathbf{u}_{12} = \begin{bmatrix} \frac{\pm\sqrt{(\mu^2+\sigma^2+\lambda-1)^2+4\mu^2}+\mu^2+\sigma^2+\lambda-1}{2\mu} \\ 1 \end{bmatrix}$$

The faster learning dimension will be $\lambda_1, \mathbf{u}_1$, where $w$ and $b$ will move in the same direction.

**Key observation** is that the $w$ and $b$'s dynamics depend on $\sigma$,

- for larger $\sigma$, $w$ will converge faster, while $b$ will slowly meander, $w$ moving with $b$, following entrainment.

$$w^*(b) = \frac{\lambda + \mu^2 - \mu b}{\lambda + \mu^2 + \sigma^2}$$

- for smaller $\sigma$, $b$ will converge faster, comparable or entrained by $w$

$$b^*(w) = (1 - w)\mu$$

### D.3 Sampling ODE and Generated Distribution

For simplicity consider the zero-mean case, where

$$\mathbf{W}(\tau;\sigma) = \mathbf{W}^* + \sum_k \left(\mathbf{W}(0;\sigma)\mathbf{u}_k - \frac{\lambda_k}{(\sigma^2 + \lambda_k)}\mathbf{u}_k\right)\mathbf{u}_k^T e^{-2\eta(\sigma^2+\lambda_k)\tau} \tag{133}$$

$$= \sum_k \frac{\lambda_k}{(\sigma^2 + \lambda_k)}\mathbf{u}_k\mathbf{u}_k^T(1 - e^{-2\eta(\sigma^2+\lambda_k)\tau}) + \mathbf{W}(0;\sigma)\sum_k \mathbf{u}_k\mathbf{u}_k^T e^{-2\eta(\sigma^2+\lambda_k)\tau} \tag{134}$$

$$\mathbf{b}(\tau;\sigma) = \mathbf{b}(0)\exp(-2\eta\tau) \tag{135}$$

To let it decompose mode by mode in the sampling ODE, we assume aligned initialization $\mathbf{u}_k^T\mathbf{W}(0;\sigma)\mathbf{u}_m = 0$ when $k \neq m$.

Then

$$\frac{d}{d\sigma}\mathbf{u}_k^T\mathbf{x} = -\frac{1}{\sigma}\left(\left(-\frac{\sigma^2}{(\sigma^2 + \lambda_k)} + (\mathbf{u}_k^T\mathbf{W}(0;\sigma)\mathbf{u}_k - \frac{\lambda_k}{(\sigma^2 + \lambda_k)})e^{-2\eta(\sigma^2+\lambda_k)\tau}\right)\mathbf{u}_k^T\mathbf{x} + \mathbf{u}_k^T\mathbf{b}(0;\sigma)e^{-2\eta\tau}\right)$$

Let the initialization along $\mathbf{u}_k$ be $\mathbf{u}_k^T\mathbf{W}(0;\sigma)\mathbf{u}_k = q_k$, then

$$\frac{d}{d\sigma}\mathbf{u}_k^T\mathbf{x} = -\frac{1}{\sigma}\left(\left(-\frac{\sigma^2}{(\sigma^2 + \lambda_k)} + (q_k - \frac{\lambda_k}{(\sigma^2 + \lambda_k)})e^{-2\eta(\sigma^2+\lambda_k)\tau}\right)\mathbf{u}_k^T\mathbf{x} + \mathbf{u}_k^T\mathbf{b}(0;\sigma)e^{-2\eta\tau}\right)$$

Using the following integration results

$$\int d\sigma \frac{1}{\sigma(\sigma^2 + \lambda_k)}\sigma^2 = \frac{1}{2}\log\left(\lambda_k + \sigma^2\right) + C$$

$$\int d\sigma \frac{1}{\sigma}e^{-2\eta(\sigma^2+\lambda_k)\tau} = -\frac{1}{2}e^{-2\eta\lambda_k\tau}\text{Ei}\left(-2\eta\tau\sigma^2\right) + C$$

$$\int d\sigma \frac{\lambda_k}{\sigma(\sigma^2 + \lambda_k)}e^{-2\eta(\sigma^2+\lambda_k)\tau} = \frac{1}{2}\left(\text{Ei}\left(-2\eta\tau\sigma^2\right)e^{-2\eta\tau\lambda_k} - \text{Ei}\left(-2\eta\tau\left(\sigma^2 + \lambda_k\right)\right)\right) + C$$

Integrating this ODE, we get

$$c_k(\sigma) = C \exp \left( \frac{1}{2} \log \left( \lambda_k + \sigma^2 \right) + \frac{1}{2} \left( \mathrm{Ei} \left( -2\eta\tau\sigma^2 \right) e^{-2\eta\tau\lambda_k} - \mathrm{Ei} \left( -2\eta\tau \left( \sigma^2 + \lambda_k \right) \right) \right) + \tag{136}$$

$$- q_k \frac{1}{2} \mathrm{Ei} \left( -2\eta\tau\sigma^2 \right) e^{-2\eta\lambda_k\tau} \Big) \tag{137}$$

$$= C\sqrt{\lambda_k + \sigma^2} \exp \left( \frac{1}{2} \left( (1 - q_k) \, \mathrm{Ei} \left( -2\eta\tau\sigma^2 \right) e^{-2\eta\tau\lambda_k} - \mathrm{Ei} \left( -2\eta\tau \left( \sigma^2 + \lambda_k \right) \right) \right) \right) \tag{138}$$

Solution of sampling dynamics ODE

$$\mathbf{x}(\sigma_0) = \sum_k \mathbf{u}_k \frac{c_k(\sigma_0)}{c_k(\sigma_T)} (\mathbf{u}_k^T \mathbf{x}(\sigma_T)) \tag{139}$$

$$= \sum_k \frac{c_k(\sigma_0)}{c_k(\sigma_T)} \mathbf{u}_k \mathbf{u}_k^T \mathbf{x}(\sigma_T) \tag{140}$$

The variance of generated distribution reads

$$\tilde{\Sigma}^\tau = \sigma_T^2 \sum_k \left( \frac{c_k(\sigma_0)}{c_k(\sigma_T)} \right)^2 \mathbf{u}_k \mathbf{u}_k^T \tag{141}$$

where the variance along eigenvector $\mathbf{u}_k$ reads

$$\tilde{\lambda}_k^\tau = \sigma_T^2 \left( \frac{c_k(\sigma_0)}{c_k(\sigma_T)} \right)^2 \tag{142}$$

$$= \sigma_T^2 \frac{\lambda_k + \sigma_0^2}{\lambda_k + \sigma_T^2} \frac{\exp \left( (1 - q_k) \, \mathrm{Ei} \left( -2\eta\tau\sigma_0^2 \right) e^{-2\eta\tau\lambda_k} - \mathrm{Ei} \left( -2\eta\tau \left( \sigma_0^2 + \lambda_k \right) \right) \right)}{\exp \left( (1 - q_k) \, \mathrm{Ei} \left( -2\eta\tau\sigma_T^2 \right) e^{-2\eta\tau\lambda_k} - \mathrm{Ei} \left( -2\eta\tau \left( \sigma_T^2 + \lambda_k \right) \right) \right)} \tag{143}$$

# E Detailed Derivations for Two-Layer Symmetric Parameterization

Here we outline the main derivation steps for the two-layer symmetric case:

$$\mathbf{D}(\mathbf{x}) = PP^T\mathbf{x} + \mathbf{b}. \tag{144}$$

## E.1 Symmetric parametrization zero mean gradient dynamics

Note that if the weight matrix $\mathbf{W}$ has internal structure, i.e., parametrized by $\theta$, we can easily derive the gradient flow of those parameters using the chain rule.

Let $\mathbf{W} = \mathbf{W}(\theta)$,

$$\nabla_\theta \mathcal{L} = \sum_{ij} (\frac{\partial \mathcal{L}}{\partial \mathbf{W}})_{ij} \frac{\partial \mathbf{W}_{ij}}{\partial \theta}$$

Here, when $\mathbf{W} = PP^T$, via the chain rule, we can derive its gradient, which depends on the symmetrized gradient of $\mathbf{W}$

$$\nabla_P \mathcal{L} = (\nabla_\mathbf{W}\mathcal{L})P + (\nabla_\mathbf{W}\mathcal{L})^T P \tag{145}$$

$$= \left[\nabla_\mathbf{W}\mathcal{L} + (\nabla_\mathbf{W}\mathcal{L})^T\right]P \tag{146}$$

where

$$\nabla_\mathbf{b}\mathcal{L} = 2(\mathbf{b} - (\mathbf{I} - \mathbf{W})\mu) \tag{147}$$

$$\nabla_\mathbf{W}\mathcal{L} = -2\mathbf{\Sigma} + 2\mathbf{W}(\sigma^2\mathbf{I} + \mathbf{\Sigma}) + [2\mathbf{W}\mu\mu^T + 2(\mathbf{b} - \mu)\mu^T] \tag{148}$$

Expanding the full gradient (non-zero mean case), we have

$$\nabla_P \mathcal{L} = \Big[ -4\mathbf{\Sigma} + 2\mathbf{W}(\sigma^2\mathbf{I} + \mathbf{\Sigma}) + [2\mathbf{W}\mu\mu^T + 2(b - \mu)\mu^T]$$

$$+ 2(\sigma^2\mathbf{I} + \mathbf{\Sigma})\mathbf{W}^T + [2\mu\mu^T\mathbf{W}^T + 2\mu(b - \mu)^T]\Big]P \tag{149}$$

$$= -4\mathbf{\Sigma}P + 2PP^T(\sigma^2\mathbf{I} + \mathbf{\Sigma})P + 2(\sigma^2\mathbf{I} + \mathbf{\Sigma})PP^T P$$

$$+ [2PP^T\mu\mu^T + 2(b - \mu)\mu^T]P + [2\mu\mu^T PP^T + 2\mu(b - \mu)^T]P \tag{150}$$

In the zero-mean case this simplifies to

$$\nabla_P \mathcal{L}_{\mu=0} = -4\mathbf{\Sigma}P + 2PP^T(\sigma^2\mathbf{I} + \mathbf{\Sigma})P + 2(\sigma^2\mathbf{I} + \mathbf{\Sigma})PP^T P \tag{151}$$

Consider representing the gradient on eigenbases, let $\mathbf{u}_k^T P = q_k^T$.

$$\mathbf{u}_k^T\nabla_P \mathcal{L}_{\mu=0} = -4\mathbf{u}_k^T\mathbf{\Sigma}P + 2\mathbf{u}_k^T PP^T(\sigma^2\mathbf{I} + \mathbf{\Sigma})P + 2\mathbf{u}_k^T(\sigma^2\mathbf{I} + \mathbf{\Sigma})PP^T P \tag{152}$$

$$= -4\lambda_k\mathbf{u}_k^T P + 2\mathbf{u}_k^T P\sum_m(\sigma^2 + \lambda_m)P^T\mathbf{u}_m\mathbf{u}_m^T P + 2(\sigma^2 + \lambda_k)\mathbf{u}_k^T PP^T P \tag{153}$$

$$= -4\lambda_k q_k^T + 2q_k^T\sum_m(\sigma^2 + \lambda_m)q_m q_m^T + 2(\sigma^2 + \lambda_k)q_k^T\sum_m q_m q_m^T \tag{154}$$

$$= -4\lambda_k q_k^T + 2\sum_m\left(2\sigma^2 + \lambda_m + \lambda_k\right)(q_k^T q_m)q_m^T \tag{155}$$

$$\nabla_{q_k^T}\mathcal{L}_{\mu=0} = -4\lambda_k q_k^T + 2\sum_m\left(2\sigma^2 + \lambda_m + \lambda_k\right)(q_k^T q_m)q_m^T \tag{156}$$

$$\nabla_{q_k}\mathcal{L}_{\mu=0} = -4\lambda_k q_k + 2\sum_m\left(2\sigma^2 + \lambda_m + \lambda_k\right)(q_k^T q_m)q_m \tag{157}$$

**Fixed points analysis**    A stationary solution at which the gradient vanishes is

$$q_k^T q_m = \begin{cases} 0, & \text{if } k \neq m, \\ \dfrac{\lambda_k}{\lambda_k + \sigma^2} \text{ or } 0, & \text{if } k = m. \end{cases} \tag{158}$$

Note, this is different from the one-layer case where there are no saddle points; here we get a bunch of zero solutions as saddle points.

**Dynamics of the overlap** Note the dynamics of the overlap

$$
\begin{aligned}
\frac{d}{d\tau}(q_k^T q_m) &= q_k^T \frac{d}{d\tau} q_m + q_m^T \frac{d}{d\tau} q_k \\
&= -\eta [q_k^T \nabla_{q_m} \mathcal{L}_{\mu=0} + q_m^T \nabla_{q_k} \mathcal{L}_{\mu=0}] \\
&= -\eta [-4\lambda_k (q_m^T q_k) + 2 \sum_n (2\sigma^2 + \lambda_n + \lambda_k)(q_k^T q_n) q_m^T q_n \\
&\quad - 4\lambda_m (q_k^T q_m) + 2 \sum_n (2\sigma^2 + \lambda_n + \lambda_m)(q_m^T q_n) q_k^T q_n] \\
&= -\eta [-4(\lambda_k + \lambda_m)(q_m^T q_k) + 2 \sum_n (4\sigma^2 + 2\lambda_n + \lambda_m + \lambda_k)(q_k^T q_n)(q_m^T q_n)] \\
&= 4\eta [(\lambda_k + \lambda_m)(q_m^T q_k) - \sum_n (2\sigma^2 + \lambda_n + \frac{\lambda_m + \lambda_k}{2})(q_k^T q_n)(q_m^T q_n)] \quad (159)
\end{aligned}
$$

This shows that when all the overlaps are initialized as zero, they will stay at zero, i.e., when weights are initialized to be aligned to the eigenbasis, they will stay aligned. We will first solve the aligned case analytically in Sec. E.1.1, and then analyze the dynamics of overlap qualitatively in Sec. E.1.2.

### E.1.1 Simplifying assumption: orthogonal initialization $q_k^T q_m = 0$

Consider the simple case where each $q_k^T q_m = 0$, $\forall k \neq m$ at network initialization, i.e., each $q$ are orthogonal to each other. Then, it's easy to show that $\frac{d}{d\tau}(q_k^T q_m) = 0$ at the start and throughout training. Thus, we know orthogonally initialized modes will evolve independently.

Note this assumption can also be written as

$$
q_k^T q_m = \mathbf{u}_k^T P P^T \mathbf{u}_m = \mathbf{u}_k^T \mathbf{W} \mathbf{u}_m = 0 \quad \forall k \neq m
$$

which means the eigenvectors of $\mathbf{\Sigma}$ are still orthogonal w.r.t. matrix $\mathbf{W}$, i.e., the matrix $\mathbf{W}$ shares eigenbases with the data covariance $\mathbf{\Sigma}$.

In such case, the gradient reads

$$
\nabla_{q_k} \mathcal{L}_{\mu=0} = -4\lambda_k q_k + 2 \sum_m (2\sigma^2 + \lambda_m + \lambda_k)(q_k^T q_m) q_m \quad (160)
$$

$$
\text{(ortho)} = -4\lambda_k q_k + 2(2\sigma^2 + 2\lambda_k)(q_k^T q_k) q_k \quad (161)
$$

$$
= -4\lambda_k q_k + 4(\sigma^2 + \lambda_k)(q_k^T q_k) q_k \quad (162)
$$

The dynamics read

$$
\frac{dq_k}{d\tau} = -\eta \nabla_{q_k} \mathcal{L}_{\mu=0} \quad (163)
$$

$$
= -\eta(-4\lambda_k q_k + 4(\sigma^2 + \lambda_k)(q_k^T q_k) q_k) \quad (164)
$$

$$
= 4\eta(\lambda_k - (\sigma^2 + \lambda_k)(q_k^T q_k)) q_k \quad (165)
$$

Since the right hand side is aligned with $q_k$, it can only move by scaling the initial value.

The fixed-point solution is $q_k = 0$ or when $q_k^T q_k = \frac{\lambda_k}{\sigma^2 + \lambda_k}$. Given the arbitrariness of $q_k$ itself, we track the dynamics of its squared norm. The learning dynamics of $q_k^T q_k$ read

$$
q_k^T \frac{dq_k}{d\tau} = 4\eta(\lambda_k - (\sigma^2 + \lambda_k)(q_k^T q_k))(q_k^T q_k) \quad (166)
$$

$$
\frac{1}{2} \frac{d(q_k^T q_k)}{d\tau} = 4\eta(\lambda_k - (\sigma^2 + \lambda_k)(q_k^T q_k))(q_k^T q_k) \quad (167)
$$

$$
\frac{d(f)}{d\tau} = 8\eta(\lambda_k - (\sigma^2 + \lambda_k) f) f \quad (168)
$$

Fortunately, this ODE has a closed-form solution.

$$(q_k^T q_k)(\tau) = \|q_k(\tau)\|^2 = \frac{a}{1/Ke^{-at} + b} \tag{169}$$

$$= \frac{8\eta\lambda_k}{1/Ke^{-8\eta\lambda_k\tau} + 8\eta(\sigma^2 + \lambda_k)} \tag{170}$$

$$= \frac{8\eta\lambda_k}{(\frac{8\eta\lambda_k}{\|q_k(0)\|^2} - 8\eta(\sigma^2 + \lambda_k))e^{-8\eta\lambda_k\tau} + 8\eta(\sigma^2 + \lambda_k)} \tag{171}$$

$$= \frac{\lambda_k}{(\frac{\lambda_k}{\|q_k(0)\|^2} - (\sigma^2 + \lambda_k))e^{-8\eta\lambda_k\tau} + (\sigma^2 + \lambda_k)} \tag{172}$$

$$= \frac{\lambda_k}{\sigma^2 + \lambda_k}\left(\frac{1}{(\frac{1}{\|q_k(0)\|^2}\frac{\lambda_k}{\sigma^2+\lambda_k} - 1)e^{-8\eta\lambda_k\tau} + 1}\right) \tag{173}$$

$$(q_k^T q_m)(\tau) = 0 \quad \text{if } k \neq m \tag{174}$$

This gives rise to the full vector solution

$$q_k(\tau) = \sqrt{\|q_k(\tau)\|^2}\frac{q_k(0)}{\|q_k(0)\|} \tag{175}$$

$$= \sqrt{\frac{\lambda_k}{\sigma^2 + \lambda_k}}\left(\frac{1}{\sqrt{(\frac{1}{\|q_k(0)\|^2}\frac{\lambda_k}{\sigma^2+\lambda_k} - 1)e^{-8\eta\lambda_k\tau} + 1}}\right)\frac{q_k(0)}{\|q_k(0)\|} \tag{176}$$

$$= \sqrt{\frac{\lambda_k}{\sigma^2 + \lambda_k}}\left(\frac{1}{\sqrt{(\frac{\lambda_k}{\sigma^2+\lambda_k} - \|q_k(0)\|^2)e^{-8\eta\lambda_k\tau} + \|q_k(0)\|^2}}\right)q_k(0) \tag{177}$$

$$= \sqrt{\frac{\lambda_k}{\sigma^2 + \lambda_k}}\left(\frac{1}{\sqrt{\frac{\lambda_k}{\sigma^2+\lambda_k}e^{-8\eta\lambda_k\tau} + (1 - e^{-8\eta\lambda_k\tau})\|q_k(0)\|^2}}\right)q_k(0) \tag{178}$$

**Learning Dynamics of score estimator** Now, reconstruct the whole estimator, $\mathbf{W} = PP^T$. Recall $\mathbf{u}_k^T P = q_k^T$

$$P = \sum_k \mathbf{u}_k\mathbf{u}_k^T P$$

$$= \sum_k \mathbf{u}_k q_k^T$$

$$PP^T = (\sum_k \mathbf{u}_k q_k^T)(\sum_m q_m\mathbf{u}_m^T)$$

$$= \sum_k\sum_m \mathbf{u}_k(q_k^T q_m)\mathbf{u}_m^T$$

Note that under our assumption, $(q_k^T q_m)(\tau) = 0$ if $k \neq m$, $q_k^T q_m$ is diagonal. So

$$PP^T = \sum_k \mathbf{u}_k(q_k^T q_k)\mathbf{u}_k^T$$

$$= \sum_k \|q_k(\tau)\|^2\mathbf{u}_k\mathbf{u}_k^T$$

Then we can rewrite

$$\|q_k(\tau)\|^2 = (q_k^T q_k)(\tau) = \frac{\lambda_k}{\sigma^2 + \lambda_k}\left(\frac{1}{(\frac{1}{\|q_k(0)\|^2}\frac{\lambda_k}{\sigma^2+\lambda_k} - 1)e^{-8\eta\lambda_k\tau} + 1}\right) \tag{179}$$

$$\mathbf{W}(\tau) = PP^T(\tau) = \sum_k \|q_k(\tau)\|^2\mathbf{u}_k\mathbf{u}_k^T$$

$$= \sum_k \frac{\lambda_k}{\sigma^2 + \lambda_k}\mathbf{u}_k\mathbf{u}_k^T\left(\frac{1}{(\frac{1}{\|q_k(0)\|^2}\frac{\lambda_k}{\sigma^2+\lambda_k} - 1)e^{-8\eta\lambda_k\tau} + 1}\right) \tag{180}$$

**Score estimation error dynamics**

$$E_s = \mathbb{E}_{\mathbf{x}} \|\mathbf{s}(\mathbf{x}) - \mathbf{s}^*(\mathbf{x})\|^2 = \frac{1}{\sigma^4} \Bigg[ \|\mathbf{b} - \mathbf{b}^*\|^2 + 2(\mathbf{b} - \mathbf{b}^*)^T (\mathbf{W} - \mathbf{W}^*) \mu \tag{181}$$

$$+ \operatorname{Tr}[(\mathbf{W} - \mathbf{W}^*)^T (\mathbf{W} - \mathbf{W}^*)(\mu\mu^T + \mathbf{\Sigma} + \sigma^2 \mathbf{I})] \Bigg]$$

$$E_s^{(\mu=0)} = \frac{1}{\sigma^4} \Bigg[ \|\mathbf{b} - \mathbf{b}^*\|^2 + \operatorname{Tr}[(\mathbf{W} - \mathbf{W}^*)^T (\mathbf{W} - \mathbf{W}^*)(\mathbf{\Sigma} + \sigma^2 \mathbf{I})] \Bigg] \tag{182}$$

$$= \frac{1}{\sigma^4} \Bigg[ \|\mathbf{b} - \mathbf{b}^*\|^2 + \operatorname{Tr}[(\mathbf{W} - \mathbf{W}^*)^T (\mathbf{W} - \mathbf{W}^*) \sum_k (\lambda_k + \sigma^2) \mathbf{u}_k \mathbf{u}_k^T] \Bigg] \tag{183}$$

Thus

$$E_s^{(\mu=0, \text{ cov term})} = \operatorname{Tr}[(\mathbf{W} - \mathbf{W}^*)^T (\mathbf{W} - \mathbf{W}^*) \sum_k (\lambda_k + \sigma^2) \mathbf{u}_k \mathbf{u}_k^T] \tag{184}$$

$$= \operatorname{Tr}[\sum_k (\lambda_k + \sigma^2)(\mathbf{W} - \mathbf{W}^*) \mathbf{u}_k \mathbf{u}_k^T (\mathbf{W} - \mathbf{W}^*)^T] \tag{185}$$

$$= \operatorname{Tr}\Bigg[ \sum_k (\lambda_k + \sigma^2)\big(Pq_k - \frac{\lambda_k}{\lambda_k + \sigma^2} \mathbf{u}_k\big)\big(Pq_k - \frac{\lambda_k}{\lambda_k + \sigma^2} \mathbf{u}_k\big)^T \Bigg] \tag{186}$$

$$= \operatorname{Tr}\Bigg[ \sum_k (\lambda_k + \sigma^2)\big(Pq_k - \frac{\lambda_k}{\lambda_k + \sigma^2} \mathbf{u}_k\big)^T \big(Pq_k - \frac{\lambda_k}{\lambda_k + \sigma^2} \mathbf{u}_k\big) \Bigg] \tag{187}$$

$$= \sum_k (\lambda_k + \sigma^2)\big(q_k^T P^T P q_k - 2\frac{\lambda_k}{\lambda_k + \sigma^2} \mathbf{u}_k^T P q_k + (\frac{\lambda_k}{\lambda_k + \sigma^2})^2 \mathbf{u}_k^T \mathbf{u}_k\big) \tag{188}$$

$$= \sum_k (\lambda_k + \sigma^2)\big((q_k^T q_k)^2 - 2\frac{\lambda_k}{\lambda_k + \sigma^2} q_k^T q_k + (\frac{\lambda_k}{\lambda_k + \sigma^2})^2\big) \tag{189}$$

$$= \sum_k (\lambda_k + \sigma^2)\big((q_k^T q_k) - \frac{\lambda_k}{\lambda_k + \sigma^2}\big)^2 \tag{190}$$

$$E_s^{(\mu=0, \text{ bias term})} = (b^0)^2 \exp(-4\eta\tau)$$

The dynamics can be applied

$$(q_k^T q_k)(\tau) = \frac{\lambda_k}{\sigma^2 + \lambda_k} \left( \frac{1}{(\frac{1}{\|q_k(0)\|^2} \frac{\lambda_k}{\sigma^2 + \lambda_k} - 1)e^{-8\eta\lambda_k\tau} + 1} \right)$$

$$E_s^{(\mu=0, \text{ cov term})} = \sum_k (\lambda_k + \sigma^2)\big(\frac{\lambda_k}{\sigma^2 + \lambda_k} \left( \frac{1}{(\frac{1}{\|q_k(0)\|^2} \frac{\lambda_k}{\sigma^2 + \lambda_k} - 1)e^{-8\eta\lambda_k\tau} + 1} \right) - \frac{\lambda_k}{\lambda_k + \sigma^2}\big)^2 \tag{191}$$

$$= \sum_k (\lambda_k + \sigma^2)(\frac{\lambda_k}{\sigma^2 + \lambda_k})^2 \Big( \left( \frac{1}{(\frac{1}{\|q_k(0)\|^2} \frac{\lambda_k}{\sigma^2 + \lambda_k} - 1)e^{-8\eta\lambda_k\tau} + 1} \right) - 1 \Big)^2 \tag{192}$$

$$= \sum_k \frac{\lambda_k^2}{\lambda_k + \sigma^2} \left( \frac{(\frac{1}{\|q_k(0)\|^2} \frac{\lambda_k}{\sigma^2 + \lambda_k} - 1)e^{-8\eta\lambda_k\tau}}{(\frac{1}{\|q_k(0)\|^2} \frac{\lambda_k}{\sigma^2 + \lambda_k} - 1)e^{-8\eta\lambda_k\tau} + 1} \right)^2 \tag{193}$$

$$= \sum_k \frac{\lambda_k^2}{\lambda_k + \sigma^2} \left( \frac{(\frac{\lambda_k}{\sigma^2 + \lambda_k} - \|q_k(0)\|^2)e^{-8\eta\lambda_k\tau}}{(\frac{\lambda_k}{\sigma^2 + \lambda_k} - \|q_k(0)\|^2)e^{-8\eta\lambda_k\tau} + \|q_k(0)\|^2} \right)^2 \tag{194}$$

The full score estimation loss is

$$E_s^{(\mu=0)} = (b^0)^2 \exp(-4\eta\tau) + \tag{195}$$

$$\sum_k \frac{\lambda_k^2}{\lambda_k + \sigma^2} \left( \frac{(\frac{\lambda_k}{\sigma^2 + \lambda_k} - \|q_k(0)\|^2)e^{-8\eta\lambda_k\tau}}{(\frac{\lambda_k}{\sigma^2 + \lambda_k} - \|q_k(0)\|^2)e^{-8\eta\lambda_k\tau} + \|q_k(0)\|^2} \right)^2$$

### E.1.2 Beyond Aligned Initialization: qualitative analysis of off diagonal dynamics

Next we can write down the dynamics of the non-diagonal part of the weight, i.e., overlaps between $q_k$ and $q_m$

$$\frac{d}{d\tau}(q_k^T q_m) \tag{196}$$

$$= q_k^T \frac{d}{d\tau} q_m + q_m^T \frac{d}{d\tau} q_k \tag{197}$$

$$= -\eta[-4(\lambda_k + \lambda_m)(q_m^T q_k) + 2\sum_n (4\sigma^2 + 2\lambda_n + \lambda_m + \lambda_k)(q_k^T q_n)(q_m^T q_n)] \tag{198}$$

$$= 4\eta[(\lambda_k + \lambda_m)(q_m^T q_k) - \sum_n (2\sigma^2 + \lambda_n + \frac{\lambda_m + \lambda_k}{2})(q_k^T q_n)(q_m^T q_n)] \tag{199}$$

$$= 4\eta\Big[(\lambda_k + \lambda_m)(q_m^T q_k) - (2\sigma^2 + \lambda_k + \frac{\lambda_m + \lambda_k}{2})(q_k^T q_k)(q_m^T q_k) \tag{200}$$

$$- (2\sigma^2 + \lambda_m + \frac{\lambda_m + \lambda_k}{2})(q_k^T q_m)(q_m^T q_m)$$

$$- \sum_{n \neq k,m} (2\sigma^2 + \lambda_n + \frac{\lambda_m + \lambda_k}{2})(q_k^T q_n)(q_m^T q_n)\Big]$$

$$= 4\eta\Big[\Big(\lambda_k + \lambda_m - (2\sigma^2 + \frac{\lambda_m + 3\lambda_k}{2})\|q_k\|^2 - (2\sigma^2 + \frac{3\lambda_m + \lambda_k}{2})\|q_m\|^2\Big)(q_m^T q_k) \tag{201}$$

$$- \sum_{n \neq k,m} (2\sigma^2 + \lambda_n + \frac{\lambda_m + \lambda_k}{2})(q_k^T q_n)(q_m^T q_n)\Big]$$

$$= 4\eta\Big[\Big(\lambda_k(1 - \|q_k\|^2) + \lambda_m(1 - \|q_m\|^2) - (2\sigma^2 + \frac{\lambda_m + \lambda_k}{2})(\|q_k\|^2 + \|q_m\|^2)\Big)(q_m^T q_k) \tag{202}$$

$$- \sum_{n \neq k,m} (2\sigma^2 + \lambda_n + \frac{\lambda_m + \lambda_k}{2})(q_k^T q_n)(q_m^T q_n)\Big]$$

As a reference, recall the dynamics of diagonal term,

$$\frac{d(q_k^T q_k)}{d\tau} = 8\eta(\lambda_k - (\sigma^2 + \lambda_k)(q_k^T q_k))(q_k^T q_k)$$

Assume the diagonal term is not stuck at zero $q_k^T q_k \neq 0$, then we know it will asymptotically go to $\|q_k(\infty)\|^2 \approx \frac{\lambda_k}{\sigma^2 + \lambda_k}$ following sigmoidal dynamics.

For overlap between $q_k^T q_m$, without loss of generality, assume $\lambda_k > \lambda_m$. Then we have three dynamic phases: 1) neither mode has emerged; 2) mode $k$ has emerged, while $m$ has not; 3) both modes have emerged.

**Phase 1: neither mode has emerged**    When neither mode has emerged, assume $\|q_k(\tau)\|^2 \approx 0$, then the overlap will grow exponentially.

$$\frac{d}{d\tau}(q_k^T q_m) \approx 4\eta\Big(\lambda_k + \lambda_m\Big)(q_m^T q_k)$$

Note given $\lambda_k > \lambda_m$, the diagonal term has a higher increasing speed than the non-diagonal overlap term $8\eta(\lambda_k - (\sigma^2 + \lambda_k)(q_k^T q_k)) > 4\eta(\lambda_k + \lambda_m)$.

**Phase 2: one mode has emerged, the other has not** When one mode has converged $\|q_k(\tau)\|^2 \approx \frac{\lambda_k}{\sigma^2 + \lambda_k}$, but the other has not ($\lambda_k > \lambda_m$)

$$\frac{d}{d\tau}(q_k^T q_m) \approx 4\eta\left(\lambda_k + \lambda_m - (2\sigma^2 + \frac{\lambda_m + 3\lambda_k}{2})\frac{\lambda_k}{\sigma^2 + \lambda_k}\right)(q_m^T q_k) \tag{203}$$

$$= 4\eta\left(\lambda_k + \lambda_m - (2\sigma^2 + 2\lambda_k + \frac{\lambda_m - \lambda_k}{2})\frac{\lambda_k}{\sigma^2 + \lambda_k}\right)(q_m^T q_k) \tag{204}$$

$$= 4\eta\left(\lambda_k + \lambda_m - 2\lambda_k - (\frac{\lambda_m - \lambda_k}{2})\frac{\lambda_k}{\sigma^2 + \lambda_k}\right)(q_m^T q_k) \tag{205}$$

$$= 4\eta\left(\lambda_m - \lambda_k - (\frac{\lambda_m - \lambda_k}{2})\frac{\lambda_k}{\sigma^2 + \lambda_k}\right)(q_m^T q_k) \tag{206}$$

$$= 4\eta(\lambda_m - \lambda_k)\left(1 - \frac{1}{2}\frac{\lambda_k}{\sigma^2 + \lambda_k}\right)(q_m^T q_k) \tag{207}$$

Since $\frac{\lambda_k}{\sigma^2 + \lambda_k} < 1$, we know $1 - \frac{1}{2}\frac{\lambda_k}{\sigma^2 + \lambda_k} > 0$. Thus, the dynamic coefficient $(\lambda_m - \lambda_k)(1 - \frac{1}{2}\frac{\lambda_k}{\sigma^2 + \lambda_k}) < 0$, so the overlap will follow an exponential decay dynamics. Further, given the same $\lambda_k$, the decay speed of $q_k^T q_m$ is proportional to the difference of the eigenvalues $\lambda_m - \lambda_k$. So, the larger the difference, the faster the decay.

**Phase 3: both modes have emerged** When both modes have emerged $\|q_k(\tau)\|^2 \approx \frac{\lambda_k}{\sigma^2 + \lambda_k}$, $\|q_m(\tau)\|^2 \approx \frac{\lambda_m}{\sigma^2 + \lambda_m}$,

$$\frac{d}{d\tau}(q_k^T q_m) \approx 4\eta\left(\lambda_k + \lambda_m - (2\sigma^2 + \frac{\lambda_m + 3\lambda_k}{2})\frac{\lambda_k}{\sigma^2 + \lambda_k} - (2\sigma^2 + \frac{3\lambda_m + \lambda_k}{2})\frac{\lambda_m}{\sigma^2 + \lambda_m}\right)(q_m^T q_k) \tag{208}$$

$$= 4\eta\left(-\lambda_k - \lambda_m + (\frac{\sigma^2}{2})(\frac{(\lambda_m - \lambda_k)^2}{(\sigma^2 + \lambda_m)(\sigma^2 + \lambda_k)})\right)(q_m^T q_k) \tag{209}$$

$$= -4\eta\frac{(\lambda_k + \lambda_m + 2\sigma^2)(2\lambda_m\lambda_k + \sigma^2(\lambda_k + \lambda_m))}{2(\lambda_k + \sigma^2)(\lambda_m + \sigma^2)}(q_m^T q_k) \tag{210}$$

The dynamic coefficient is negative, so the overlap decays even more rapidly.

**Summary: qualitative description of off diagonal elements $q_k^T q_m$ dynamics**

- The off-diagonal term will exponentially rise after the rise of the larger variance dimension, and before the rise of smaller variance dimension;
- after one has risen it will decay to zero;
- after both have risen it will decay faster.

Thus, it will follow non-monotonic rise and fall dynamics.

## E.2 Sampling ODE and Generated Distribution

Here, for tractability purposes, we will focus on the zero-mean and aligned initialization case.

Recall the weight learning solutions above (180),

$$\mathbf{W}(\tau; \sigma) = PP^T(\tau) = \sum_k \|q_k(\tau)\|^2 \mathbf{u}_k \mathbf{u}_k^T$$

$$= \sum_k \frac{\lambda_k}{\sigma^2 + \lambda_k} \mathbf{u}_k \mathbf{u}_k^T\left((\frac{1}{\|q_k(0;\sigma)\|^2}\frac{\lambda_k}{\sigma^2 + \lambda_k} - 1)e^{-8\eta\tau\lambda_k} + 1\right)^{-1} \tag{211}$$

with the weight initialization

$$\mathbf{W}(0; \sigma) = \sum_k \|q_k(0;\sigma)\|^2 \mathbf{u}_k \mathbf{u}_k^T \tag{212}$$

Note here we assume the initialization $\|q_k(0;\sigma)\|^2$ is the same for all $\sigma$ levels, with no $\sigma$ dependency. $\|q_k(0;\sigma)\|^2 = \|q_k(0)\|^2 = Q_k$

Per our general analysis (Sec. C.5), the key factors are

$$\psi_k(\sigma;\tau) = \frac{\lambda_k}{\sigma^2 + \lambda_k} \frac{1}{(\frac{1}{Q_k}\frac{\lambda_k}{\sigma^2+\lambda_k} - 1)e^{-8\eta\tau\lambda_k} + 1} \tag{213}$$

and its integration

$$\Phi_k(\sigma) = \exp\left(-\int^\sigma \frac{\psi_k(\sigma';\tau) - 1}{\sigma'} d\sigma'\right)$$

$$\frac{d}{d\sigma}\mathbf{u}_k^T\mathbf{x} = -\frac{1}{\sigma}\left(\psi_k(\sigma;\tau) - 1\right)\mathbf{u}_k^T\mathbf{x} \tag{214}$$

$$= -\frac{1}{\sigma}\left(\frac{\lambda_k}{\sigma^2 + \lambda_k}\frac{1}{(\frac{1}{\|q_k(0)\|^2}\frac{\lambda_k}{\sigma^2+\lambda_k} - 1)e^{-8\eta\tau\lambda_k} + 1} - 1\right)\mathbf{u}_k^T\mathbf{x} \tag{215}$$

Note that the integrand is just a fractional function of $\sigma^2$ which can be integrated analytically.

$$\int d\sigma - \frac{1}{\sigma}\left(\frac{\lambda_k}{\sigma^2 + \lambda_k}\frac{1}{(\frac{1}{Q_k}\frac{\lambda_k}{\sigma^2+\lambda_k} - 1)e^{-8\eta\tau\lambda_k} + 1} - 1\right)$$

$$= \frac{Q_k e^{8\eta\tau\lambda_k} \log\left(\lambda_k + Q_k\left(e^{8\eta\tau\lambda_k} - 1\right)\left(\lambda_k + \sigma^2\right)\right) - 2Q_k\log(\sigma) + 2\log(\sigma)}{2\left(e^{8\eta\tau\lambda_k} - 1\right)Q_k + 2}$$

$$= \frac{Q_k e^{8\eta\tau\lambda_k} \log\left(\lambda_k + Q_k\left(e^{8\eta\tau\lambda_k} - 1\right)\left(\lambda_k + \sigma^2\right)\right) + 2(1 - Q_k)\log(\sigma)}{2\left(e^{8\eta\tau\lambda_k} - 1\right)Q_k + 2}$$

$$= \frac{Q_k \log\left(\lambda_k + Q_k\left(e^{8\eta\tau\lambda_k} - 1\right)\left(\lambda_k + \sigma^2\right)\right) + 2(1 - Q_k)\log(\sigma)e^{-8\eta\tau\lambda_k}}{2\left(1 - e^{-8\eta\tau\lambda_k}\right)Q_k + 2e^{-8\eta\tau\lambda_k}}$$

$$= \frac{Q_k \log[e^{8\eta\tau\lambda_k}\left(\lambda_k e^{-8\eta\tau\lambda_k} + Q_k\left(1 - e^{-8\eta\tau\lambda_k}\right)\left(\lambda_k + \sigma^2\right)\right)] + 2(1 - Q_k)\log(\sigma)e^{-8\eta\tau\lambda_k}}{2Q_k + 2(1 - Q_k)e^{-8\eta\tau\lambda_k}}$$

$$= \frac{Q_k \log\left(\lambda_k e^{-8\eta\tau\lambda_k} + Q_k\left(1 - e^{-8\eta\tau\lambda_k}\right)\left(\lambda_k + \sigma^2\right)\right) + 8\eta\tau\lambda_k Q_k + 2(1 - Q_k)\log(\sigma)e^{-8\eta\tau\lambda_k}}{2Q_k + 2(1 - Q_k)e^{-8\eta\tau\lambda_k}}$$

Thus, the general solution to the sampling ODE is

$$c_k(\sigma) = C \, \exp\left(\frac{Q_k \log\left(\lambda_k e^{-8\eta\tau\lambda_k} + Q_k\left(1 - e^{-8\eta\tau\lambda_k}\right)\left(\lambda_k + \sigma^2\right)\right) + 8\eta\tau\lambda_k Q_k + 2(1 - Q_k)\log(\sigma)e^{-8\eta\tau\lambda_k}}{2Q_k + 2(1 - Q_k)e^{-8\eta\tau\lambda_k}}\right)$$

$$= C\left[\lambda_k e^{-8\eta\tau\lambda_k} + Q_k\left(1 - e^{-8\eta\tau\lambda_k}\right)\left(\lambda_k + \sigma^2\right)\right]^{\frac{Q_k}{2Q_k + 2(1 - Q_k)e^{-8\eta\tau\lambda_k}}} \times$$

$$\exp\left(\frac{8\eta\tau\lambda_k Q_k}{2Q_k + 2(1 - Q_k)e^{-8\eta\tau\lambda_k}}\right)\exp\left(\frac{(1 - Q_k)e^{-8\eta\tau\lambda_k}\log(\sigma)}{Q_k + (1 - Q_k)e^{-8\eta\tau\lambda_k}}\right) \tag{216}$$

The scaling ratio is

$$\frac{c_k(\sigma_0)}{c_k(\sigma_T)} = \left[\frac{\lambda_k e^{-8\eta\tau\lambda_k} + Q_k\left(1 - e^{-8\eta\tau\lambda_k}\right)\left(\lambda_k + \sigma_0^2\right)}{\lambda_k e^{-8\eta\tau\lambda_k} + Q_k\left(1 - e^{-8\eta\tau\lambda_k}\right)\left(\lambda_k + \sigma_T^2\right)}\right]^{\frac{Q_k}{2Q_k + 2(1 - Q_k)e^{-8\eta\tau\lambda_k}}} \tag{217}$$

$$\exp\left(\frac{(1 - Q_k)e^{-8\eta\tau\lambda_k}(\log(\sigma_0) - \log(\sigma_T))}{Q_k + (1 - Q_k)e^{-8\eta\tau\lambda_k}}\right)$$

$$= \left[\frac{\lambda_k e^{-8\eta\tau\lambda_k} + Q_k\left(1 - e^{-8\eta\tau\lambda_k}\right)\left(\lambda_k + \sigma_0^2\right)}{\lambda_k e^{-8\eta\tau\lambda_k} + Q_k\left(1 - e^{-8\eta\tau\lambda_k}\right)\left(\lambda_k + \sigma_T^2\right)}\right]^{\frac{Q_k}{2Q_k + 2(1 - Q_k)e^{-8\eta\tau\lambda_k}}}\left(\frac{\sigma_0}{\sigma_T}\right)^{\frac{(1 - Q_k)e^{-8\eta\tau\lambda_k}}{Q_k + (1 - Q_k)e^{-8\eta\tau\lambda_k}}} \tag{218}$$

$$\Phi_k(\sigma) = \left[\lambda_k e^{-8\eta\tau\lambda_k} + Q_k \left(1 - e^{-8\eta\tau\lambda_k}\right)\left(\lambda_k + \sigma^2\right)\right]^{\frac{Q_k}{2Q_k + 2(1-Q_k)e^{-8\eta\tau\lambda_k}}} (\sigma)^{\frac{(1-Q_k)e^{-8\eta\tau\lambda_k}}{Q_k + (1-Q_k)e^{-8\eta\tau\lambda_k}}}$$

$$(219)$$

**Evolution of distribution**   Following the derivation above,

$$\Sigma[\mathbf{x}(\sigma_0)] = \sum_k \left(\sigma_T \frac{c_k(\sigma_0)}{c_k(\sigma_T)}\right)^2 \mathbf{u}_k \mathbf{u}_k^T$$

The generated variance along eigenvector $\mathbf{u}_k$ is

$$\tilde{\lambda}_k(\tau) = \left(\sigma_T \frac{c_k(\sigma_0)}{c_k(\sigma_T)}\right)^2$$

$$= \sigma_T^2 \left[\frac{\lambda_k e^{-8\eta\tau\lambda_k} + Q_k \left(1 - e^{-8\eta\tau\lambda_k}\right)\left(\lambda_k + \sigma_0^2\right)}{\lambda_k e^{-8\eta\tau\lambda_k} + Q_k \left(1 - e^{-8\eta\tau\lambda_k}\right)\left(\lambda_k + \sigma_T^2\right)}\right]^{\frac{Q_k}{Q_k + (1-Q_k)e^{-8\eta\tau\lambda_k}}} \left(\frac{\sigma_0}{\sigma_T}\right)^{\frac{2(1-Q_k)e^{-8\eta\tau\lambda_k}}{Q_k + (1-Q_k)e^{-8\eta\tau\lambda_k}}}$$

$$(220)$$

**Asymptotics of Learning: early and late training**   At late training stage $\eta\tau \to \infty$,

$$\lim_{\eta\tau \to \infty} \frac{c_k(\sigma_0)}{c_k(\sigma_T)} = \left[\frac{Q_k \left(\lambda_k + \sigma_0^2\right)}{Q_k \left(\lambda_k + \sigma_T^2\right)}\right]^{\frac{Q_k}{2Q_k}} \exp\left(\frac{0}{Q_k}\right)$$

$$= \sqrt{\frac{\lambda_k + \sigma_0^2}{\lambda_k + \sigma_T^2}} \tag{221}$$

$$\lim_{\eta\tau \to \infty} \tilde{\lambda}_k(\tau) = \sigma_T^2 \frac{\lambda_k + \sigma_0^2}{\lambda_k + \sigma_T^2} \tag{222}$$

which approximately recovers the correct scaling factor to generate correct variance $\lambda_k$, if we take $\sigma_T \to \infty, \sigma_0 \to 0$.

At early training stage $\eta\tau \to 0$,

$$\lim_{\eta\tau \to 0} \frac{c_k(\sigma_0)}{c_k(\sigma_T)} = \left[\frac{\lambda_k 1 + Q_k \left(1 - 1\right)\left(\lambda_k + \sigma_0^2\right)}{\lambda_k 1 + Q_k \left(1 - 1\right)\left(\lambda_k + \sigma_T^2\right)}\right]^{\frac{Q_k}{2Q_k + 2(1-Q_k)}} \left(\frac{\sigma_0}{\sigma_T}\right)^{\frac{(1-Q_k)}{Q_k + (1-Q_k)}}$$

$$= \left[1\right]^{\frac{Q_k}{2Q_k + 2(1-Q_k)}} \left(\frac{\sigma_0}{\sigma_T}\right)^{1-Q_k}$$

$$= \left(\frac{\sigma_0}{\sigma_T}\right)^{1-Q_k} \tag{223}$$

which shows the initial generated variance is determined by weight initialization scale and integration limits $(\sigma_0, \sigma_T)$.

# F Detailed Derivations for Deep Linear Network

Consider a general deep linear network. $\mathbf{W} = \prod_i \mathbf{W}_i = \mathbf{W}_L \mathbf{W}_{L-1} ... \mathbf{W}_1$

**Gradient structure**   Using the chain rule

$$\nabla_y \mathcal{L} = (\frac{\partial \mathcal{L}}{\partial \mathbf{W}})_{ij} \frac{\partial \mathbf{W}_{ij}}{\partial y}$$

we have the gradient with respect to matrix $\mathbf{W}_l$

$$\frac{\partial \mathbf{W}_{mn}}{\partial \mathbf{W}_{l,ij}} = (\mathbf{W}_L ... \mathbf{W}_{l+1})_{mi} (\mathbf{W}_{l-1} ... \mathbf{W}_1)_{jn}$$

In vector notation,

$$[\nabla_{\mathbf{W}_l} \mathcal{L}]_{ij} = (\frac{\partial \mathcal{L}}{\partial \mathbf{W}})_{mn} \frac{\partial \mathbf{W}_{mn}}{\partial \mathbf{W}_{l,ij}}$$

$$= (\mathbf{W}_L ... \mathbf{W}_{l+1})_{mi} (\frac{\partial \mathcal{L}}{\partial \mathbf{W}})_{mn} (\mathbf{W}_{l-1} ... \mathbf{W}_1)_{jn}$$

$$\nabla_{\mathbf{W}_l} \mathcal{L} = (\mathbf{W}_L ... \mathbf{W}_{l+1})^T (\frac{\partial \mathcal{L}}{\partial \mathbf{W}}) (\mathbf{W}_{l-1} ... \mathbf{W}_1)^T$$

where $\nabla_{\mathbf{W}} \mathcal{L}$ can be found in (5)

$$\nabla_b \mathcal{L} = 2(b - (I - \mathbf{W})\mu)$$
$$\nabla_{\mathbf{W}} \mathcal{L} = -2\Sigma + 2\mathbf{W}(\sigma^2 I + \Sigma) + [2\mathbf{W}\mu\mu^T + 2(b - \mu)\mu^T]$$

## F.1 Aligned assumption

This gradient structure can be substantially simplified if the weight matrices are aligned at initialization.

Consider the singular value decomposition of each weight matrix $\mathbf{W}_l$

$$\mathbf{W}_l = U_l \Lambda_l V_l^T$$

and for each pair of neighboring layers, the singular modes are aligned.

$$V_l^T U_{l-1} = I$$

or equivalently $U_{l-1} = V_l, \ \forall l \in [2, L]$

**Gradient structure under aligned assumption**   Then the product of weight matrices is

$$\mathbf{W}_L ... \mathbf{W}_{l+1} = U_L \prod_{k=l+1}^{L} \Lambda_k V_{l+1}^T$$

$$\mathbf{W}_{l-1} ... \mathbf{W}_1 = U_{l-1} \prod_{k=1}^{l-1} \Lambda_k V_1^T$$

$$\mathbf{W}_L ... \mathbf{W}_1 = U_L \prod_{k=1}^{L} \Lambda_k V_1^T$$

Then the evolution of hidden weights reads

$$\nabla_{\mathbf{W}_l} \mathcal{L} = (U_L \prod_{k=l+1}^{L} \Lambda_k V_{l+1}^T)^T \nabla_{\mathbf{W}} \mathcal{L} (U_{l-1} \prod_{k=1}^{l-1} \Lambda_k V_1^T)^T$$

$$= V_{l+1} \prod_{k=l+1}^{L} \Lambda_k U_L^T \nabla_{\mathbf{W}} \mathcal{L} V_1 \prod_{k=1}^{l-1} \Lambda_k U_{l-1}^T$$

$$= U_l \left[ \prod_{k=l+1}^{L} \Lambda_k U_L^T \nabla_{\mathbf{W}} \mathcal{L} V_1 \prod_{k=1}^{l-1} \Lambda_k \right] V_l^T$$

Substituting the gradient of the loss (5), assuming centered data $\mu = 0$,

$$\nabla_{\mathbf{W}}\mathcal{L}_{\mu=0} = -2\Sigma + 2\mathbf{W}(\sigma^2 I + \Sigma)$$

Then

$$\nabla_{\mathbf{W}_l}\mathcal{L} = U_l\left[\prod_{k=l+1}^{L}\Lambda_k U_L^T\left(-2\Sigma + 2\mathbf{W}(\sigma^2 I + \Sigma)\right)V_1\prod_{k=1}^{l-1}\Lambda_k\right]V_l^T$$

$$= U_l\left[\prod_{k=l+1}^{L}\Lambda_k U_L^T\left(-2\Sigma + 2U_L\prod_{k=1}^{L}\Lambda_k V_1^T(\sigma^2 I + \Sigma)\right)V_1\prod_{k=1}^{l-1}\Lambda_k\right]V_l^T$$

$$= -2U_l\prod_{k=l+1}^{L}\Lambda_k U_L^T\Sigma V_1\prod_{k=1}^{l-1}\Lambda_k V_l^T + 2\sigma^2 U_l\prod_{k=l+1}^{L}\Lambda_k\prod_{k=1}^{L}\Lambda_k\prod_{k=1}^{l-1}\Lambda_k V_l^T$$

$$+ 2U_l\prod_{k=l+1}^{L}\Lambda_k\prod_{k=1}^{L}\Lambda_k V_1^T\Sigma V_1\prod_{k=1}^{l-1}\Lambda_k V_l^T$$

Under the simplifying assumption that $U_L = V_1$ and they exactly match the eigenbasis of $\Sigma$, denoted as $U$, then

$$V_1^T\Sigma V_1 = \Lambda, \quad U_L^T\Sigma V_1 = \Lambda$$

$$\nabla_{\mathbf{W}_l}\mathcal{L} = -2U_l\Lambda\prod_{k=l+1}^{L}\Lambda_k\prod_{k=1}^{l-1}\Lambda_k V_l^T + 2\sigma^2 U_l\prod_{k=l+1}^{L}\Lambda_k\prod_{k=1}^{L}\Lambda_k\prod_{k=1}^{l-1}\Lambda_k V_l^T + 2U_l\prod_{k=l+1}^{L}\Lambda_k\prod_{k=1}^{L}\Lambda_k\Lambda\prod_{k=1}^{l-1}\Lambda_k V_l^T$$

$$= 2U_l\left[-\Lambda + \sigma^2\prod_{k=1}^{L}\Lambda_k + \Lambda\prod_{k=1}^{L}\Lambda_k\right]\prod_{k=1,k\neq l}^{L}\Lambda_k V_l^T$$

$$= 2U_l\left[-\Lambda + (\sigma^2 + \Lambda)\Lambda_l\prod_{k=1,k\neq l}^{L}\Lambda_k\right]\prod_{k=1,k\neq l}^{L}\Lambda_k V_l^T$$

Consider the singular values of each weight matrix

$$\nabla_{\Lambda_l}\mathcal{L} = U_l^T\nabla_{\mathbf{W}_l}\mathcal{L}V_l$$

$$= 2\left[-\Lambda + (\sigma^2 + \Lambda)\Lambda_l\prod_{k=1,k\neq l}^{L}\Lambda_k\right]\prod_{k=1,k\neq l}^{L}\Lambda_k$$

The weight dynamics have a surprisingly simple mode-by-mode form. For the $m$-th mode, the gradient is

$$\nabla_{\Lambda_{l,m}}\mathcal{L} = 2\left[-\lambda_m + (\sigma^2 + \lambda_m)\Lambda_{l,m}\prod_{k=1,k\neq l}^{L}\Lambda_{k,m}\right]\prod_{k=1,k\neq l}^{L}\Lambda_{k,m}$$

To simplify notation, we define $\Xi_{l,m} := \prod_{k=1,k\neq l}^{L}\Lambda_{k,m}$. Then the gradient flow dynamics reads

$$\frac{d}{d\tau}\Lambda_{l,m} = -\eta\left[-\lambda_m + (\sigma^2 + \lambda_m)\Lambda_{l,m}\prod_{k=1,k\neq l}^{L}\Lambda_{k,m}\right]\prod_{k=1,k\neq l}^{L}\Lambda_{k,m}$$

$$= -\eta\left[-\lambda_m + (\sigma^2 + \lambda_m)\Lambda_{l,m}\Xi_{l,m}\right]\Xi_{l,m}$$

At first glance, this is a linear system for $\Lambda_{l,m}$, with fixed point at $\lambda_m/(\sigma^2 + \lambda_m)/\Xi_{l,m}$. But the effect of other layers $\Xi_{l,m}$ is also dynamic, forming a coupled nonlinear system. $\Xi_{l,m}$ affects both the time constant and convergence level.

## F.2 Two layer linear network

We start by considering the two-layer case of this equation. Under the aligned initialization assumption,

$$\mathbf{W}_1 = U_1 \Lambda_1 V_1^T, \qquad\qquad \mathbf{W}_2 = U_2 \Lambda_2 V_2^T \qquad (224)$$

$$U_1 = V_2, \qquad\qquad U_2 = V_1 = U \qquad (225)$$

So the main degrees of freedom come from $\Lambda_1, \Lambda_2$, and we discuss the two cases depending on whether they are equal to each other.

### F.2.1 Special case: homogeneous initialization $\Lambda_1 = \Lambda_2$ (symmetric two layer case)

If $\Lambda_1 = \Lambda_2$, then

$$\mathbf{W}_1 = U_1 \Lambda_1 V_1^T = V_2 \Lambda_2 U_2^T = \mathbf{W}_2^T$$

We are reduced to the symmetric two-layer case $\mathbf{W} = \mathbf{W}_1^T \mathbf{W}_1$ as we discussed above in E. Due to weight sharing, the gradients to each layer are accumulated.

$$\nabla_{\Lambda_1} \mathcal{L} = 2[-\Lambda + (\sigma^2 + \Lambda)\Lambda_1 \Lambda_2]\Lambda_2$$

$$= 2[-\Lambda + (\sigma^2 + \Lambda)\Lambda_1^2]\Lambda_1$$

$$\frac{d}{d\tau}\Lambda_1 = -2\eta[-\Lambda + (\sigma^2 + \Lambda)\Lambda_1^2]\Lambda_1$$

$$\frac{d}{d\tau}(\Lambda_1)^2 = -4\eta[-\Lambda + (\sigma^2 + \Lambda)\Lambda_1^2]\Lambda_1^2$$

This recovers the dynamics we obtained in the previous section 5.1.

### F.2.2 General Case: general initialization (general two layer $\mathbf{W} = PQ$)

Generally, if the initialization is not homogeneous, $\Lambda_1 \neq \Lambda_2$, the matrices are not the same $\mathbf{W} = PQ$, and we have the general two-layer parametrization.

Under the aligned assumption F.1, the weight learning dynamics are reduced to those of $\Lambda_1, \Lambda_2$. For simplicity of notation, let $f_k = \Lambda_{1,k}, g_k = \Lambda_{2,k}$. We have

$$\nabla_{f_k} \mathcal{L} = -\lambda_k g_k + (\sigma^2 + \lambda_k)g_k^2 f_k$$

$$\nabla_{g_k} \mathcal{L} = -\lambda_k f_k + (\sigma^2 + \lambda_k)f_k^2 g_k$$

which is a two-dimensional coupled system. Let us focus on the dynamics along one eigenmode $k$ and drop the subscript $k$ from $f_k, g_k$. Let the constants $A = \eta\lambda_k, B = \eta(\sigma^2 + \lambda_k)$, we have

$$\frac{d}{d\tau}f = Ag - Bg^2 f \qquad (226)$$

$$\frac{d}{d\tau}g = Af - Bf^2 g \qquad (227)$$

There are three important variables: $f_k g_k$, $f_k^2 + g_k^2$, and $f^2 - g^2$. This system can be represented in the following way, which exposes its conserved quantity:

$$\frac{d}{d\tau}(fg) = f\frac{d}{d\tau}g + g\frac{d}{d\tau}f$$

$$= Af^2 - Bf^3 g + Ag^2 - Bg^3 f$$

$$= (f^2 + g^2)(A - Bfg)$$

$$\frac{d}{d\tau}(f^2 + g^2) = 2f\frac{d}{d\tau}f + 2g\frac{d}{d\tau}g$$

$$= 2(Afg - Bg^2 f^2) + 2(Afg - Bf^2 g^2)$$

$$= 2fg(A - Bfg)$$

$$\frac{d}{d\tau}(f^2 - g^2) = 0$$

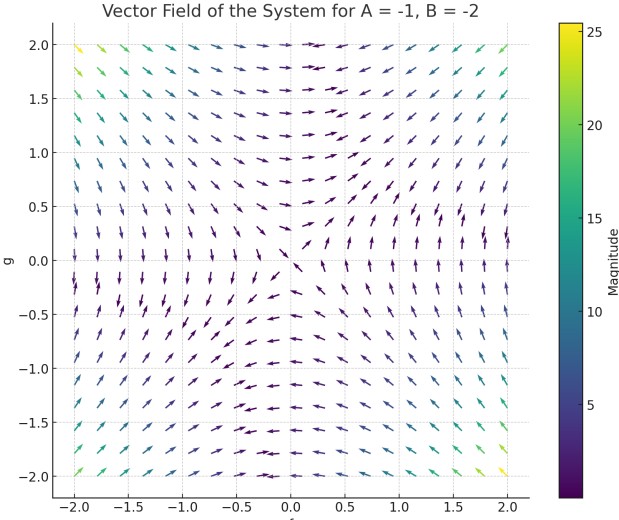

Figure 28: **Phase diagram for the simplified 2D dynamic system**. Above: $\frac{d}{d\tau}f = Ag - Bg^2f, \frac{d}{d\tau}g = Af - Bf^2g$ for $A = 1, B = 2$. We can see the manifold of stable attractors in the 1st and 3rd quadrants: $fg = A/B$, and the conserved quantity along the hyperbolic lines.

Thus the $(f, g)$ pair can only flow along hyperbolic lines or diagonal lines where $f^2 - g^2 = \text{const}$. We can leverage this fact to write down the overall solution.

Notice that

$$(f^2 + g^2)^2 - (f^2 - g^2)^2 = 4f^2g^2$$

So we can represent

$$f^2 + g^2 = \sqrt{4f^2g^2 + (f^2 - g^2)^2}$$
$$= \sqrt{4f^2g^2 + C^2}$$

Now the equation for $fg$ becomes closed:

$$\frac{d}{d\tau}(fg) = \sqrt{4f^2g^2 + C^2}(A - Bfg)$$

Let $h = fg$

$$\frac{d}{d\tau}h = \sqrt{4h^2 + C^2}(A - Bh)$$

Note that for this self-contained equation, unless $C = 0$, it shall have only one attractive fixed point which is $h^* = A/B$. Thus asymptotically, it will always converge to the correct value.

When we face the weight tying case where $C = 0$, the equation becomes

$$\frac{d}{d\tau}h = 2|h|(A - Bh)$$

It will have another fixed point at $h = 0$, which makes it impossible to converge to the fixed point $A/B$ if the initialization $h(0)$ has the opposite sign.

### F.3 General deep linear network

**Weight tying assumption / initialization** The weight tying deep network can be regarded as a special case. The key gradient equation is

$$\nabla_{\Lambda_l}\mathcal{L} = [-\Lambda + (\sigma^2 + \Lambda)\Lambda_l \prod_{k=1, k\neq l}^{L} \Lambda_k] \prod_{k=1, k\neq l}^{L} \Lambda_k$$

Let us assume all $\Lambda_l$ are equal at initialization[2]. Then

$$\nabla_{\Lambda_l}\mathcal{L} = [-\Lambda + (\sigma^2 + \Lambda)\Lambda_l^L]\Lambda_l^{L-1} \tag{228}$$

Consider a single element of the diagonal at mode $k$, $a_k^{(l)} := \Lambda_{l,k}$, then we have

$$\frac{\partial\mathcal{L}}{\partial a_k^{(l)}} = [-\lambda_k + (\sigma^2 + \lambda_k)\,(a_k^{(l)})^L](a_k^{(l)})^{L-1} \tag{229}$$

Further consider the aggregated effect $c_k = \prod_l a_k^{(l)}$. Assuming weight tying, the gradient flow of this variable reads

$$\begin{aligned}
\frac{\partial c_k}{\partial\tau} &= L(a_k^{(l)})^{L-1}\frac{\partial a_k^{(l)}}{\partial\tau}\\
&= -\eta L(a_k^{(l)})^{L-1}\frac{\partial\mathcal{L}}{\partial a_k^{(l)}}\\
&= -\eta L\,[-\lambda_k + (\sigma^2 + \lambda_k)\,(a_k^{(l)})^L]\,(a_k^{(l)})^{2L-2}\\
&= -\eta L\,[-\lambda_k + (\sigma^2 + \lambda_k)c_k]\,c_k^{\frac{2L-2}{L}}\\
&= -\eta L\,[-\lambda_k + (\sigma^2 + \lambda_k)c_k]\,c_k^{2-\frac{2}{L}}
\end{aligned}$$

We arrive at the dynamics of the weights along eigenmode:

$$\frac{\partial c_k}{\partial\tau} = \eta L[\lambda_k - (\sigma^2 + \lambda_k)c_k]c_k^{2-\frac{2}{L}} \tag{230}$$

**Depth one and two cases** Note this form also encompasses the solution of one-layer and two-layer symmetric linear models by setting $L = 1, 2$, as in (**??**) and (**??**).

**General depth case** For general $L$, there is no closed-form solution; one only has an implicit solution to $c_k$ involving the hypergeometric function ${}_2F_1$:

$$\left(\frac{A\,c(\tau)}{B}\right)^{2/L}{}_2F_1\left(1, \frac{2}{L}; 1 + \frac{2}{L}; \frac{A\,c(\tau)}{B}\right) = \left(\frac{A\,c(0)}{B}\right)^{2/L}{}_2F_1\left(1, \frac{2}{L}; 1 + \frac{2}{L}; \frac{A\,c(0)}{B}\right)e^{-\eta LB\tau}. \tag{231}$$

with substitution $A = (\sigma^2 + \lambda_k), B = \lambda_k$.

**Infinite depth limit** The limiting case is $L \to \infty$, then the dynamics read

$$\begin{aligned}
\frac{\partial c_k}{\partial\tau} &= -\eta L\,[-\lambda_k + (\sigma^2 + \lambda_k)c_k]\,c_k^{2-\frac{2}{L}}\\
(L \to \infty) &= \eta L\,[\lambda_k - (\sigma^2 + \lambda_k)c_k]\,c_k^2
\end{aligned} \tag{232}$$

Then we have

$$\frac{dc_k}{[\lambda_k - (\sigma^2 + \lambda_k)c_k]c_k^2} = \eta L\,d\tau$$

$$\frac{\sigma^2 + \lambda_k}{\lambda_k^2}\ln\frac{c_k}{\lambda_k - (\sigma^2 + \lambda_k)c_k} - \frac{1}{\lambda_k c_k} = \eta L\tau + C.$$

Setting the initial values as $c_k^{(0)}$, we have the implicit solution of $c_k$:

$$\frac{\sigma^2 + \lambda_k}{\lambda_k^2}\ln\frac{c_k}{\lambda_k - (\sigma^2 + \lambda_k)c_k} - \frac{1}{\lambda_k c_k} = \eta L\tau + \frac{\sigma^2 + \lambda_k}{\lambda_k^2}\ln\frac{c_k^{(0)}}{\lambda_k - (\sigma^2 + \lambda_k)c_k^{(0)}} - \frac{1}{\lambda_k c_k^{(0)}}$$

where training time $\tau$ can be expressed as a function of $c_k$:

$$\tau = \frac{1}{\eta L}\left[\frac{\sigma^2 + \lambda_k}{\lambda_k^2}\ln\frac{c_k(\lambda_k - (\sigma^2 + \lambda_k)c_k^{(0)})}{c_k^{(0)}(\lambda_k - (\sigma^2 + \lambda_k)c_k)} - \frac{1}{\lambda_k c_k} + \frac{1}{\lambda_k c_k^{(0)}}\right] \tag{233}$$

---

[2]$\Lambda_l$ is not to be confused with $\Lambda$, which is the spectrum of data

### F.3.1 Deep Residual network

If the network is parametrized with a residual connection at each layer, i.e., $\mathbf{W} = \prod_i (\mathbf{I} + \mathbf{W}_i) = (\mathbf{I} + \mathbf{W}_L)(\mathbf{I} + \mathbf{W}_{L-1})...(\mathbf{I} + \mathbf{W}_1)$, then it is a linear shift in parametrization. Using the same derivation and notation, assuming aligned initialization and homogeneous initialization, we can see

$$\frac{\partial \mathcal{L}}{\partial a_k^{(l)}} = \left[ -\lambda_k + (\sigma^2 + \lambda_k)\left(1 + a_k^{(l)}\right)^L \right]\left(1 + a_k^{(l)}\right)^{L-1} \tag{234}$$

Let the overall effective weight be $c_k = \left(1 + a_k^{(l)}\right)^L$, we have the Jacobian:

$$\frac{\partial c_k}{\partial a_k^{(l)}} = L\left(1 + a_k^{(l)}\right)^{L-1}$$

Thus the dynamics of the effective weight is

$$\begin{aligned}
\frac{\partial c_k}{\partial \tau} &= L\left(1 + a_k^{(l)}\right)^{L-1}\frac{\partial a_k^{(l)}}{\partial \tau} \\
&= -\eta L\left(1 + a_k^{(l)}\right)^{L-1}\nabla_{a_k^{(l)}}\mathcal{L} \\
&= -\eta L\left(1 + a_k^{(l)}\right)^{L-1}\left[ -\lambda_k + (\sigma^2 + \lambda_k)\left(1 + a_k^{(l)}\right)^L \right]\left(1 + a_k^{(l)}\right)^{L-1} \\
&= -\eta L\left[ -\lambda_k + (\sigma^2 + \lambda_k)\left(1 + a_k^{(l)}\right)^L \right]\left(1 + a_k^{(l)}\right)^{2L-2} \\
&= -\eta L\left[ -\lambda_k + (\sigma^2 + \lambda_k)c_k \right]c_k^{2 - \frac{2}{L}}
\end{aligned} \tag{235}$$

We recover the same dynamics as the normal deep linear network:

$$\frac{\partial c_k}{\partial \tau} = \eta L[\lambda_k - (\sigma^2 + \lambda_k)c_k]c_k^{2 - \frac{2}{L}} \tag{236}$$

Here the initialization is $c_k(0) = \left(1 + a_k^{(l)}\right)^L$.

Thus in linear networks, residual connections do not change the learning dynamics, but just change (shift) the initialization of weights.

# G Detailed Derivations for Linear Convolutional Network

In this section, we study a linear convolutional denoiser. For simplicity, let's consider the space is 1d, with a 1d convolutional filter $\mathbf{w}$. For simplicity, we ignore the bias term and assume zero-mean $\mu = 0$. The convolutional denoiser is defined as

$$\mathbf{D}(\mathbf{x}; \sigma) = \mathbf{w}_\sigma * \mathbf{x} \tag{237}$$

This convolution $*$ can be equivalently represented as matrix multiplication, where the weight matrix $\mathbf{W}_\sigma$ is a Toeplitz or circulant (when circular boundary condition).

$$\mathbf{D}(\mathbf{x}; \sigma) = \mathbf{W}_\sigma \mathbf{x}$$

In this case, the score learning problem becomes a *circulant or Toeplitz regression* problem, which has been studied before [59, 60, 61]. This problem is similar to finding the best convolutional or deconvolutional kernel to a 1d sequence.

## G.1 General set up

**Cyclic weight matrix and spectral representation**  Let's consider the circular convolution case, where we ignore the boundary effect, and assume the matrix $\mathbf{W}(\sigma)$ is cyclic at any noise scale. Since it's cyclic, they can be diagonalized by discrete Fourier transform (DFT).

Let's diagonalize the weight matrix with the DFT matrix,

$$\mathbf{W}(\sigma) = F\Gamma(\sigma)F^*$$

specifically the DFT matrix is defined as

$$F_{mk} = \frac{1}{\sqrt{N}} \exp\left(-2\pi i \frac{mk}{N}\right), \quad m, k = 0, 1, 2, \ldots, N-1$$

## G.2 General analysis of sampling dynamics

Note that, if we assume weight matrix at every noise scale is circulant, then we always can solve the sampling dynamics on the Fourier basis, mode by mode. At its core, this is because all circulant matrices commute.

$$\frac{d}{d\sigma}\mathbf{x} = -\frac{F\Gamma(\sigma)F^*\mathbf{x} - \mathbf{x}}{\sigma}$$

$$F^* \frac{d}{d\sigma}\mathbf{x} = -\frac{\Gamma(\sigma)F^*\mathbf{x} - F^*\mathbf{x}}{\sigma}$$

$$= -\frac{\Gamma(\sigma) - I}{\sigma}F^*\mathbf{x}$$

We can perform integration mode by mode. Let $c_k = (F^*\mathbf{x})_k$, $\mathbf{x} = F\mathbf{c}$

$$\frac{d}{d\sigma}c_k = -\frac{\gamma_k(\sigma) - 1}{\sigma}c_k$$

Thus, if we know the Fourier parametrization of weights $\Gamma(\sigma)$, we can integrate the sampling of $\mathbf{x}$ in closed form. We will need to solve these Fourier modes $\Gamma(\sigma)$ of the weights during training dynamics. Integrating the ODE, we get

$$\Phi_k(\sigma) = \exp\left(\int -\frac{\gamma_k(\lambda) - 1}{\lambda}d\lambda\right)$$

$$c_k(\sigma_0) = \frac{\Phi_k(\sigma_0)}{\Phi_k(\sigma_T)}c_k(\sigma_T) \tag{238}$$

Then the generated variance will be

$$\tilde{\lambda}_k = \sigma_T^2 \left(\frac{\Phi_k(\sigma_0)}{\Phi_k(\sigma_T)}\right)^2 \tag{239}$$

$$\tilde{\Sigma} = F \operatorname{diag}(\tilde{\lambda}_k) F^* \tag{240}$$

Because of this we can easily prove Proposition 5.2

*Proof.* Generated distributions from linear denoisers are Gaussian distributions, with covariance $\tilde{\Sigma}$. Since the generated covariance is diagonalized by discrete Fourier transform $F$, $\tilde{\Sigma}$ is circulant i.e. translation invariant. Given that the generated $\mathbf{x}$ is Gaussian distributed, $\mathbf{x}$ conforms to a stationary Gaussian process. While the integration function $\Phi_k(\sigma)$ determines the covariance kernel of this Gaussian process. $\square$

## G.3 Full width linear convolutional network

### G.3.1 Training dynamics of full width linear convolutional network

**Gradient structure in spectral basis** Using the Fourier representation of the circulant weight

$$\mathbf{W}(\sigma) = F\mathbf{\Gamma}(\sigma)F^*$$

we can derive the gradient to the Fourier parameters $\mathbf{\Gamma}(\sigma)$ which is a diagonal matrix.

Recall that

$$\nabla_{\mathbf{b}}\mathcal{L} = 2\mathbf{b}$$
$$\nabla_{\mathbf{W}}\mathcal{L} = -2\Sigma + 2\mathbf{W}(\sigma^2 I + \Sigma)$$

Then the gradient to the Fourier parametrization reads

$$\langle \nabla_{\mathbf{W}}\mathcal{L}, d\mathbf{W} \rangle$$
$$= \langle \nabla_{\mathbf{W}}\mathcal{L}, F d\mathbf{\Gamma}(\sigma)F^* \rangle$$
$$= \langle F^{-1}\nabla_{\mathbf{W}}\mathcal{L}F^{-1*}, d\mathbf{\Gamma}(\sigma) \rangle$$
$$= \langle F^*\nabla_{\mathbf{W}}\mathcal{L}F, d\mathbf{\Gamma}(\sigma) \rangle$$

Thus

$$\nabla_{\mathbf{\Gamma}}\mathcal{L} = F^*\nabla_{\mathbf{W}}\mathcal{L}F$$
$$= -2F^*\Sigma F + 2F^*\mathbf{W}(\sigma^2 I + \Sigma)F$$
$$= -2F^*\Sigma F + 2F^*F\mathbf{\Gamma}F^*(\sigma^2 I + \Sigma)F$$
$$= -2F^*\Sigma F + 2\mathbf{\Gamma}(\sigma^2 I + F^*\Sigma F)$$

Under circular convolution assumption, $\mathbf{\Gamma}$ is a diagonal matrix, $\mathbf{\Gamma}_{ij} = \delta_{ij}\gamma_i$ then

$$\frac{\partial \mathcal{L}}{\partial \gamma_i} = [\nabla_{\mathbf{\Gamma}}\mathcal{L}]_{ii}$$
$$= -2[F^*\Sigma F]_{ii} + 2\big[\mathbf{\Gamma}(\sigma^2 I + F^*\Sigma F)\big]_{ii}$$
$$= -2[F^*\Sigma F]_{ii} + 2\big[\delta_{ij}\gamma_i(\sigma^2 I + F^*\Sigma F)\big]_{ii}$$
$$= -2[F^*\Sigma F]_{ii} + 2\gamma_i\big(\sigma^2 + [F^*\Sigma F]_{ii}\big)$$

The key entity is the $F^*\Sigma F$ matrix, let's define it as $\tilde{\Sigma}$

**Interpretation of $F^*\Sigma F$**

- We can regard it as covariance matrix in the spectral basis, i.e. covariance matrix of the complex variable $F^*\bar{\mathbf{x}} \in \mathbb{C}^N$.

$$\Sigma = \text{cov}(\mathbf{x}) = \mathbb{E}[\bar{\mathbf{x}}\bar{\mathbf{x}}^T]$$
$$F^*\Sigma F = F^*\mathbb{E}[\bar{\mathbf{x}}\bar{\mathbf{x}}^T]F$$
$$= \mathbb{E}[F^*\bar{\mathbf{x}}\bar{\mathbf{x}}^T F]$$
$$= \mathbb{E}[F^*\bar{\mathbf{x}}\bar{\mathbf{x}}^T F^T]$$
$$= \mathbb{E}[(F^*\bar{\mathbf{x}})(F^*\bar{\mathbf{x}})^\dagger]$$
$$= \text{cov}(F^*\mathbf{x})$$

- Diagonal values tell us about the power spectrum of the data.

$$[F^*\Sigma F]_{ii} = \text{Var}([F^*\bar{\mathbf{x}}]_i)$$

$$F_{jk} = \frac{1}{\sqrt{N}}e^{-2\pi i\, jk/N}$$

$$[F^*\Sigma F]_{jk} = \sum_{mn} F_{jm}^* \Sigma_{mn} F_{nk}$$

$$= \frac{1}{N}\sum_{mn}\Sigma_{mn}e^{+2\pi i\, jm/N}e^{-2\pi i\, nk/N}$$

$$= \frac{1}{N}\sum_{mn}\Sigma_{mn}e^{2\pi i\,\frac{jm-kn}{N}}$$

$$[F^*\Sigma F]_{jj} = \frac{1}{N}\sum_{mn}\Sigma_{mn}e^{2\pi i\,\frac{j(m-n)}{N}}$$

**Gradient flow on spectral parameter**    Given the gradient structure,

$$\frac{\partial\mathcal{L}}{\partial\gamma_k} = -2[F^*\Sigma F]_{kk} + 2\gamma_k\big(\sigma^2 + [F^*\Sigma F]_{kk}\big)$$

$$= -2\tilde{\Sigma}_{kk} + 2\gamma_k\big(\sigma^2 + \tilde{\Sigma}_{kk}\big)$$

if we *directly perform gradient descent* on the $\gamma_k$ variable, we have

$$\frac{d}{d\tau}\gamma_k = -\eta\frac{\partial\mathcal{L}}{\partial\gamma_k}$$

$$\frac{d}{d\tau}\gamma_k = -\eta\Big[-2\tilde{\Sigma}_{kk} + 2\gamma_k\big(\sigma^2 + \tilde{\Sigma}_{kk}\big)\Big]$$

$$= 2\eta\Big[\tilde{\Sigma}_{kk} - \gamma_k\big(\sigma^2 + \tilde{\Sigma}_{kk}\big)\Big]$$

**Fixed point**    Fixed point of the gradient flow is

$$\gamma_k^* = \frac{\tilde{\Sigma}_{kk}}{\sigma^2 + \tilde{\Sigma}_{kk}} \tag{241}$$

**Learning dynamics of Fourier modes**    This is basically a first order dynamics ODE

$$\gamma_k(\tau) = \frac{\tilde{\Sigma}_{kk}}{\sigma^2 + \tilde{\Sigma}_{kk}} + \Big(\gamma_k(0) - \frac{\tilde{\Sigma}_{kk}}{\sigma^2 + \tilde{\Sigma}_{kk}}\Big)e^{-2\eta(\sigma^2+\tilde{\Sigma}_{kk})\tau}$$

$$= \gamma_k^* + \big(\gamma_k(0) - \gamma_k^*\big)e^{-2\eta(\sigma^2+\tilde{\Sigma}_{kk})\tau} \tag{242}$$

**Remarks**

- This is exactly the same story as the one layer linear case, where $\sigma^2 + \tilde{\Sigma}_{kk}$ determines the convergence speed per mode.

- Spectral modes $\tilde{\Sigma}_{kk}$ with higher power will converge faster, which usually corresponds to the lower spatial frequency modes.

- Higher noise level will converge faster

**Learning dynamics of weight entry as rotated Fourier mode dynamics**

**Lemma G.1** (Convolution Spectrum-entry relation). *The relationship between spectral and entry parametrization of filter weights is*

$$w_\ell = \frac{1}{N}\sum_{k=0}^{N-1}e^{2\pi i\,\frac{k\ell}{N}}\gamma_k, \qquad \ell = 0,\dots,N-1 \tag{243}$$

*Proof.* **Derivation** Using our previous convention

$$\mathbf{W}_{ij} = w_{j-i}$$

Let the vector of the first row in $\mathbf{W}$ be

$$\mathbf{W}_{0k} = w_k$$

Let the vector parameter be

$$\mathbf{w} = [w_0, w_1, ..., w_{N-1}]^T$$
$$= (\mathbf{W}_{0k})^T$$

Then the connection between $\gamma_k$ and weights $w_\ell$ are the following

$$\mathbf{W} = F\Gamma F^*$$

$$\mathbf{W}_{jk} = \sum_{m,n} F_{jm}\Gamma_{mn}F_{nk}^*$$
$$= \sum_{m} \gamma_m F_{jm}F_{mk}^*$$
$$= \sum_{m} \gamma_m F_{jm}F_{mk}^*$$
$$= \frac{1}{N} \sum_{m} \gamma_m e^{-2\pi i\, jm/N} e^{+2\pi i\, mk/N}$$
$$= \frac{1}{N} \sum_{m} \gamma_m e^{2\pi i \frac{(k-j)m}{N}}$$

Thus

$$w_k = \mathbf{W}_{j,j+k}$$
$$= \frac{1}{N} \sum_{m} e^{+2\pi i \frac{km}{N}} \gamma_m$$
$$= \frac{1}{\sqrt{N}} \sum_{m} F_{km}^* \gamma_m$$

$\square$

In vector notation, we have

$$\mathbf{w} = \frac{1}{\sqrt{N}} F^* \boldsymbol{\gamma}$$
$$\boldsymbol{\gamma} = \sqrt{N} F \mathbf{w}$$

Thus, they are related by a $\sqrt{N}$ scaling and a unitary transform.

Similarly, the gradient to $\gamma$ and that of $\mathbf{w}$ has following relation

$$\nabla_{\mathbf{w}}\mathcal{L} = \sqrt{N} F^{-T} \nabla_{\boldsymbol{\gamma}}\mathcal{L}$$
$$= \sqrt{N} F^* \nabla_{\boldsymbol{\gamma}}\mathcal{L}$$

*Proof.* **Derivation**

$$\langle \nabla_{\boldsymbol{\gamma}}\mathcal{L}, d\boldsymbol{\gamma} \rangle$$
$$= \langle \nabla_{\boldsymbol{\gamma}}\mathcal{L}, \sqrt{N} F d\mathbf{w} \rangle$$
$$= \langle \sqrt{N} F^{-T} \nabla_{\boldsymbol{\gamma}}\mathcal{L}, d\mathbf{w} \rangle$$
$$= \langle \nabla_{\mathbf{w}}\mathcal{L}, d\mathbf{w} \rangle$$

$\square$

If we let the optimization parameter be the filters $\mathbf{w}$, then we have gradient flow in $\mathbf{w}$ space

$$\frac{d}{d\tau}\mathbf{w} = -\eta\nabla_{\mathbf{w}}\mathcal{L}$$

The corresponding gradient flow of $\gamma$ is

$$\frac{d}{d\tau}\mathbf{w} = -\eta\nabla_{\mathbf{w}}\mathcal{L}$$

$$\frac{d}{d\tau}\frac{1}{\sqrt{N}}F^*\gamma = -\eta\sqrt{N}F^*\nabla_{\gamma}\mathcal{L}$$

$$\frac{d}{d\tau}\gamma = -\eta N\nabla_{\gamma}\mathcal{L}$$

So gradient flow in $\mathbf{w}$ space is equivalent to gradient flow in $\gamma$, but scaling learning rate by $\eta \to N\eta$!

Conversely if we treat $\gamma$ as optimization parameter and use gradient flow, then it's equivalent to scale learning rate $\eta \to \eta/N$

$$\frac{d}{d\tau}\gamma = -\eta\nabla_{\gamma}\mathcal{L}$$

$$\frac{d}{d\tau}\sqrt{N}F\mathbf{w} = -\eta\frac{1}{\sqrt{N}}F\nabla_{\mathbf{w}}\mathcal{L}$$

$$\frac{d}{d\tau}\mathbf{w} = -\frac{\eta}{N}\nabla_{\mathbf{w}}\mathcal{L}$$

*Remark* G.2. • Thus we can solve gradient flow in any representation, but translating to solution of the other variable just by rescaling the learning rate.

• This trick only works when the kernel width is $N$, or the spectral parameter will be constrained in a subspace, which will alter its dynamics.

• Direct spectral parametrization without any constraint is basically equivalent to the entry wise parametrization. It does not provide extra inductive bias.

**Corollary G.3.** *For linear circular convolutional denoiser,* $\mathbf{D}(\mathbf{x}, \sigma) = \mathbf{w} * \mathbf{x}$, *the solution to gradient flow on filter weight* $\mathbf{w}$ *is* $\mathbf{w}(\tau, \sigma) = \frac{1}{\sqrt{N}}F^*\gamma(\tau, \sigma)$, *where the spectral parameter of $k$-th Fourier mode evolves as*

$$\gamma_k(\tau, \sigma) = \frac{\tilde{\Sigma}_{kk}}{\sigma^2 + \tilde{\Sigma}_{kk}} + \left(\gamma_k(0, \sigma) - \frac{\tilde{\Sigma}_{kk}}{\sigma^2 + \tilde{\Sigma}_{kk}}\right)e^{-2N\eta(\sigma^2 + \tilde{\Sigma}_{kk})\tau}$$

$$= \gamma_k^*(\sigma) + \left(\gamma_k(0, \sigma) - \gamma_k^*(\sigma)\right)e^{-2N\eta(\sigma^2 + \tilde{\Sigma}_{kk})\tau} \tag{244}$$

*where* $\gamma_k^*(\sigma) = \frac{\tilde{\Sigma}_{kk}}{\sigma^2 + \tilde{\Sigma}_{kk}}$, *and* $\tilde{\Sigma}_{kk}$ *is the variance of Fourier mode.*

### G.3.2 Sampling dynamics of full width linear convolutional network

Let's project the sampling equation on Fourier modes,

$$F^*\frac{d}{d\sigma}\mathbf{x} = -\frac{\Gamma(\sigma) - I}{\sigma}F^*\mathbf{x}$$

Let $c_k = (F^*\mathbf{x})_k$, $\mathbf{x} = F\mathbf{c}$,

$$\frac{d}{d\sigma}c_k = -\frac{\gamma_k(\sigma) - 1}{\sigma}c_k$$

The variance amplification factor is expressed through the integral,

$$\Phi_k(\sigma) = \exp\left(\int -\frac{\gamma_k(\lambda) - 1}{\lambda}d\lambda\right)$$

**Generated distribution after convergence** For the converged denoiser we have $\gamma_k^* = \frac{\tilde{\Sigma}_{kk}}{\sigma^2 + \tilde{\Sigma}_{kk}}$

$$
\begin{aligned}
\Phi_k(\sigma) &= \exp\Big(\int_\epsilon^\sigma -\frac{\gamma_k(\lambda) - 1}{\lambda} d\lambda\Big) \\
&= \exp\Big(\int_\epsilon^\sigma -\frac{\frac{\tilde{\Sigma}_{kk}}{\lambda^2 + \tilde{\Sigma}_{kk}} - 1}{\lambda} d\lambda\Big) \\
&= \exp\Big(\int_\epsilon^\sigma \frac{\lambda}{\lambda^2 + \tilde{\Sigma}_{kk}} d\lambda\Big) \\
&= \exp\Big(\frac{1}{2} \ln(\lambda^2 + \tilde{\Sigma}_{kk})|_{\lambda=\epsilon}^{\lambda=\sigma}\Big) \\
&= \sqrt{\frac{\sigma^2 + \tilde{\Sigma}_{kk}}{\epsilon^2 + \tilde{\Sigma}_{kk}}}
\end{aligned}
$$

Variance of the learned distribution

$$
\tilde{\lambda}_k = \sigma_T^2 \Big(\frac{\Phi_k(\sigma_0)}{\Phi_k(\sigma_T)}\Big)^2 = \sigma_T^2 \frac{\sigma_0^2 + \tilde{\Sigma}_{kk}}{\sigma_T^2 + \tilde{\Sigma}_{kk}} \approx \tilde{\Sigma}_{kk}
$$

The generated covariance will be

$$
\begin{aligned}
\tilde{\Sigma} &= F\mathrm{diag}(\tilde{\lambda}_k)F^* \\
&\approx F\mathrm{diag}(\tilde{\Sigma}_{kk})F^*
\end{aligned}
$$

*Remark* G.4.     • Basically it's the same story as the one-layer linear network, the major difference is, instead of evolution along the eigenbasis of data covariance (PCs), the parametrization of convolutional network enforces the learning dynamics to align with the Fourier basis regardless of the original PC of the data. It treats the data as if it's stationary / translation invariant.

- The generated data will be independent along each Fourier mode, just like the fully connected case, the generated data is independent along each eigen-mode.

- This amounts to decorrelating the original covariance in the spectral domain, making it translation invariant / circulant.

**Generated distribution during training** Using the solution to gradient flow of **spectral parametrization**, for each Fourier mode,

$$
\gamma_k(\tau; \sigma) = \gamma_k^* + \big(\gamma_k(0; \sigma) - \gamma_k^*\big)e^{-2\eta(\sigma^2 + \tilde{\Sigma}_{kk})\tau}
$$

We can solve the sampling ODE during training. Assume the initial value of each Fourier mode is the same across noise scale $\gamma_k(0; \sigma) = \gamma_k(0), \forall \sigma$. We can write down the generated variance through training time as, [3]

$$
\begin{aligned}
\Phi_k(\sigma) &= \exp\Big(\int^\sigma -\frac{\gamma_k(\lambda) - 1}{\lambda} d\lambda\Big) \\
&= \exp\Big(\int^\sigma -\frac{\frac{\tilde{\Sigma}_{kk}}{\lambda^2 + \tilde{\Sigma}_{kk}} + \big(\gamma_k(0) - \frac{\tilde{\Sigma}_{kk}}{\lambda^2 + \tilde{\Sigma}_{kk}}\big)e^{-2\eta(\lambda^2 + \tilde{\Sigma}_{kk})\tau} - 1}{\lambda} d\lambda\Big) \\
&= \sqrt{\tilde{\Sigma}_{kk} + \sigma^2} \exp\Big(\frac{1}{2}\Big[(1 - \gamma_k(0))\,\mathrm{Ei}\big(-2\eta\tau\sigma^2\big)e^{-2\eta\tau\tilde{\Sigma}_{kk}} - \mathrm{Ei}\big(-2\eta\tau\big(\sigma^2 + \tilde{\Sigma}_{kk}\big)\big)\Big]\Big)
\end{aligned}
$$

Using the solution to gradient flow of **filter weights parametrization**, we effectively amplify the learning rate $\eta \to N\eta$ in the expression.

$$
\gamma_k(\tau; \sigma) = \gamma_k^* + \big(\gamma_k(0; \sigma) - \gamma_k^*\big)e^{-2N\eta(\sigma^2 + \tilde{\Sigma}_{kk})\tau}
$$

---

[3]One thing to note is that it's highly likely that the initial value of $\gamma_k(0; \sigma)$ can differ for each Fourier mode, i.e. different Fourier modes are initialized at different weights due to the filter initialization.

The generated distribution has variance $\check{\Sigma} = F\check{\Lambda}F^*$, where $\check{\Lambda}_{kk} = \sigma_T^2 \left(\frac{\Phi_k(\sigma_0)}{\Phi_k(\sigma_T)}\right)^2$ controls variance along Fourier modes.

$$\Phi_k(\sigma) = \sqrt{\tilde{\Sigma}_{kk} + \sigma^2} \exp\left(\frac{1}{2}\left[(1-\gamma_k(0))\,\text{Ei}\left(-2N\eta\tau\sigma^2\right)e^{-2N\eta\tau\tilde{\Sigma}_{kk}} - \text{Ei}\left(-2N\eta\tau\left(\sigma^2 + \tilde{\Sigma}_{kk}\right)\right)\right]\right)$$
$$(245)$$

### G.4  Local patch linear convolutional network

Now, let's consider a local filter $\mathbf{w}$.

**Spectral representation and spatial locality constraint**   The major difference is that if we have a locality constraint in $\mathbf{w}$, the corresponding constraint in $\gamma$ will be constrained in a lower dimensional space

$$w_\ell = \frac{1}{N} \sum_{k=0}^{N-1} e^{2\pi i \frac{kl}{N}} \gamma_k$$

$$\gamma = \sqrt{N} F \mathbf{w}$$

$$\gamma_l = \sum_{k=0}^{N-1} e^{-2\pi i \frac{kl}{N}} w_k$$

But if our $w_k = 0$, $\forall k \geq K$, then the spectral moment just sums over $K$

$$\gamma_l = \sum_{k=0}^{K-1} e^{-2\pi i \frac{kl}{N}} w_k$$

**Specific case: kernel 3 convolution**   Consider the case where the kernel size is 3, so only $w_1, w_0, w_{-1} \neq 0$. Then the spectrum can only have a DC and a cosine and sine component

$$\gamma_l = \sum_{k \in \{-1,0,1\}} e^{-2\pi i \frac{kl}{N}} w_k$$

$$= w_0 + w_1 e^{-2\pi i \frac{l}{N}} + w_{-1} e^{2\pi i \frac{l}{N}}$$

$$= w_0 + (w_1 + w_{-1}) \cos\left(\frac{2\pi l}{N}\right) + i(w_{-1} - w_1) \sin\left(\frac{2\pi l}{N}\right)$$

Basically, the small kernel size mandates that the spectral view of it $\gamma$ cannot vary very fast! Thus the spectral representation has to be very smooth, just sine and cosine waves.

### G.4.1  Training dynamics of patch linear convolutional net

To study the training dynamics of the linear local convolutional network, it's easier to directly work with the entries of the filters.

Consider the Toeplitz weight matrix (general boundary condition),

$$\mathbf{W} = \begin{pmatrix} w_0 & w_1 & w_2 & \cdots & w_{d-1} \\ w_{-1} & w_0 & w_1 & \cdots & w_{d-2} \\ w_{-2} & w_{-1} & w_0 & \cdots & w_{d-3} \\ \vdots & \vdots & \vdots & \ddots & \vdots \\ w_{-d+1} & w_{-d+2} & w_{-d+3} & \cdots & w_0 \end{pmatrix}.$$

**Shift matrix formulation**   Generally, a Toeplitz matrix can be written as

$$\mathbf{W} = \sum_{k=0}^{d-1} w_k P^k + \sum_{k=1}^{d-1} w_{-k} (P^T)^k$$

where the (non-circulant) shift matrix is $P$,

$$P = \begin{bmatrix} 0 & 1 & 0 & \cdots & 0 \\ 0 & 0 & 1 & \cdots & 0 \\ \vdots & \vdots & \vdots & \ddots & \vdots \\ 0 & 0 & 0 & \cdots & 1 \\ 0 & 0 & 0 & \cdots & 0 \end{bmatrix}.$$

This compactly expresses the weight matrix as a weighted sum of shift matrix powers, enabling us to write down the gradient efficiently.

**Remark:**

- For circulant shift matrix $P^T = P^{-1}$. $P^T P = 0$ exactly.
- For non-circulant shift matrix, approximately $P^T = P^{-1}$ but there will be some issues in the boundary condition.
- Trace with it also allows us to elegantly extract entries from a matrix.
- Multiplying this shift matrix allows us to shift the matrix

This decomposition of weights allows us to write down the loss and the gradient.

For $k \geq 0$

$$
\begin{aligned}
\frac{\partial \mathcal{L}}{\partial w_k} &= \sum_{j-i=k} G_{ij} \\
&= \sum_{j-i=k} \frac{\partial \mathcal{L}}{\partial \mathbf{W}_{ij}} \\
&= \langle P^k, G \rangle \\
&= \text{Tr}[(P^k)^T G] \\
&= \text{Tr}[(P^k)^T \frac{\partial \mathcal{L}}{\partial \mathbf{W}}]
\end{aligned}
$$

for the other $k$,

$$
\begin{aligned}
\frac{\partial \mathcal{L}}{\partial w_{-k}} &= \langle (P^T)^k, G \rangle \\
&= \text{Tr}[P^k G] \\
&= \text{Tr}[P^k \frac{\partial \mathcal{L}}{\partial \mathbf{W}}]
\end{aligned}
$$

Using this formulation, we can express the gradient to each entry.

$$
\begin{aligned}
\frac{\partial \mathcal{L}}{\partial w_k} &= 2\text{Tr}[-(P^k)^T \Sigma + (P^k)^T W(\sigma^2 I + \Sigma)] \\
&= -2\text{Tr}[(P^k)^T \Sigma] + 2\text{Tr}[(P^k)^T W(\sigma^2 I + \Sigma)] \\
&= -2\text{Tr}[(P^k)^T \Sigma] + 2\text{Tr}[(P^k)^T \Big( \sum_{m=0}^{d-1} w_m P^m + \sum_{m=1}^{d-1} w_{-m}(P^T)^m \Big)(\sigma^2 I + \Sigma)] \\
&= -2\text{Tr}[(P^k)^T \Sigma] + 2\text{Tr}[\Big( \sum_{m=0}^{d-1} w_m (P^k)^T P^m + \sum_{m=1}^{d-1} w_{-m}(P^T)^{k+m} \Big)(\sigma^2 I + \Sigma)] \\
&= -2\text{Tr}[(P^k)^T \Sigma] + 2 \sum_{m=0}^{d-1} w_m \text{Tr}[(P^k)^T P^m (\sigma^2 I + \Sigma)] + 2 \sum_{m=1}^{d-1} w_{-m} \text{Tr}[(P^T)^{k+m}(\sigma^2 I + \Sigma)]
\end{aligned}
$$

Similarly,

$$
\begin{aligned}
\frac{\partial \mathcal{L}}{\partial w_{-k}} &= 2\text{Tr}[-P^k \Sigma + P^k W(\sigma^2 I + \Sigma)] \\
&= -2\text{Tr}[P^k \Sigma] + 2\text{Tr}[P^k W(\sigma^2 I + \Sigma)] \\
&= -2\text{Tr}[P^k \Sigma] + 2\text{Tr}[P^k \Big( \sum_{m=0}^{d-1} w_m P^m + \sum_{m=1}^{d-1} w_{-m}(P^T)^m \Big)(\sigma^2 I + \Sigma)] \\
&= -2\text{Tr}[P^k \Sigma] + 2\text{Tr}[\Big( \sum_{m=0}^{d-1} w_m P^{k+m} + \sum_{m=1}^{d-1} w_{-m} P^k (P^T)^m \Big)(\sigma^2 I + \Sigma)] \\
&= -2\text{Tr}[P^k \Sigma] + 2 \sum_{m=0}^{d-1} w_m \text{Tr}[P^{k+m}(\sigma^2 I + \Sigma)] + 2 \sum_{m=1}^{d-1} w_{-m} \text{Tr}[P^k (P^T)^m (\sigma^2 I + \Sigma)]
\end{aligned}
$$

We denote the general k-shift operator (non-circulant boundary condition)

$$S(k) = \begin{cases} P^k, & k \geq 0, \\ (P^T)^{-k}, & k < 0, \end{cases}$$

$$S(k)^T = (P^k)^T = (P^T)^k = S(-k)$$

Define a special function with two indices

$$\mathcal{T}[k,m] = \text{Tr}[S(m)(\sigma^2 I + \Sigma)S(k)^T]$$
$$= \sigma^2 \text{Tr}[S(m)S(k)^T] + \text{Tr}[S(m) \Sigma S(k)^T]$$

The first term is diagonal, the second term is not. Non-diagonal terms all come from the second term.

Note that this special function is symmetric, which is understandable, since it will be functioning as the new covariance matrix.

$$\mathcal{T}[m,k] = \text{Tr}[S(m)(\sigma^2 I + \Sigma)S(k)^T]$$
$$= \text{Tr}[S(k)(\sigma^2 I + \Sigma)S(m)^T]$$
$$= \mathcal{T}[k,m]$$

Then the gradient can be written as this simplified equation, of a similar structure as before.

$$\frac{\partial \mathcal{L}}{\partial w_k} = -2\text{Tr}[S(k)^T \Sigma] + 2 \sum_{m=1-d}^{d-1} \text{Tr}[S(m)(\sigma^2 I + \Sigma)S(k)^T]w_m$$

$$= -2\text{Tr}[S(k)^T \Sigma] + 2 \sum_{m=1-d}^{d-1} \mathcal{T}[k,m]w_m$$

$$= -2\text{Tr}[S(k)\Sigma] + 2 \sum_{m=1-d}^{d-1} \mathcal{T}[k,m]w_m \qquad (246)$$

Thus to study the learning dynamics of convolutional linear models, the object that needs to be focused on is $\mathcal{T}[k,m]$, the spectrum of it. Thus, the learning dynamics of $w$ will be first along higher eigenvalues of $\mathcal{T}$, and then lower eigen ones.

**Interpretation of $\mathcal{T}$ matrix**   Note that the $\mathcal{T}[k,m]$ matrix can be regarded as the spatially averaged version of the covariance matrix, especially by averaging local cross covariances of pixels at distance $k, m$.

**Optimal solution**   The fixed point of the equation is the following linear equation

$$\sum_m \mathcal{T}[k,m]w_m = \text{Tr}[S(k)\Sigma]$$

Let the vector be $R_k = \text{Tr}[S(k)\Sigma]$, $k = \{1-d, ...d-1\}$

$$w^* = \mathcal{T}^{-1}R \qquad (247)$$

Here we successfully derived the matrix equation for gradient flow for weights $w$.

**Cyclic case (circular convolution)**   Now consider the cyclic shift matrix

$$P = \begin{bmatrix} 0 & 1 & 0 & \cdots & 0 \\ 0 & 0 & 1 & \cdots & 0 \\ \vdots & \vdots & \vdots & \ddots & \vdots \\ 0 & 0 & 0 & \cdots & 1 \\ 1 & 0 & 0 & \cdots & 0 \end{bmatrix}.$$

then $P^T = P^{-1}$, so we have

$$\mathcal{T}[k,m] = \text{Tr}[S(m)(\sigma^2 I + \Sigma)S(k)^T]$$
$$= \text{Tr}[P^m(\sigma^2 I + \Sigma)P^{-k}]$$
$$= \text{Tr}[P^{m-k}(\sigma^2 I + \Sigma)]$$
$$= \sigma^2\text{Tr}[P^{m-k}] + \text{Tr}[P^{m-k}\Sigma]$$

Note that

$$\text{Tr}[P^{m-k}] = N\delta_{mk}$$

So

$$\mathcal{T}[k,m] = N\sigma^2\delta_{mk} + \text{Tr}[P^{m-k}\Sigma]$$
$$= N\left(\sigma^2\delta_{mk} + \frac{1}{N}\text{Tr}[P^{m-k}\Sigma]\right)$$

Note that the second term is the shift-averaged version of the covariance matrix

$$\frac{1}{N}\text{Tr}\left[P^{m-k}\Sigma\right]$$
$$= \frac{1}{N}\sum_{i=1}^N \Sigma_{i,i+(m-k)\bmod N}$$

Let's define a new averaged cross-covariance vector

$$\chi[k] = \frac{1}{N}\text{Tr}[P^k\Sigma]$$
$$= \frac{1}{N}\sum_{i=1}^N \Sigma_{i,i+k\bmod N}$$

Note $\chi[k] = \chi[-k]$

$$R_k = N\chi[k]$$
$$\mathcal{T}[k,m] = N\left(\sigma^2\delta_{mk} + \chi[m-k]\right)$$

We want to solve the linear system

$$\sum_m \mathcal{T}[k,m]w_m - R_k = 0$$

Note that $\mathcal{T}[k,m]$ is a Toeplitz matrix.

In the circulant case the optimal solution satisfies,

$$\sum_{m=-K}^K (N\sigma^2\delta_{km} + R_{k-m})w_m^* = R_k$$

where $R_k$ is the averaged version of the covariance matrix. So again it's a circulant regression with a Ridge-like penalty.

$$R_k = \text{Tr}[P^k\Sigma]$$

If we let $\check{\Sigma}_{k,m} = \frac{1}{N}R_{k-m} \in \mathbb{R}^{2K+1\times 2K+1}$ be the Toeplitz patch covariance matrix, then we can write $R_k$ as the center column of it

$$\sum_{m=-K}^K (N\sigma^2\delta_{km} + N\check{\Sigma}_{k,m})w_m^* = N\check{\Sigma}_{k,0}$$

The solution can be written in vector form as

$$(N\sigma^2 I + N\check{\Sigma})w^* = N\check{\Sigma}_{:,0}$$
$$w^* = (N\sigma^2 I + N\check{\Sigma})^{-1}N\check{\Sigma}_{:,0}$$
$$= \left((\sigma^2 I + \check{\Sigma})^{-1}\check{\Sigma}\right)_{:,0}$$

Thus the denoiser equals the central column from the Gaussian solution to the patch covariance.

The gradient flow dynamics is

$$\frac{\partial \mathcal{L}}{\partial w_k} = -2R_k + 2 \sum_{m=1-d}^{d-1} (N\sigma^2 \delta_{km} + R_{k-m}) w_m \tag{248}$$

We can write the flow dynamics in vector form

$$\begin{aligned}
\frac{d\mathbf{w}}{d\tau} &= -\eta \frac{\partial \mathcal{L}}{\partial \mathbf{w}} \\
&= 2\eta \left( N\check{\Sigma}_{:,0} - N(\sigma^2 I + \check{\Sigma})\mathbf{w} \right) \\
&= 2\eta N \left( \check{\Sigma}_{:,0} - (\sigma^2 I + \check{\Sigma})\mathbf{w} \right)
\end{aligned}$$

Solution will be

$$\mathbf{w} = \mathbf{w}^* + \exp\left( -2\eta N(\sigma^2 I + \check{\Sigma}) \right) (\mathbf{w}(0) - \mathbf{w}^*) \tag{249}$$

with $\mathbf{w}^* = \left( (\sigma^2 I + \check{\Sigma})^{-1} \check{\Sigma} \right)_{:,0}$.

### G.4.2 Sampling dynamics of patch linear convolutional net

Consider a kernel of width $2K + 1$,

$$\begin{aligned}
\gamma_l &= \sum_{k \in -K:K} e^{-2\pi i \frac{kl}{N}} w_k \\
&= w_0 + \sum_{k=1}^{K} w_k e^{-2\pi i \frac{kl}{N}} + w_{-k} e^{2\pi i \frac{kl}{N}} \\
&= w_0 + \sum_{k=1}^{K} (w_k + w_{-k}) \cos\left(\frac{2\pi kl}{N}\right) + i(w_{-k} - w_k) \sin\left(\frac{2\pi kl}{N}\right)
\end{aligned}$$

For symmetric filter weights $w_k = w_{-k}$, we have

$$\gamma_l = w_0 + \sum_{k=1}^{K} 2w_k \cos\left(\frac{2\pi kl}{N}\right) \tag{250}$$

During sampling we have

$$\frac{d}{d\sigma} c_k = -\frac{\gamma_k(\sigma) - 1}{\sigma} c_k - b_k(\sigma) \tag{251}$$

The key integral governing the variance is

$$\begin{aligned}
\Phi_k(\sigma) &= \exp\left( \int^{\sigma} -\frac{\gamma_k(\lambda) - 1}{\lambda} d\lambda \right) \tag{252} \\
&= \exp\left( \int^{\sigma} -\frac{w_0(\lambda) + \sum_{l=1}^{K} 2w_l(\lambda) \cos\left(\frac{2\pi lk}{N}\right) - 1}{\lambda} d\lambda \right)
\end{aligned}$$

Thus we can see it integrates these sinusoidal modulations in the frequency domain.

### G.5 Appendix: Useful math

**Discrete Fourier Transformation (DFT) matrix**

$$F_{jk} = \frac{1}{\sqrt{N}} e^{-2\pi i \, jk/N} \tag{253}$$

Property

Symmetry
$$F = F^T$$

Conjugacy
$$FF^* = I$$

$$F_{jk}F_{km}^* = \frac{1}{N}\sum_{k}^{N} e^{-2\pi i\, jk/N} e^{+2\pi i\, km/N}$$

$$= \frac{1}{N}\sum_{k}^{N} e^{-2\pi i\, \frac{k(j-m)}{N}}$$

$$= \delta_{jm}$$

**Properties of Circulant and DFT**   Consider matrices of the following form
$$M = F\Lambda F^* \tag{254}$$

Explicitly, the matrix reads
$$M_{jk} = \sum_{m} F_{jm}\lambda_m F_{mk}^*$$

$$= \sum_{m} \lambda_m \frac{1}{\sqrt{N}}\exp\left(-2\pi i\frac{jm}{N}\right)\frac{1}{\sqrt{N}}\exp\left(2\pi i\frac{mk}{N}\right)$$

$$= \frac{1}{N}\sum_{m} \lambda_m \exp\left(-2\pi i\frac{jm}{N} + 2\pi i\frac{mk}{N}\right)$$

$$= \frac{1}{N}\sum_{m} \lambda_m \exp\left(-2\pi i\frac{m(j-k)}{N}\right)$$

Thus we see $M_{jk}$ depends only on $j - k$, hence is circulant. Let's define the circulant coefficients $c_{j-k} := M_{j,k}$.

Then we have for $\Delta = 0, 1, ...N - 1$
$$c_\Delta = \frac{1}{N}\sum_{m} \lambda_m \exp\left(-2\pi i\frac{m\Delta}{N}\right) \tag{255}$$

The special case is the "DC" non-oscillating term, which are the diagonal values in $M$
$$c_0 = \frac{1}{N}\sum_{m} \lambda_m$$

the adjacent sub-diagonal values in $M$ are

$$c_1 = \frac{1}{N}\sum_{m} \lambda_m \exp\left(-2\pi i\frac{m}{N}\right)$$

$$c_{N-1} = \frac{1}{N}\sum_{m} \lambda_m \exp\left(-2\pi i\frac{m(N-1)}{N}\right)$$

$$= \frac{1}{N}\sum_{m} \lambda_m \exp\left(+2\pi i\frac{m}{N}\right)$$

More generally $c_k = c_{N-k}^*$ are complex conjugates, or equal in the real case.

**Properties of the $\Lambda$ spectrum**   Since $\Sigma$ is real symmetric, the eigenvalues exhibit mirror, i.e. even symmetry $\lambda_k = \lambda_{N-k}$ for $k = 1, 2, ..., N - 1$

There are one or two standalone eigenvalues: when $N$ is odd, $\lambda_0$ is the zero-th frequency, DC component, which is unpaired.

When $N$ is even, $\lambda_0$ (DC component) and $\lambda_{N//2}$ (Nyquist frequency) are both standing alone, which are both unpaired.

# H  Detailed derivation of Flow Matching model

Consider the objective of flow matching [28], at a certain $t$

$$\mathcal{L} = \mathbb{E}_{\mathbf{x}_0 \sim \mathcal{N}(0,\mathbf{I}),\ \mathbf{x}_1 \sim p_1} \|u(\mathbf{x}_t; t) - (\mathbf{x}_1 - \mathbf{x}_0)\|^2 \tag{256}$$

$$= \mathbb{E}_{\mathbf{x}_0 \sim \mathcal{N}(0,\mathbf{I}),\ \mathbf{x}_1 \sim p_1} \|u((1-t)\mathbf{x}_0 + t\mathbf{x}_1; t) - (\mathbf{x}_1 - \mathbf{x}_0)\|^2 \tag{257}$$

$$\mathbf{x}_t = (1-t)\mathbf{x}_0 + t\mathbf{x}_1 \tag{258}$$

Given linear function approximator of the velocity field,

$$u(\mathbf{x}; t) = \mathbf{W}_t \mathbf{x} + \mathbf{b}_t \tag{259}$$

$$\mathcal{L} = \mathbb{E}_{\mathbf{x}_0 \sim \mathcal{N}(0,\mathbf{I}),\ \mathbf{x}_1 \sim p_1} \|\mathbf{W}_t((1-t)\mathbf{x}_0 + t\mathbf{x}_1) + \mathbf{b}_t - (\mathbf{x}_1 - \mathbf{x}_0)\|^2$$

$$= \mathbb{E}_{\mathbf{x}_0 \sim \mathcal{N}(0,\mathbf{I}),\ \mathbf{x}_1 \sim p_1} \left(\mathbf{W}_t((1-t)\mathbf{x}_0 + t\mathbf{x}_1) + \mathbf{b}_t - (\mathbf{x}_1 - \mathbf{x}_0)\right)^T \left(\mathbf{W}_t((1-t)\mathbf{x}_0 + t\mathbf{x}_1) + \mathbf{b}_t - (\mathbf{x}_1 - \mathbf{x}_0)\right)$$

$$= \mathbb{E}_{\mathbf{x}_0 \sim \mathcal{N}(0,\mathbf{I}),\ \mathbf{x}_1 \sim p_1} \mathrm{Tr}\Big[((1-t)\mathbf{x}_0 + t\mathbf{x}_1)^T \mathbf{W}_t^T \mathbf{W}_t((1-t)\mathbf{x}_0 + t\mathbf{x}_1) + \mathbf{b}^T\mathbf{b} + (\mathbf{x}_1 - \mathbf{x}_0)^T(\mathbf{x}_1 - \mathbf{x}_0)$$

$$- 2(\mathbf{x}_1 - \mathbf{x}_0)^T\mathbf{b}_t - 2(\mathbf{x}_1 - \mathbf{x}_0)^T\mathbf{W}_t((1-t)\mathbf{x}_0 + t\mathbf{x}_1) + 2\mathbf{b}_t^T\mathbf{W}_t((1-t)\mathbf{x}_0 + t\mathbf{x}_1)\Big]$$

$$= \mathbb{E}_{\mathbf{x}_0 \sim \mathcal{N}(0,\mathbf{I}),\ \mathbf{x}_1 \sim p_1} \mathrm{Tr}\Big[\mathbf{W}_t^T \mathbf{W}_t((1-t)\mathbf{x}_0 + t\mathbf{x}_1)((1-t)\mathbf{x}_0 + t\mathbf{x}_1)^T + \mathbf{b}^T\mathbf{b} + (\mathbf{x}_1 - \mathbf{x}_0)^T(\mathbf{x}_1 - \mathbf{x}_0)$$

$$- 2(\mathbf{x}_1 - \mathbf{x}_0)^T\mathbf{b}_t - 2\mathbf{W}_t((1-t)\mathbf{x}_0 + t\mathbf{x}_1)(\mathbf{x}_1 - \mathbf{x}_0)^T + 2\mathbf{b}_t^T\mathbf{W}_t((1-t)\mathbf{x}_0 + t\mathbf{x}_1)\Big]$$

Similar to the diffusion case, it will also depend only on the mean and covariance of $p_1$.

$$\mathbb{E}_{\mathbf{x}_0,\mathbf{x}_1}[(1-t)\mathbf{x}_0 + t\mathbf{x}_1] = t\mu$$

$$\mathbb{E}_{\mathbf{x}_0,\mathbf{x}_1}[\mathbf{x}_1 - \mathbf{x}_0] = \mu$$

$$\mathbb{E}_{\mathbf{x}_0,\mathbf{x}_1}[(\mathbf{x}_1 - \mathbf{x}_0)^T(\mathbf{x}_1 - \mathbf{x}_0)] = \mathbb{E}_{\mathbf{x}_0,\mathbf{x}_1}[\mathbf{x}_1^T\mathbf{x}_1 - 2\mathbf{x}_1^T\mathbf{x}_0 + \mathbf{x}_0^T\mathbf{x}_0]$$

$$= \mathrm{Tr}[\Sigma + \mu\mu^T + \mathbf{I}]$$

$$\mathbb{E}_{\mathbf{x}_0,\mathbf{x}_1}[((1-t)\mathbf{x}_0 + t\mathbf{x}_1)(\mathbf{x}_1 - \mathbf{x}_0)^T] = t(\Sigma + \boldsymbol{\mu}\boldsymbol{\mu}^T) - (1-t)\mathbf{I}$$

$$\mathbb{E}_{\mathbf{x}_0,\mathbf{x}_1}[((1-t)\mathbf{x}_0 + t\mathbf{x}_1)((1-t)\mathbf{x}_0 + t\mathbf{x}_1)^T] = t^2(\Sigma + \boldsymbol{\mu}\boldsymbol{\mu}^T) + (1-t)^2\mathbf{I}$$

Taking full expectation, the average loss reads.

$$\mathcal{L} = \mathrm{Tr}\Big[\mathbf{W}_t^T\mathbf{W}_t\big(t^2(\Sigma + \boldsymbol{\mu}\boldsymbol{\mu}^T) + (1-t)^2\mathbf{I}\big) + \mathbf{b}^T\mathbf{b} + (\Sigma + \mu\mu^T + \mathbf{I})$$

$$- 2\mu^T\mathbf{b}_t - 2\mathbf{W}_t\big(t(\Sigma + \boldsymbol{\mu}\boldsymbol{\mu}^T) - (1-t)\mathbf{I}\big) + 2t\mathbf{b}_t^T\mathbf{W}_t\mu\Big] \tag{260}$$

The gradients with respect to parameters are

$$\nabla_{\mathbf{b}}\mathcal{L} = 2[\mathbf{b} - \mu + t\mathbf{W}\mu] \tag{261}$$

$$\nabla_{\mathbf{W}}\mathcal{L} = 2\Big[\mathbf{W}\big(t^2(\Sigma + \mu\mu^T) + (1-t)^2\mathbf{I}\big) - \big(t(\Sigma + \mu\mu^T) - (1-t)\mathbf{I}\big) + t\mathbf{b}\mu^T\Big], \tag{262}$$

**Simplifying case** $\mu = 0$  Note the special case $\mu = 0$

$$\nabla_{\mathbf{b}}\mathcal{L} = 2\mathbf{b} \tag{263}$$

$$\nabla_{\mathbf{W}}\mathcal{L} = 2\Big[\mathbf{W}\big(t^2\Sigma + (1-t)^2\mathbf{I}\big) - \big(t\Sigma - (1-t)\mathbf{I}\big)\Big], \tag{264}$$

**Optimal solution**  The optimal solution to the full case is

$$\mathbf{b}^* = \mu - t\mathbf{W}^*\mu \tag{265}$$

$$\mathbf{W}^* = \big(t\Sigma - (1-t)\mathbf{I}\big)\big(t^2\Sigma + (1-t)^2\mathbf{I}\big)^{-1} \tag{266}$$

We can represent it on the eigenbasis of $\Sigma$, $[u_1, ...u_d]$

$$\mathbf{W}^* = \sum_k \frac{t\lambda_k - (1-t)}{t^2\lambda_k + (1-t)^2} \mathbf{u}_k \mathbf{u}_k^T \tag{267}$$

**Asymptotics**   Consider the limit $t \to 0$,

$$\mathbf{W}^*_{t \to 0} = -\mathbf{I}$$
$$\mathbf{b}^*_{t \to 0} = \mu$$

Consider the limit $t \to 1$,

$$\mathbf{W}^*_{t \to 1} = \mathbf{I}$$
$$\mathbf{b}^*_{t \to 1} = \mu - \mathbf{W}^*_{t \to 1}\mu = 0$$

## H.1   Solution to the flow matching sampling ODE with optimal linear solution

Solving the sampling ODE of flow matching integrating from 0 to 1, with the linear vector field

$$\frac{d\mathbf{x}}{dt} = u(\mathbf{x}; t) \tag{268}$$

Under the linear solution case

$$\frac{d\mathbf{x}}{dt} = \mathbf{W}^*_t \mathbf{x} + \mathbf{b}^*_t \tag{269}$$

**Simplified zero mean case** $\mu = 0$

$$\frac{d\mathbf{x}}{dt} = \sum_k \frac{t\lambda_k - (1-t)}{t^2\lambda_k + (1-t)^2} \mathbf{u}_k \mathbf{u}_k^T \mathbf{x}$$

Solving the flow-matching sampling ODE mode by mode

$$\mathbf{u}_k^T \frac{d\mathbf{x}}{dt} = \frac{t\lambda_k - (1-t)}{t^2\lambda_k + (1-t)^2} \mathbf{u}_k^T \mathbf{x}$$
$$\frac{dc_k(t)}{dt} = \frac{t\lambda_k - (1-t)}{t^2\lambda_k + (1-t)^2} c_k(t)$$

$$\ln c_k(t) = \frac{1}{2}\ln\left| t^2\lambda_k + (1-t)^2 \right| + C.$$

$$c_k(t) = C\sqrt{t^2\lambda_k + (1-t)^2} \tag{270}$$

$$\frac{c_k(t)}{c_k(0)} = \sqrt{t^2\lambda_k + (1-t)^2}$$

$$\frac{c_k(1)}{c_k(0)} = \sqrt{\lambda_k}$$

This is the correct scaling of $\mathbf{x}$. The sampling trajectory of $\mathbf{x_t}$ reads

$$\mathbf{x}_t = \sum_k c_k(t)\mathbf{u}_k \tag{271}$$
$$= \sum_k \sqrt{t^2\lambda_k + (1-t)^2}\mathbf{u}_k \mathbf{u}_k^T \mathbf{x}_0 \tag{272}$$

Thus, at time $t$ the covariance of the sampled points is

$$\mathbb{E}[\mathbf{x}_t \mathbf{x}_t^T] = \sum_k (t^2\lambda_k + (1-t)^2)\mathbf{u}_k \mathbf{u}_k^T \tag{273}$$

with variance $\tilde{\lambda}_k = t^2\lambda_k + (1-t)^2$ along eigenmode $\mathbf{u}_k$

**General case $\mu \neq 0$**

$$\frac{d\mathbf{x}}{dt} = \mathbf{W}_t^*\mathbf{x} + \mu - t\mathbf{W}_t^*\mu$$

$$= \mu + \sum_k \frac{t\lambda_k - (1-t)}{t^2\lambda_k + (1-t)^2}\mathbf{u}_k\mathbf{u}_k^T(\mathbf{x} - t\mu) \tag{274}$$

It can also be solved mode by mode. Redefine variable $\mathbf{y}_t = \mathbf{x}_t - t\mu$

$$\frac{d\mathbf{y}_t}{dt} = \frac{d\mathbf{x}_t}{dt} - \mu$$

$$= \sum_k \frac{t\lambda_k - (1-t)}{t^2\lambda_k + (1-t)^2}\mathbf{u}_k\mathbf{u}_k^T\mathbf{y}_t$$

Then each mode can be solved accordingly, $c_k(t) = \mathbf{u}_k^T\mathbf{y}_t$

$$\frac{dc_k(t)}{dt} = \frac{t\lambda_k - (1-t)}{t^2\lambda_k + (1-t)^2}c_k(t)$$

Using the same solution as above, we get the full solution for the sampling equation with any $\mu$

$$\mathbf{x}_t = t\mu + \sum_k c_k(t)\mathbf{u}_k$$

$$= t\mu + \sum_k \sqrt{t^2\lambda_k + (1-t)^2}\mathbf{u}_k\mathbf{u}_k^T\mathbf{x}_0 \tag{275}$$

## H.2 Learning dynamics of flow matching objective (single layer)

**Simplifying case $\mu = 0$, single layer network**  Note the special case $\mu = 0$

$$\nabla_{\mathbf{b}}\mathcal{L} = 2\mathbf{b} \tag{276}$$

$$\nabla_{\mathbf{W}}\mathcal{L} = 2\Big[\mathbf{W}\big(t^2\mathbf{\Sigma} + (1-t)^2\mathbf{I}\big) - \big(t\mathbf{\Sigma} - (1-t)\mathbf{I}\big)\Big] \tag{277}$$

$$\frac{d\mathbf{W}}{d\tau} = -\eta\nabla_{\mathbf{W}}\mathcal{L} \tag{278}$$

$$\frac{d\mathbf{W}}{d\tau} = -2\eta\Big[\mathbf{W}\big(t^2\mathbf{\Sigma} + (1-t)^2\mathbf{I}\big) - \big(t\mathbf{\Sigma} - (1-t)\mathbf{I}\big)\Big] \tag{279}$$

Using the eigenbasis projection

$$\frac{d\mathbf{W}\mathbf{u}_k}{d\tau} = -2\eta\Big[\mathbf{W}\big(t^2\mathbf{\Sigma} + (1-t)^2\mathbf{I}\big) - \big(t\mathbf{\Sigma} - (1-t)\mathbf{I}\big)\Big]\mathbf{u}_k \tag{280}$$

$$= -2\eta\Big[\mathbf{W}\mathbf{u}_k\big(t^2\lambda_k + (1-t)^2\big) - \big(t\lambda_k - (1-t)\big)\mathbf{u}_k\Big] \tag{281}$$

$$= -2\eta\big(t^2\lambda_k + (1-t)^2\big)\Big[\mathbf{W}\mathbf{u}_k - \frac{t\lambda_k - (1-t)}{t^2\lambda_k + (1-t)^2}\mathbf{u}_k\Big] \tag{282}$$

$$\mathbf{W}(\tau)\mathbf{u}_k - \frac{t\lambda_k - (1-t)}{t^2\lambda_k + (1-t)^2}\mathbf{u}_k = A\exp\Big(-2\eta\tau\big(t^2\lambda_k + (1-t)^2\big)\Big) \tag{283}$$

$$\mathbf{W}(\tau)\mathbf{u}_k = \frac{t\lambda_k - (1-t)}{t^2\lambda_k + (1-t)^2}\mathbf{u}_k + \Big(\mathbf{W}(0)\mathbf{u}_k - \frac{t\lambda_k - (1-t)}{t^2\lambda_k + (1-t)^2}\mathbf{u}_k\Big)\exp\Big(-2\eta\tau\big(t^2\lambda_k + (1-t)^2\big)\Big) \tag{284}$$

The full solutions of the weight and bias are

$$\mathbf{W}(\tau) = \mathbf{W}^* + \sum_k \Big(\mathbf{W}(0)\mathbf{u}_k - \frac{t\lambda_k - (1-t)}{t^2\lambda_k + (1-t)^2}\mathbf{u}_k\Big)\mathbf{u}_k^T\exp\Big(-2\eta\tau\big(t^2\lambda_k + (1-t)^2\big)\Big) \tag{285}$$

$$\mathbf{b}(\tau) = \mathbf{b}(0)\exp(-2\eta\tau) \tag{286}$$

It is easy to see this solution has a similar structure to that of the denoising score matching objective for diffusion models.

**Remarks**

- The learning dynamics of weight eigenmodes at different times $t$ were visualized in Fig. 8**A**.

- Note that the convergence speed of each mode is $\exp(-2\eta\tau(t^2\lambda_k + (1-t)^2))$. Similarly, at the same time $t$, the higher the data variance $\lambda_k$, the faster the convergence speed.

- At smaller $t$, all eigenmodes converge at similar speed.

- At larger $t \sim 1$, the eigenmodes are resolved at distinct speeds depending on the eigenvalues.

- Note that for each eigenvalue $\lambda_k$, there is a special time point where the convergence speed is maximized: $t^* = 1/(\lambda_k + 1)$.

### H.2.1  Interaction of weight learning and flow sampling

Consider the sampling dynamics of a flow matching model with learned weights:

$$\frac{d\mathbf{x}}{dt} = \mathbf{W}(\tau, t)\mathbf{x} + \mathbf{b}(\tau, t) \tag{287}$$

Assume the weight initialization is aligned and the same across $t$:

$$\mathbf{W}(0, t) = \sum_k Q_k \mathbf{u}_k \mathbf{u}_k^T \tag{288}$$

then

$$\mathbf{W}(\tau, t) = \mathbf{W}^* + \sum_k \left(\mathbf{W}(0) - \mathbf{W}^*\right)\mathbf{u}_k\mathbf{u}_k^T \exp\left(-2\eta\tau\left(t^2\lambda_k + (1-t)^2\right)\right) \tag{289}$$

$$= \sum_k \frac{t\lambda_k - (1-t)}{t^2\lambda_k + (1-t)^2}\mathbf{u}_k\mathbf{u}_k^T + \sum_k \left(Q_k - \frac{t\lambda_k - (1-t)}{t^2\lambda_k + (1-t)^2}\right)\mathbf{u}_k\mathbf{u}_k^T \exp\left(-2\eta\tau\left(t^2\lambda_k + (1-t)^2\right)\right) \tag{290}$$

Ignoring the bias part, consider the weight integration along $c_k(t) = \mathbf{u}_k^T\mathbf{x}(t)$:

$$\frac{d}{dt}c_k(t) = \left[\frac{t\lambda_k - (1-t)}{t^2\lambda_k + (1-t)^2} + \left(Q_k - \frac{t\lambda_k - (1-t)}{t^2\lambda_k + (1-t)^2}\right)\exp\left(-2\eta\tau\left(t^2\lambda_k + (1-t)^2\right)\right)\right]c_k(t) \tag{291}$$

We have the integration of the coefficient:

$$I = \int_0^1 dt\left[\frac{t\lambda_k - (1-t)}{t^2\lambda_k + (1-t)^2} + \left(Q_k - \frac{t\lambda_k - (1-t)}{t^2\lambda_k + (1-t)^2}\right)\exp\left(-2\eta\tau\left(t^2\lambda_k + (1-t)^2\right)\right)\right]$$

$$= \frac{\sqrt{2\pi}Q_k e^{-\frac{2\eta\tau\lambda_k}{\lambda_k+1}}\left(\text{erf}\left(\sqrt{2}\sqrt{\frac{\eta\tau}{\lambda_k+1}}\right) + \text{erf}\left(\sqrt{2}\lambda_k\sqrt{\frac{\eta\tau}{\lambda_k+1}}\right)\right)}{4\sqrt{\eta\tau\left(\lambda_k + 1\right)}} +$$

$$\frac{1}{2}\left(\text{Ei}(-2\eta\tau) - \text{Ei}\left(-2\eta\tau\lambda_k\right) + \log\left(\lambda_k\right)\right)$$

$$= \frac{1}{2}\log\left(\lambda_k\right) + \frac{1}{2}\left(\text{Ei}(-2\eta\tau) - \text{Ei}\left(-2\eta\tau\lambda_k\right)\right) + \frac{1}{2}\sqrt{\frac{\pi}{2\eta\tau\left(\lambda_k + 1\right)}}Q_k e^{-\frac{2\eta\tau\lambda_k}{\lambda_k+1}}$$

$$\left(\text{erf}\left(\sqrt{\frac{2\eta\tau}{\lambda_k + 1}}\right) + \text{erf}\left(\lambda_k\sqrt{\frac{2\eta\tau}{\lambda_k + 1}}\right)\right) \tag{292}$$

$$\Phi_k(1)$$

$$= \exp(I)$$

$$= \exp\left(\int_0^1 dt\left[\frac{t\lambda_k - (1-t)}{t^2\lambda_k + (1-t)^2} + \left(Q_k - \frac{t\lambda_k - (1-t)}{t^2\lambda_k + (1-t)^2}\right)\exp\left(-2\eta\tau\left(t^2\lambda_k + (1-t)^2\right)\right)\right]\right)$$

$$= \exp\left(\frac{1}{2}\log\left(\lambda_k\right) + \frac{1}{2}\left(\text{Ei}(-2\eta\tau) - \text{Ei}\left(-2\eta\tau\lambda_k\right)\right) + \frac{1}{2}\sqrt{\frac{\pi}{2\eta\tau\left(\lambda_k + 1\right)}}Q_k e^{-\frac{2\eta\tau\lambda_k}{\lambda_k + 1}}\right.$$

$$\left(\text{erf}\left(\sqrt{\frac{2\eta\tau}{\lambda_k + 1}}\right) + \text{erf}\left(\lambda_k\sqrt{\frac{2\eta\tau}{\lambda_k + 1}}\right)\right)\right)$$

$$= \sqrt{\lambda_k}\sqrt{\exp\left(\text{Ei}(-2\eta\tau) - \text{Ei}\left(-2\eta\tau\lambda_k\right)\right)}\exp\left(\frac{1}{2}\sqrt{\frac{\pi}{2\eta\tau\left(\lambda_k + 1\right)}}Q_k e^{-\frac{2\eta\tau\lambda_k}{\lambda_k + 1}}\right.$$

$$\left(\text{erf}\left(\sqrt{\frac{2\eta\tau}{\lambda_k + 1}}\right) + \text{erf}\left(\lambda_k\sqrt{\frac{2\eta\tau}{\lambda_k + 1}}\right)\right)\right) \tag{293}$$

$$\frac{\tilde{\lambda}_k}{\lambda_k} = \frac{\Phi_k(1)^2}{\lambda_k}$$

$$= \exp\left(\text{Ei}(-2\eta\tau) - \text{Ei}\left(-2\eta\tau\lambda_k\right)\right) \times \tag{294}$$

$$\exp\left(\sqrt{\frac{\pi}{2\eta\tau\left(\lambda_k + 1\right)}}Q_k e^{-\frac{2\eta\tau\lambda_k}{\lambda_k + 1}}\left(\text{erf}\left(\sqrt{\frac{2\eta\tau}{\lambda_k + 1}}\right) + \text{erf}\left(\lambda_k\sqrt{\frac{2\eta\tau}{\lambda_k + 1}}\right)\right)\right)$$

**Remarks**

- The learning dynamics of generated variance were visualized in Fig. 8**B**.

- The power law relationship between the convergence time $\tau_k^*$ of generated variance and the target variance $\lambda_k$ was shown in Fig. 9. For the harmonic mean criterion, the power law coefficient was also close to $-1$.

### H.3 Learning dynamics of flow matching objective (two layers)

Let $\mathbf{W} = PP^T$. Given the general loss,

$$\nabla_{\mathbf{W}}\mathcal{L} = 2\left[\mathbf{W}\left(t^2\boldsymbol{\Sigma} + (1-t)^2\mathbf{I}\right) - \left(t\boldsymbol{\Sigma} - (1-t)\mathbf{I}\right)\right] \tag{295}$$

$$\nabla_P\mathcal{L} = (\nabla_{\mathbf{W}}\mathcal{L})P + (\nabla_{\mathbf{W}}\mathcal{L})^T P$$

$$= \left[\nabla_{\mathbf{W}}\mathcal{L} + (\nabla_{\mathbf{W}}\mathcal{L})^T\right]P$$

$$\nabla_P\mathcal{L} = 2\left[PP^T\left(t^2\boldsymbol{\Sigma} + (1-t)^2\mathbf{I}\right) - \left(t\boldsymbol{\Sigma} - (1-t)\mathbf{I}\right)\right]P$$

$$+ 2\left[\left(t^2\boldsymbol{\Sigma} + (1-t)^2\mathbf{I}\right)PP^T - \left(t\boldsymbol{\Sigma} - (1-t)\mathbf{I}\right)\right]P$$

$$= 2\left[-2\left(t\boldsymbol{\Sigma} - (1-t)\mathbf{I}\right)P + \right.$$

$$\left. PP^T\left(t^2\boldsymbol{\Sigma} + (1-t)^2\mathbf{I}\right)P + \left(t^2\boldsymbol{\Sigma} + (1-t)^2\mathbf{I}\right)PP^T P\right]$$

Similarly, let $\mathbf{u}_k^T P = q_k^T$:

$$\mathbf{u}_k^T \nabla_P \mathcal{L} = 2\Big[-2\mathbf{u}_k^T\big(t\boldsymbol{\Sigma} - (1-t)\mathbf{I}\big)P +$$

$$\mathbf{u}_k^T P P^T\big(t^2\boldsymbol{\Sigma} + (1-t)^2\mathbf{I}\big)P + \mathbf{u}_k^T\big(t^2\boldsymbol{\Sigma} + (1-t)^2\mathbf{I}\big)PP^T P\Big]$$

$$= 2\Big[-2\big(t\lambda_k - (1-t)\big)\mathbf{u}_k^T P + q_k^T \sum_m P^T\mathbf{u}_m\big(t^2\lambda_m + (1-t)^2\big)\mathbf{u}_m^T P$$

$$+ \big(t^2\lambda_k + (1-t)^2\big)\mathbf{u}_k^T P \sum_n P^T\mathbf{u}_n\mathbf{u}_n^T P\Big]$$

$$= 2\Big[-2\big(t\lambda_k - (1-t)\big)q_k^T + q_k^T \sum_m q_m\big(t^2\lambda_m + (1-t)^2\big)q_m^T + \big(t^2\lambda_k + (1-t)^2\big)q_k^T \sum_n q_n q_n^T\Big]$$

$$\nabla_{q_k}\mathcal{L} = -4\big(t\lambda_k - (1-t)\big)q_k + 2\sum_m (q_k^T q_m)\big(t^2\lambda_m + (1-t)^2\big)q_m + 2\big(t^2\lambda_k + (1-t)^2\big)\sum_n (q_k^T q_n)q_n$$

$$= -4\big(t\lambda_k - (1-t)\big)q_k + 2\sum_m \big(t^2\lambda_m + (1-t)^2 + t^2\lambda_k + (1-t)^2\big)\,(q_k^T q_m)q_m$$

$$= -4\big(t\lambda_k - (1-t)\big)q_k + 2\sum_m \big(t^2\lambda_m + t^2\lambda_k + 2(1-t)^2\big)\,(q_k^T q_m)\,q_m \qquad (296)$$

**Simplifying assumption: aligned initialization**   Assume at initialization $q_k^T q_m = 0, \forall k \neq m$:

$$\nabla_{q_k}\mathcal{L} = -4\big(t\lambda_k - (1-t)\big)q_k + 4\big(t^2\lambda_k + (1-t)^2\big)(q_k^T q_k)q_k \qquad (297)$$

The learning dynamics follow:

$$\frac{d}{d\tau}q_k = -\eta\nabla_{q_k}\mathcal{L}$$

$$= 4\eta\Big[\big(t\lambda_k - (1-t)\big) - \big(t^2\lambda_k + (1-t)^2\big)(q_k^T q_k)\Big]q_k$$

$$\frac{1}{2}\frac{d}{d\tau}(q_k^T q_k) = 4\eta\Big[\big(t\lambda_k - (1-t)\big) - \big(t^2\lambda_k + (1-t)^2\big)(q_k^T q_k)\Big](q_k^T q_k) \qquad (298)$$

Using abbreviations:

$$A := 8\eta\big(t\lambda_k - (1-t)\big)$$
$$B := 8\eta\big(t^2\lambda_k + (1-t)^2\big)$$
$$r(\tau) := \|q_k\|^2$$

we can see the core structure:

$$\frac{dr(\tau)}{d\tau} = Ar - Br^2 \qquad (299)$$

With initialization $r(0) = q_k^T q_k(\tau = 0) = Q_k$, the solution reads:

$$\|q_k\|^2(\tau) = \frac{A}{B}\frac{1}{1 + (\frac{A}{BQ_k} - 1)e^{-A\tau}}$$

$$= \frac{t\lambda_k - (1-t)}{t^2\lambda_k + (1-t)^2}\frac{Q_k}{Q_k + (\frac{t\lambda_k-(1-t)}{t^2\lambda_k+(1-t)^2} - Q_k)e^{-8\eta\tau\big(t\lambda_k-(1-t)\big)}}$$

$$= Q_k^* \frac{Q_k}{Q_k + (Q_k^* - Q_k)e^{-8\eta\tau\big(t\lambda_k-(1-t)\big)}} \qquad (300)$$

where

$$Q_k^* = \frac{t\lambda_k - (1-t)}{t^2\lambda_k + (1-t)^2} \qquad (301)$$

So the overall weight dynamics read:

$$\mathbf{W}(\tau; t) = \sum_k \|q_k\|^2(\tau) \mathbf{u}_k \mathbf{u}_k^T$$

$$= \sum_k \frac{Q_k}{Q_k + (Q_k^* - Q_k)e^{-8\eta\tau\left(t\lambda_k - (1-t)\right)}} Q_k^* \mathbf{u}_k \mathbf{u}_k^T$$

$$= \sum_k \frac{Q_k}{Q_k + (Q_k^* - Q_k)e^{-8\eta\tau\left(t\lambda_k - (1-t)\right)}} \frac{t\lambda_k - (1-t)}{t^2\lambda_k + (1-t)^2} \mathbf{u}_k \mathbf{u}_k^T \qquad (302)$$

Note that the optimal weight (301) is not positive definite, but the two-layer symmetric weight network is. So, different from the diffusion case with denoiser loss, there are two scenarios:

- When $t > \frac{1}{\lambda_k + 1}$, $A > 0$ and $Q^* > 0$. The solution converges to $Q^*$: $\lim_{\tau \to \infty} \|q_k\|^2(\tau) = Q_k^*$.

- When $t < \frac{1}{\lambda_k + 1}$, $A < 0$ and $Q^* < 0$. In this case, the optimal solution is "non-achievable" by a two-layer symmetric network. $\lim_{\tau \to \infty} e^{-A\tau} \to \infty$, so $\lim_{\tau \to \infty} \|q_k\|^2(\tau) = 0$. In other words, 0 becomes a stable fixed point instead of $Q^*$, and the solution $\|q_k\|^2(\tau)$ will be attracted to and stuck at 0.

Thus,

$$\lim_{\tau \to \infty} \mathbf{W}(\tau; t) = \sum_{k, \text{ where } \lambda_k < \frac{1}{t} - 1} \frac{t\lambda_k - (1-t)}{t^2\lambda_k + (1-t)^2} \mathbf{u}_k \mathbf{u}_k^T \qquad (303)$$

Because of this, asymptotically speaking, the symmetric network architecture $P^T P$ will not approximate this vector field very well.

Thus, for the purpose of studying the learning dynamics of flow matching models, some extension beyond the symmetric two-layer linear network is required for a thorough analysis.

# I Detailed Experimental Procedure

## I.1 Computational Resources

All experiments were conducted on research cluster. Model training was performed on single A100 / H100 GPU. MLP training experiments took 20mins-2hrs while CNN based UNet training experiments took 5-8 hours, using around 20GB RAM.

Evaluations were also done on single A100 / H100 GPU, with heavy covariance computation done with CUDA and trajectory plotting and fitting on CPU. Covariance computation for generated samples generally took a few minutes.

## I.2 MLP architecture inspired by UNet

We used the following custom architecture inspired by UNet in [26] and [62] paper. The basic block is the following

```python
class UNetMLPBlock(torch.nn.Module):
    def __init__(self,
        in_features, out_features, emb_features, dropout=0, skip_scale=1, eps=1e-5,
        adaptive_scale=True, init=dict(), init_zero=dict(),
    ):
        super().__init__()
        self.in_features = in_features
        self.out_features = out_features
        self.emb_features = emb_features
        self.dropout = dropout
        self.skip_scale = skip_scale
        self.adaptive_scale = adaptive_scale

        self.norm0 = nn.LayerNorm(in_features, eps=eps)
            #GroupNorm(num_channels=in_features, eps=eps)
        self.fc0 = Linear(in_features=in_features, out_features=out_features, **init)
        self.affine = Linear(in_features=emb_features, out_features=out_features*(2
            if adaptive_scale else 1), **init)
        self.norm1 = nn.LayerNorm(out_features, eps=eps)
            #GroupNorm(num_channels=out_features, eps=eps)
        self.fc1 = Linear(in_features=out_features, out_features=out_features,
            **init_zero)

        self.skip = None
        if out_features != in_features:
            self.skip = Linear(in_features=in_features, out_features=out_features,
                **init)

    def forward(self, x, emb):
        orig = x
        x = self.fc0(F.silu(self.norm0(x)))

        params = self.affine(emb).to(x.dtype) # .unsqueeze(1)
        if self.adaptive_scale:
            scale, shift = params.chunk(chunks=2, dim=1)
            x = F.silu(torch.addcmul(shift, self.norm1(x), scale + 1))
        else:
            x = F.silu(self.norm1(x.add_(params)))

        x = self.fc1(F.dropout(x, p=self.dropout, training=self.training))
        x = x.add_(self.skip(orig) if self.skip is not None else orig)
        x = x * self.skip_scale

        return x
```

and the full architecture backbone

```python
class UNetBlockStyleMLP_backbone(nn.Module):
    """A time-dependent score-based model."""

    def __init__(self, ndim=2, nlayers=5, nhidden=64, time_embed_dim=64,):
        super().__init__()
        self.embed = GaussianFourierProjection(time_embed_dim, scale=1)
        layers = nn.ModuleList()
        layers.append(UNetMLPBlock(ndim, nhidden, time_embed_dim))
        for _ in range(nlayers-2):
            layers.append(UNetMLPBlock(nhidden, nhidden, time_embed_dim))
        layers.append(nn.Linear(nhidden, ndim))
        self.net = layers

    def forward(self, x, t_enc, cond=None):
        # t_enc : preconditioned version of sigma, usually
        # ln_std_vec = torch.log(std_vec) / 4
        if cond is not None:
            raise NotImplementedError("Conditional training is not implemented")
        t_embed = self.embed(t_enc)
        for layer in self.net[:-1]:
            x = layer(x, t_embed)
        pred = self.net[-1](x)
        return pred
```

```python
class EDMPrecondWrapper(nn.Module):
    def __init__(self, model, sigma_data=0.5, sigma_min=0.002, sigma_max=80,
     rho=7.0):
        super().__init__()
        self.model = model
        self.sigma_data = sigma_data
        self.sigma_min = sigma_min
        self.sigma_max = sigma_max
        self.rho = rho

    def forward(self, X, sigma, cond=None, ):
        sigma[sigma == 0] = self.sigma_min
        ## edm preconditioning for input and output
        ## https://github.com/NVlabs/edm/blob/main/training/networks.py#L632
        # unsqueze sigma to have same dimension as X (which may have 2-4 dim)
        sigma_vec = sigma.view([-1, ] + [1, ] * (X.ndim - 1))
        c_skip = self.sigma_data ** 2 / (sigma_vec ** 2 + self.sigma_data ** 2)
        c_out = sigma_vec * self.sigma_data / (sigma_vec ** 2 + self.sigma_data **
            2).sqrt()
        c_in = 1 / (self.sigma_data ** 2 + sigma_vec ** 2).sqrt()
        c_noise = sigma.log() / 4
        model_out = self.model(c_in * X, c_noise, cond=cond)
        return c_skip * X + c_out * model_out
```

This architecture can efficiently learn point cloud distributions. More details about the architecture and training can be found in code supplementary.

## I.3  EDM Loss Function

We employ the loss function $\mathcal{L}_{\text{EDM}}$ introduced in the Elucidated Diffusion Model (EDM) paper [26], which is one specific weighting scheme for training diffusion models.

For each data point $\mathbf{x} \in \mathbb{R}^d$, the loss is computed as follows. The noise level for each data point is sampled from a log-normal distribution with hyperparameters $P_{\text{mean}}$ and $P_{\text{std}}$ (e.g., $P_{\text{mean}} = -1.2$ and

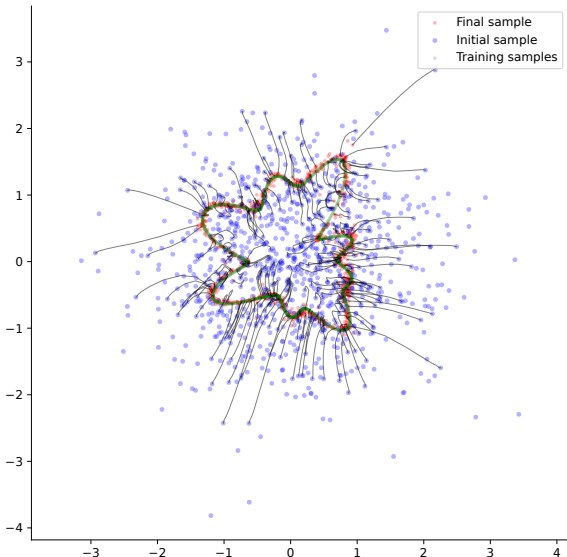

Figure 29: **Example of learning to generate low-dimensional manifold with Song UNet-inspired MLP denoiser.**

$P_{\text{std}} = 1.2$). Specifically, the noise level $\sigma$ is sampled via

$$\sigma = \exp\left(P_{\text{mean}} + P_{\text{std}}\,\epsilon\right), \quad \epsilon \sim \mathcal{N}(0,1).$$

The weighting function per noise scale is defined as:

$$w(\sigma) = \frac{\sigma^2 + \sigma_{\text{data}}^2}{\left(\sigma\,\sigma_{\text{data}}\right)^2},$$

with hyperparameter $\sigma_{\text{data}}$ (e.g., $\sigma_{\text{data}} = 0.5$). The noisy input $\mathbf{y}$ is created by the following,

$$\mathbf{y} = \mathbf{x} + \sigma\mathbf{n}, \quad \mathbf{n} \sim \mathcal{N}\left(\mathbf{0}, \mathbf{I}_d\right),$$

Let $D_\theta(\mathbf{y}, \sigma, \text{labels})$ denote the output of the denoising network when given the noisy input $\mathbf{y}$, the noise level $\sigma$, and optional conditioning labels. The EDM loss per data point can be computed as:

$$\mathcal{L}(\mathbf{x}) = w(\sigma)\,\|D_\theta(\mathbf{x} + \sigma\mathbf{n}, \sigma, \text{labels}) - \mathbf{x}\|^2.$$

Taking expectation over the data points and noise scales, the overall loss reads

$$\mathcal{L}_{EDM} = \mathbb{E}_{\mathbf{x} \sim p_{data}} \mathbb{E}_{\mathbf{n} \sim \mathcal{N}(0, \mathbf{I}_d)} \mathbb{E}_\sigma \left[ w(\sigma)\,\|D_\theta(\mathbf{x} + \sigma\mathbf{n}, \sigma, \text{labels}) - \mathbf{x}\|^2 \right] \tag{304}$$

```python
class EDMLoss:
    def __init__(self, P_mean=-1.2, P_std=1.2, sigma_data=0.5):
        self.P_mean = P_mean
        self.P_std = P_std
        self.sigma_data = sigma_data

    def __call__(self, net, X, labels=None, ):
        rnd_normal = torch.randn([X.shape[0],] + [1, ] * (X.ndim - 1),
            device=X.device)
        # unsqueeze to match the ndim of X
        sigma = (rnd_normal * self.P_std + self.P_mean).exp()
        weight = (sigma ** 2 + self.sigma_data ** 2) / (sigma * self.sigma_data) ** 2
        # maybe augment
        n = torch.randn_like(X) * sigma
        D_yn = net(X + n, sigma, cond=labels, )
        loss = weight * ((D_yn - X) ** 2)
        return loss
```

### I.4 Experiment 1: Diffusion Learning of High-dimensional Gaussian Data

#### I.4.1 Data Generation and Covariance Specification

We consider learning a score-based generative model on synthetic data drawn from a high-dimensional Gaussian distribution of dimension $d = 128, 256, 512$. Specifically, we first sample a vector of variances

$$\boldsymbol{\sigma}^2 = \left(\sigma_1^2, \sigma_2^2, \ldots, \sigma_d^2\right),$$

where each $\sigma_i^2$ is drawn from a log-normal distribution (implemented via `torch.exp(torch.randn(...))`). We then sort them in descending order and normalize these variances to have mean equals 1 to fix the overall scale. Denoting

$$\mathbf{D} = \mathrm{diag}\left(\sigma_1^2, \ldots, \sigma_d^2\right),$$

we generate a random rotation matrix $\mathbf{R} \in \mathbb{R}^{d \times d}$ by performing a QR decomposition of a matrix of i.i.d. Gaussian entries. This allows us to construct the covariance

$$\boldsymbol{\Sigma} = \mathbf{R} \, \mathbf{D} \, \mathbf{R}^\mathsf{T}.$$

This rotation matrix $\mathbf{R}$ is the eigenbasis of the true covariance matrix. To obtain training samples $\{\mathbf{x}_i\} \subset \mathbb{R}^d$, we draw $\mathbf{x}_i$ from $\mathcal{N}\left(\mathbf{0}, \boldsymbol{\Sigma}\right)$. In practice, we generate a total of $10,000$ samples and stack them as `pnts`. We compute the empirical covariance of the training set, $\boldsymbol{\Sigma}_{\mathrm{emp}} = \mathrm{Cov}($`pnts`$)$, and verify that it is close to the prescribed true covariance $\boldsymbol{\Sigma}$.

#### I.4.2 Network Architecture and Training Setup

We train a multi-layer perceptron (MLP) to approximate the noise conditional score function. The base network, implemented as

`model=UNetBlockStyleMLP_backbone(ndim=d, nlayers=5, nhidden=256, time_embed_dim=256)`

maps a data vector $\mathbf{x} \in \mathbb{R}^d$ and a time embedding $\tau$ to a vector of the same dimension $\mathbb{R}^d$. This backbone is then wrapped in an EDM-style preconditioner via:

`model_precd = EDMPrecondWrapper(model, `$\sigma_{\mathrm{data}} = 0.5$, $\sigma_{\mathrm{min}} = 0.002$, $\sigma_{\mathrm{max}} = 80$, $\rho = 7.0$`)`,

which standardizes and scales the input according to the EDM framework [26].

We use EDM loss with hyperparameters `P_mean` $= -1.2$, `P_std` $= 1.2$, and $\sigma_{\mathrm{data}} = 0.5$. We train the model for $5000$ steps using mini-batches of size $1024$. The Adam optimizer is used with a learning rate `lr` $= 10^{-4}$. Each training step processes a batch of data from `pnts`, adds noise with randomized noise scales, and backpropagates through the EDM loss. The loss values at each training steps are recorded.

#### I.4.3 Sampling and Trajectory Visualization

To visualize the sampling evolution, we sample from the diffusion model using the Heun's 2nd order deterministic sampler, starting from $\mathbf{z} \sim \mathcal{N}(\mathbf{0}, \mathbf{I}_d)$

`edm_sampler(model, `$\mathbf{z}$`, num_steps` $= 20$, $\sigma_{\mathrm{min}} = 0.002$, $\sigma_{\mathrm{max}} = 80$, $\rho = 7$`)`.

We store these samples in `sample_store` to track how sampled distribution evolves over training.

#### I.4.4 Covariance Evaluation in the True Eigenbasis

To measure how well the trained model captures the true covariance structure, we compute the sample covariance from the final generated samples, denoted $\widehat{\boldsymbol{\Sigma}}_{\mathrm{sample}}$. We then project $\widehat{\boldsymbol{\Sigma}}_{\mathrm{sample}}$ and the true $\boldsymbol{\Sigma}$ into the eigenbasis of $\boldsymbol{\Sigma}$. Specifically, letting $\mathbf{R}$ be the rotation used above, we compute

$$\mathbf{R}^\mathsf{T} \, \widehat{\boldsymbol{\Sigma}}_{\mathrm{sample}} \, \mathbf{R} \quad \text{and} \quad \mathbf{R}^\mathsf{T} \, \boldsymbol{\Sigma} \, \mathbf{R}.$$

Since $\boldsymbol{\Sigma} = \mathbf{R} \, \mathbf{D} \, \mathbf{R}^\mathsf{T}$ is diagonal in that basis, we then compare the diagonal elements of $\mathbf{R}^\mathsf{T} \, \widehat{\boldsymbol{\Sigma}}_{\mathrm{sample}} \, \mathbf{R}$ with $\mathrm{diag}(\mathbf{D})$. As training proceeds, we track the ratio $\mathrm{diag}(\mathbf{R}^\mathsf{T} \, \widehat{\boldsymbol{\Sigma}}_{\mathrm{sample}} \, \mathbf{R})/\mathrm{diag}(\mathbf{D})$ to observe convergence toward 1 across the spectrum.

All intermediate results, including loss values and sampled trajectories, are stored to disk for later analysis.

## I.5  Experiment 2: Diffusion Learning of MNIST | MLP

### I.5.1  Data Preprocessing

For our second experiment, we apply the same EDM architecture to several natural image datasets: MNIST, CIFAR, AFHQ32, FFHQ32, FFHQ32-fixword, FFHQ32-randomword. All dataset except for MNIST are RGB images with 32 resolution, while MNIST is BW images with 28 resolution. These images were flattened as vectors, (784d for MNIST, 3072 for others) and stacked as `pnts` matrix. We normalize these intensities from $[0,1]$ to $[-1,1]$ by $\mathbf{x} \mapsto \frac{\mathbf{x}-0.5}{0.5}$. The resulting data tensor `pnts` is then transferred to GPU memory for training, and we estimate its empirical covariance $\mathbf{\Sigma}_{\mathrm{emp}} = \mathrm{Cov}(\texttt{pnts})$ for reference.

### I.5.2  Network Architecture and Training Setup

Since the natural dataset is higher dimensional than the synthetic data in the previous experiment, we use a deeper MLP network: For MNIST:

$\texttt{model} = \texttt{UNetBlockStyleMLP\_backbone}(\texttt{ndim} = 784, \texttt{nlayers} = 8, \texttt{nhidden} = 1024, \texttt{time\_embed\_dim} = 128)$.

For others

$\texttt{model} = \texttt{UNetBlockStyleMLP\_backbone}(\texttt{ndim} = 3072, \texttt{nlayers} = 8, \texttt{nhidden} = 3072, \texttt{time\_embed\_dim} = 128)$.

We again wrap this MLP in an EDM preconditioner:

$\texttt{model\_precd} = \texttt{EDMPrecondWrapper}(\texttt{model}, \sigma_{\mathrm{data}} = 0.5, \sigma_{\mathrm{min}} = 0.002, \sigma_{\mathrm{max}} = 80, \rho = 7.0)$.

The model is trained using the `EDMLoss` described in the previous section, with parameters `P_mean` $= -1.2$, `P_std` $= 1.2$, and $\sigma_{\mathrm{data}} = 0.5$. We set the training hyperparameters to $\texttt{lr} = 10^{-4}$, $\texttt{n\_steps} = 100000$, and $\texttt{batch\_size} = 2048$.

### I.5.3  Sampling and Analysis

As before, we define a callback function `sampling_callback_fn` that periodically draws i.i.d. Gaussian noise $\mathbf{z} \sim \mathcal{N}(\mathbf{0}, \mathbf{I}_{784})$ and applies the EDM sampler to produce generated samples. These intermediate samples are stored in `sample_store` for later analysis.

In addition, we assess convergence of the mean of the generated samples by computing

$$\|\mathbb{E}[\texttt{x\_out}] \; - \; \mathbb{E}[\texttt{pnts}]\|^2,$$

and we track how this mean-squared error evolves over training steps. We also examine the sample covariance $\widehat{\mathbf{\Sigma}}_{\mathrm{sample}}$ of the final outputs, comparing its diagonal in a given eigenbasis to a target spectrum (e.g. the diagonal variances of the training data or a reference covariance).

All trajectories and intermediate statistics are saved to disk for further inspection. In particular, we plot the difference between $\widehat{\mathbf{\Sigma}}_{\mathrm{sample}}$ and $\mathbf{\Sigma}$ in an eigenbasis to illustrate whether the learned samples capture the underlying covariance structure of the training data.

## I.6  Experiment 3: Diffusion learning of Image Datasets with EDM-style CNN UNet

We used model configuration similar to https://github.com/NVlabs/edm, but with simplified training code more similar to previous experiments.

For the MNIST dataset, we trained a UNet-based CNN (with four blocks, each containing one layer, no attention, and channel multipliers of 1, 2, 3, and 4) on MNIST for 50,000 steps using a batch size of 2,048, a learning rate of $10^{-4}$, 16 base model channels, and an evaluation sample size of 5,000.

For the CIFAR-10 dataset, we trained a UNet model (with three blocks, each containing one layer, wide channels of size 128, and attention at resolution 16) for 50,000 steps using a batch size of 512, a learning rate of $10^{-4}$, and an evaluation sample size of 2,000 (evaluated in batches of 1,024) with 20 sampling steps.

For the AFHQ, FFHQ (32 pixels) dataset, we used the same UNet architecture and training setup, with four blocks, wide channels of size 128, and attention at resolution 8, trained for 50,000 steps

with a batch size of 256 and a learning rate of $1 \times 10^{-4}$. Evaluation was conducted on 2,000 samples in batches of 512.

For the AFHQ, FFHQ (64 pixels) dataset, we trained a UNet model with four blocks (each containing one layer, wide channels of size 128, and attention at resolution 8) for 250,000 steps using a batch size of 256, a learning rate of $1 \times 10^{-4}$, and an evaluation sample size of 2,000 (evaluated in batches of 512).

## I.7 Architectural Ablation: Diffusion Learning of Image Datasets with EDM-style CNN ResNet

To systematically examine the effects of network depth and width on diffusion learning dynamics, we conducted a controlled set of experiments using EDM-style CNN ResNet architectures trained on the FFHQ-32×32 dataset.

We designed a simplified single-resolution ResNet denoiser (`SongUNetResNet`) without skip connections or attention. It consists of a stack of residual convolutional blocks conditioned on positional timestep embeddings, followed by a normalization and output convolution. This minimal EDM-style architecture isolates the effects of *depth and width* on diffusion learning dynamics.

All models followed the EDM training configuration [26] with simplified code consistent with previous experiments, and were trained for 50,000 steps using a batch size of 256, Adam optimizer, a learning rate of $1 \times 10^{-4}$. 2,000 samples are generated at given training steps for evaluation. No attention layers were used.

We systematically varied two architectural factors: (1) *network depth*, by adjusting the number of residual layers per block ($L \in \{1, 2, 3, 5\}$); and (2) *network width*, by varying the base channel dimension ($C \in \{4, 6, 8, 12, 16, 32, 128, 256\}$). Each configuration was trained independently with identical optimizer settings and no data augmentations to isolate the contribution of architecture to convergence and spectral scaling behavior.

