# OpenReview forum: "An Analytical Theory of Spectral Bias in the Learning Dynamics of Diffusion Models"
_NeurIPS.cc/2025/Conference — NeurIPS 2025 spotlight_

### Official Review · Reviewer_d6G6 · 2025-06-30

**Clarity:** 3
**Significance:** 4
**Originality:** 4
**Rating:** 5
**Confidence:** 4

**Summary:**

This manuscript delivers a rigorous analysis of diffusion model training dynamics under a linear denoiser assumption. Assuming the data follow a Gaussian distribution, the optimal denoiser becomes a linear operator whose convergence rate is governed by the eigenvalue spectrum of the data covariance matrix. While the theoretical results hold exactly for linear denoisers, the authors demonstrate through experiments that nonlinear neural networks exhibit qualitatively similar convergence behaviour.

**Questions:**

1. In line 94, does $s_\theta$ denote the score function and $D_\theta$ the denoiser? Please clarify to avoid confusion.
2. Equation (6) defines $W_\sigma=\Sigma(\Sigma + \sigma^2 I)^{-1}$, but Equation (3) suggests $W_\sigma=(\Sigma + \sigma^2 I)^{-1}\Sigma$. Which form is correct?
3. In Section 5.2, Proposition 5.3 uses a Fourier basis, whereas Proposition 5.4 relies on principal components of local patches. What justifies this change given that the filter width is the only differing parameter?

**Ethical Concerns:**

["NO or VERY MINOR ethics concerns only"]

**Final Justification:**

The paper is already solid and the authors provide further explanations to my questions especially regarding Fourier basis vs local patches. I will hence maintain my score.

**Limitations:**

yes

**Quality:**

4

**Strengths And Weaknesses:**

**Strengths**

- Provides a sound theoretical framework linking covariance eigenvalues to denoiser convergence speed.
- Empirical validation supports the theoretical predictions and reveals analogous trends in nonlinear denoisers.
- Section 6.2’s experiments highlighting distinct CNN behaviors enrich the manuscript.
- This work will be important in helping researchers understanding the generalization power of diffusion models.

**Weaknesses**

- The analysis is limited to linear denoisers. However, the authors do acknowledge this limitation and I wouldn't consider it a major drawback, as the empirical results suggest that the theory is still relevant for nonlinear denoisers.

---

> ### Author Rebuttal · Authors · 2025-07-29
>
> We thank the reviewer for their careful reading and insightful feedback. Below, we address each of the questions in turn:
>
> **Q1**: Yes indeed, at line 94, $s_\theta$ denotes the score function and $D_\theta$ denotes the denoiser. We will add an explicit statement in the camera-ready manuscript to avoid any ambiguity.
>
> **Q2**: Both forms in Eq.3 and Eq.6 are correct. Since $\Sigma$ is the data covariance, it’s real symmetric and positive (semi-)definite, thus it admits the eigen-decomposition,  $\Sigma=U\Lambda U^\top$, i.e. the principal component of data. It follows that, $\Sigma+\sigma^2 I=U(\sigma^2+\Lambda)U^\top;\ (\Sigma+\sigma^2 I)^{-1}=U(\sigma^2+\Lambda)^{-1}U^\top$.
>
> Thus all three matrices, $\Sigma$, $\Sigma+\sigma^2 I$ and $(\Sigma+\sigma^2 I)^{-1}$ share the same eigenbasis, hence ***commute***. So Eq.6 and Eq.3 are equivalent to each other.
>
> **Q3**: Thanks for noticing this differences!
>
> At first glance one might hope to carry out the learning dynamics analysis in the Fourier basis, since convolution is diagonal there. However, when the filter support K is strictly smaller than the signal length $N$, the DFT coefficients of the truncated filter $\gamma_\sigma=\sqrt{N}Fw_\sigma$ lie in only a K-dimensional subspace of $\mathbb{C}^N$. In other words, the Fourier coefficients of $w_\sigma$ are no longer independent but constrained to the column space of the corresponding DFT submatrix, turning the gradient dynamics into a constrained optimization problem in Fourier space.
>
> By contrast, in the spatial domain the $K$ raw filter parameters remain independent. This allows us to leverage the well-known properties of circulant and shift matrices and obtain a much cleaner analysis (see Appendix G.4.1 and Proposition 5.4). Intuitively, each local convolutional filter “sees” only the statistics of its $K$-pixel patch and uses those statistics to linearly predict the denoised value at the *center pixel*.
>
> We will expand our discussion of this technical choice in the camera-ready version to make the motivation and technical benefits clearer.
>
> We appreciate the reviewer’s suggestions and will incorporate these clarifications into the final manuscript.

---

> > ### Comment · Reviewer_d6G6 · 2025-08-03
> >
> > Thanks for answering my questions.
> >
> > For Q2, I would suggest to simply stick with one order of matrix multiplication to avoid any confusion.
> >
> > Thank you for the explanation of Q3. I think this part is very interesting and it would be great if some further explanation like your answer can be incorporated in a revised version.
> >
> > I'll maintain my supporting score.

---

> > > ### Author Response · Authors · 2025-08-05
> > > **Thanks for your supporting score and effort in reviewing our paper!**
> > >
> > > Thanks for your great suggestion!
> > > We will stick to one order of the formula to avoid confusion and incorporate the discussion of this technical choice in our final paper.

---

### Official Review · Reviewer_PL6U · 2025-07-01

**Clarity:** 2
**Significance:** 2
**Originality:** 2
**Rating:** 4
**Confidence:** 3

**Summary:**

This paper presents a theoretical analysis of the learning dynamics in diffusion models, focusing on the simplified setting of linear denoisers trained on data equivalent to a single anisotropic Gaussian distribution. The authors derive a closed-form solution for the weight dynamics under gradient flow (Equation 7) and, assuming commutativity of the weight matrices across noise scales, provide an analytical expression for the generated data distribution at any point during training (Proposition 4.2).

The central implication of such is the existence of a "spectral bias", where the time required to learn an eigenmode is inversely proportional to its variance ($\tau_k\propto \lambda_k^{-1}$). This results in a coarse-to-fine learning trajectory, with high-variance modes being learned much faster than low-variance ones. The analysis is extended to deeper models, such as a two-layer symmetric linear network (Proposition 5.1), and to linear convolutional networks, where a similar spectral bias is predicted in the Fourier domain (Propositions 5.3 and 5.4).

To validate these theoretical predictions, the authors conduct empirical studies using MLP and CNN-based denoisers. The results show that the learning dynamics of MLPs align well with the theory. In contrast, CNNs show a different behavior where many modes are learned near-simultaneously, suggesting that the bias shifts from the global image covariance to the patch-level covariance, a phenomenon the authors attribute to the inductive bias of local convolutions.

**Questions:**

Can you elaborate on Figure 17~20? While I agree they resemble the increase-decrease separation in Figure 2, the tailing eigenmodes seem not to show strong incremental learning, also the scatter points seem not to agree with the fitted line (both increase fit and decrease fit).

**Ethical Concerns:**

["NO or VERY MINOR ethics concerns only"]

**Final Justification:**

While I still have reservations about the practical utility of linear denoisers, I recognize that the paper is of high quality regarding extensive experiment results and rigorous derivations. Therefore, I am raising my score to 4.

**Limitations:**

While I commend the authors for their effort in building an analytical theory, the paper offers limited practical insight due to its restrictive core assumptions. Contemporary work is beginning to offer precise insights into phenomena like the effect of early stopping (https://arxiv.org/abs/2505.17638), but this paper's impact is constrained by its idealized setting.

**Quality:**

3

**Strengths And Weaknesses:**

Strengths:
* The paper provides a thorough and rigorous gradient-flow analysis for the learning dynamics of linear denoisers, with extension to two-layer and convolutional cases.
* The theoretical claims are supported by empirical validation on both synthetic and real-world image datasets (Figures 3 and 9).


Weaknesses:
* The primary limitation is the reliance on strong simplifying assumptions, a linear denoiser and data drawn from a single Gaussian distribution which may not fully capture the complexity of practical diffusion models.
* The extensions to symmetric linear and convolutional networks feel less developed. For example, while the authors attribute the faster convergence of CNNs to the theoretical "N-fold speed-up" from weight sharing, this connection is not rigorously established. The empirical speed-up could be influenced by many confounding factors in a modern U-Net architecture and optimization.
* The presentation could be streamlined to better emphasize the central theoretical contributions. For instance, the key result for weight dynamics (Equation 7) is presented as an intermediate step rather than a standalone proposition or theorem. The frequent use of subsections for brief interpretive points also fragments the narrative flow.

---

> ### Author Rebuttal · Authors · 2025-07-30
>
> We thank the reviewer for their thorough summary and thoughtful feedback!
>
> **W1**: With respect, we find this critique not entirely accurate. First, while our theoretical analysis assumes a linear, independent denoiser, it imposes no constraint on the data distribution. Second, on the empirical side (Experiment 1, Fig. 9), we show that fully connected (MLP‐based) diffusion models quantitatively exhibit the predicted inverse‐variance spectral bias and scaling coefficient, despite violating both linearity and independence assumptions. This behavior qualitatively persists across multiple natural image datasets (Experiment 2, Fig. 11). These results indicate that, although our theory relies on simplified assumptions, these do not limit its applicability to real world diffusion models with fully connected structure. However, convolutional architectures do alter the observed learning dynamics. We believe a more targeted analysis of convolutional structures could explain these differences.
>
> **W2**: We acknowledge that, due to page constraints, the treatment of deeper linear and convolutional cases in the main text is relatively brief. However, the full derivations and analyses are provided in Appendices F and G. In theory, the N-fold speed-up is rigorously established in Proposition 5.3. We recognize, however, that linking this result directly to the faster empirical convergence of practical U-Nets is more challenging—isolating a single architectural component responsible for this speed-up is nontrivial.
>
> Additionally, we performed more extensive simulation of the distributional learning dynamics for patch convolutional network. We found generally the power law scaling is attenuated, and the specific scaling exponent is strongly affected by the patch size: smaller filter size leads to more simultaneous emergence of modes and closer to 0 scaling coefficients. We think these new numerical results can better link to our empirical observations of attenuated scaling in practical UNet training. The details of the new results are presented in the section at the end.
>
> **W3**: Thank you for the suggestion. Although we originally presented Equation 7 inline given its straightforward derivation, we agree that it merits its own formal proposition alongside Proposition 5.1 in the camera-ready version.
>
> **Q1**: We appreciate the reviewer’s scrutiny of Appendix Figures 17–20. In those figures, we quantify the distribution of generated patches and track how their covariances evolve along the principal components of training patches. While these plots hint at a patch-space spectral bias, the effect is relatively small and less consistent than in the MLP setting or as predicted by our theory. Given this variability, we refrain from making strong claims about spectral bias in practical U-Net learning at the patch level. We believe that a refined theory incorporating additional details of the U-Net architecture may better capture these empirical curves.
>
> **L1**: We appreciate the reviewer drawing our attention to the concurrent work (arXiv:2505.17638). We value their careful experiments, analytical theory, and practical insights. But we regard our contributions as complementary. Their theory assumes data drawn from a white Gaussian distribution and leverages the Gaussian equivalence principle of random features to derive the feature covariance spectrum, naturally explaining the separation of timescales between generalization and memorization. However, it does not account for nontrivial data covariance structure or its interaction with sampling dynamics—precisely the focus of our paper.
>
> In summary, there are multiple tractable models for analyzing learning dynamics: our work focuses on linear and deep linear networks, while the concurrent work explores random feature models. The linear setting renders the probability‐flow ODE sampling dynamics analytically tractable, enabling us to track the spectral evolution of the learned distribution. In contrast, random feature models facilitate discussions of feature number scaling. A promising future direction is to unify these approaches by analyzing random feature models with non-white data covariance, yielding a richer understanding of memorization and generalization as functions of the data spectrum. We will incorporate this comparison and discussion in the final version of our manuscript.
>
> ----
> **Numerical simulation of the patch convolution learning dynamics**
>
> We took the FFHQ32 dataset as our example, and used the statistics of patches cropped from it to compute the optimal linear filter at each noise scale $w_\sigma^*$. Then we used proposition 5.4 to obtain the learned filter at different training time, then solve the PF-ODE in the spatial domain to get samples. We computed their variance along eigenmodes and the scaling laws.
>
> During numerical simulation, we found there are two key factors affecting the observed power law relation between image eigenvalues and convergence time, namely, the **patch size** of the filter $P$ and the **initialization of the filter weights**.
>
> If filter weights are initialized as identity without scaling, the initial sample variance will be way too high, and the exponent will be around ~ $-0.4$ — already attenuated than the linear case solution. One practically relevant alternative is to initialize the filter weight as identity scaled by  $\sigma_{data}^2 / (\sigma^2 + \sigma_{data}^2)$, inspired by the coefficient of skip connection $c_{skip}$ in EDM preconditioning scheme.
>
> If so, we’d observe modes with both increasing and decreasing variances. The specific scaling relation is affected by the filter patch size (summarized in following table). We found that when the patch size become smaller, scaling exponent of decreasing mode approach zero. from around -0.4 for P=15 to around -0.21 for P=3 (3x3 convolution), and the convergence time generally increases with smaller patch sizes. For the mode with increasing variance, a smaller patch size also attenuated the exponent to $0.0$ (P=7) or even flip the law i.e. $0.48$ for $P=3$. Intuitively, we think, 3x3 convolution ties many modes together in the Fourier space, so during generation, many modes would be learned at the same time.
>
> In summary, even we no longer have analytical prediction of the scaling coefficients, we can still obtain prediction through numerical simulation. It shows that the patch size of the convolution could significantly affect the scaling relation. We hypothesize that the attenuated or flat scaling relation for practical UNet trained on natural images could partially be related to its use of small 3x3 convolution.
>
> Tab.S1 Spectral convergence time computed for patch linear convolutional networks | FFHQ32
>
> |  | Increase scaling | Decease scaling |
> | --- | --- | --- |
> | P=3 | $0.06\lambda^{0.48}$ | $11.88\lambda^{-0.21}$ |
> | P=5 | $0.05\lambda^{0.24}$ | $3.66\lambda^{-0.36}$ |
> | P=7 | $0.05\lambda^{0.01}$ | $3.24\lambda^{-0.37}$ |
> | P=11 | $0.07 \lambda^{-0.22}$ | $2.61 \lambda^{-0.40}$ |
> | P=15 | $0.08\lambda^{-0.28}$ | $2.57\lambda^{-0.40}$ |
> | P=25 | $0.09 \lambda^{-0.30}$ | $2.54 \lambda^{-0.40}$ |
>
> We will add the corresponding figures to Appendix upon acceptance. If requested, we are more than happy to present the figures of these scaling laws or code generating them for reviewers to view them.

---

> > ### Comment · Reviewer_PL6U · 2025-08-05
> >
> > I thank the authors for their detailed rebuttal (W2) and hard work on the manuscript. I also appreciate their transparency regarding the claims (Q1, L1). While I still have reservations about the practical utility of linear denoisers, I recognize that the paper is of high quality regarding extensive experiment results and rigorous derivations. Therefore, I am raising my score to 4.

---

> > > ### Author Response · Authors · 2025-08-05
> > > **Thank you for raising the evaluation!**
> > >
> > > Thank you for raising your evaluation and recognizing the quality of our paper. We’ll build on this analytically tractable framework to uncover more practical insights in future work.

---

### Official Review · Reviewer_gTYo · 2025-07-02

**Clarity:** 4
**Significance:** 3
**Originality:** 3
**Rating:** 5
**Confidence:** 4

**Summary:**

The paper presents an analytical study of how diffusion models learn the data distribution over time during training. Focusing on linear denoisers trained with full-batch gradient flow, the authors derive closed-form expressions for both the model parameters and the generated distribution. They find that modes with higher variance are learned faster, leading to what they describe as a spectral bias. This inverse-variance effect is extended to deeper linear networks and linear convolutional architectures. The authors also run experiments with MLP and convolutional UNet models on synthetic and image datasets to compare against the theoretical predictions.

**Questions:**

**1.** Is there any difference between the EDM and DSM formulations, or are they equivalent in the context considered here?


**2.** Do the authors consider feasible and useful to carry a similar analysis with a random features model parametrising the score? While still linear in the parameters, the model is non linear in the data.


**3.** I believe there is a typo in Line 130: it should refer to Appendix C.4, not C.3.

**4.** In the discussion, the authors mention local convolutional structure. Could they elaborate on how one might, in principle, build a theoretical framework to handle this?

**Ethical Concerns:**

["NO or VERY MINOR ethics concerns only"]

**Final Justification:**

I thank the authors for their careful and detailed rebuttal addressing all questions and remarks. I believe that simplified solvable settings are highly valuable for providing insights into complex systems. As shown by the authors, some trends of more complicated models are captured. Therefore, I am raising my evaluation.

**Limitations:**

Yes.

**Paper Formatting Concerns:**

The paper follows the NeurIPS 2025 Paper Formatting Instructions

**Quality:**

3

**Strengths And Weaknesses:**

**Quality**

The analytical results are clearly stated under well-specified assumptions. The paper links training dynamics to a mode-wise description of learning speed, which is insightful. The experimental validation is well done, spanning both synthetic and natural image data.

A limitation is that the theory relies on idealized conditions (linear networks, full-batch gradient flow, and aligned initialization) which restricts its direct applicability to practical settings. The experiments with CNNs show deviations from the theory, which are acknowledged.


**Clarity**

The paper is generally well-written and well-structured. Assumptions are stated clearly, and the figures effectively illustrate the main points. The appendix is very detailed, and it's clear that the authors have put significant effort into it.

**Significance**

Understanding how diffusion models learn the data distribution over time is an important problem, and the paper provides theoretical insights on this front. The proposed framework gives a nice picture of how different parts of the data are learned at different speeds, which could be relevant for understanding failure modes or improving training strategies.

That said, the practical impact is somewhat limited by the idealized assumptions behind the theory, though qualitative features of the analysis are noted in more practical settings.


**Originality**

The work offers an interesting perspective on spectral bias in diffusion models, focusing on the training phase rather than the sampling process. The mode-wise view of convergence time and its link to inverse-variance scaling seems novel. Some elements of the setup, like the linear score approximation and Gaussian assumption, are standard, but the way they’re combined here leads to new observations about the learning dynamics in this context.

---

> ### Author Rebuttal · Authors · 2025-07-29
>
> We appreciate the reviewer’s positive assessment and detailed comment on our paper! Here are the response to the questions.
>
> **Q1**: The key dynamical features are the same between EDM and DSM formulations, i.e. gradient flowing along eigenmodes of data covariance or Fourier modes (convolutional case), since the two formalisms are equivalent up to scaling the state $x$ and time / noise scale (specifically if we denote $$x_{t}(\epsilon\_{t})=\sqrt{\bar{\alpha}\_{t}}x\_{0}+\sqrt{1-\bar{\alpha}\_{t}}\epsilon\_{t}$$ in DDPM formalism, then we can substitute our $$x\_{t}\mapsto\frac{x\_{t}}{\sqrt{\bar{\alpha}\_{t}}}, \sigma\_{t}\mapsto\frac{\sqrt{1-\bar{\alpha}\_{t}}}{\sqrt{\bar{\alpha}\_{t}}}$$ and translate our solution to DDPM formulation.).
>
> The difference exists in the convergence speed and the asymptotic value of the weights at each noise scale, and (potentially) weighting of each noise scales. In the appendix, we provided a comprehensive table (Tab. 2, App. C.4) of the asymptotic value $w^*$ and convergence time scale $\tau^*$ for different variants of diffusion objective and their corresponding dynamics, including EDM, the classic DSM (eps objective), and recent flow matching.
>
> We use the EDM formulation since it largely simplifies the notations in derivation: it uses a single symbol for noise scale, and the solutions we obtain can be translated to other formulations without much pain.
>
> **Q2**:  Random feature model is indeed an interesting and tractable next step, and we were planning to carry similar analysis for them.
>
> As a matter of fact, some very recent work has carried out learning dynamics analysis for diffusion models focusing on the random feature model ([1] arxiv 2505.17638). The general structure of the learning dynamics remains very similar: leveraging the *Gaussian equivalence principle*, the key factor determining the dynamics is the *covariance of the random feature* of the first layer. Tools from statistical mechanics and random matrix theory can be leveraged to calculate the limiting spectrum of these covariances, which determines the convergence speed of eigenmodes. Allbeit interesting, the theoretical analysis in this work [1], was performed for the white gaussian data case. Generalization to non-white structured covariance case is likely tractable through random matrix tools, e.g. free probability.
>
> In our camera ready version, we will add reference to this paper as concurrent work and discuss the connection and difference from our work, for readers who are interested.
>
> One major challenge for the random feature set up is that the sampling dynamics become harder to solve using current technique (Lemma 4.1), since the score function are nonlinear in the state, the PF-ODE becomes a nonlinear dynamic system. So we can no longer track the evolution of generated distributions as easily as we did in the paper.
>
> **Q3**: Thanks for mentioning the linking error, the Table on L130 should be Tab.2 and the section should be App. C.4. We will fix the typo in our final version.
>
> **Q4**: In the main paper, proposition 5.4 provided the gradient flow solution of linear convolutional model with local filter, which is basically mode by mode convergence along patch eigenmodes. To obtain the sampled distribution, we’d need to translate these learned local filter to Fourier basis and solve the PF-ODE on the Fourier basis mode by mode. This part can be done by numerical integration. For any specific dataset, we can also instantiate the learned filter as an actual linear convolutional filter and solve the PF-ODE numerically in the spatial domain.
>
> **Numerical simulation of the patch convolution learning dynamics**
> Specifically, we took the FFHQ32 dataset as our example, and used the statistics of patches cropped from it to compute the optimal linear filter at each noise scale $w_\sigma^*$. Then we used proposition 5.4 to obtain the learned filter at different training time, then solve the PF-ODE in the spatial domain to get samples. We computed their variance along eigenmodes and the scaling laws.
>
> During numerical simulation, we found there are two key factors affecting the observed power law relation between image eigenvalues and convergence time, namely, the **patch size** of the filter $P$ and the **initialization of the filter weights**.
>
> If filter weights are initialized as identity without scaling, the initial sample variance will be way too high, and the exponent will be around ~ $-0.4$ — already attenuated than the linear case solution. One practically relevant alternative is to initialize the filter weight as identity scaled by  $\sigma_{data}^2 / (\sigma^2 + \sigma_{data}^2)$, inspired by the coefficient of skip connection $c_{skip}$ in EDM preconditioning scheme.
>
> If so, we’d observe modes with both increasing and decreasing variances. The specific scaling relation is affected by the filter patch size (summarized in following table). We found that when the patch size become smaller, scaling exponent of decreasing mode approach zero. from around $-0.4$ for P=15 to around $-0.21$ for P=3 (3x3 convolution), and the convergence time generally increases with smaller patch sizes. For the mode with increasing variance, a smaller patch size also attenuated the exponent to $0.0$ (P=7) or even flip the law i.e. $0.48$ for $P=3$. Intuitively, we think, 3x3 convolution ties many modes together in the Fourier space, so during generation, many modes would be learned at the same time.
>
> In summary, even we no longer have analytical prediction of the scaling coefficients, we can still obtain prediction through numerical simulation. It shows that the patch size of the convolution could significantly affect the scaling relation. We hypothesize that the attenuated or flat scaling relation for practical UNet trained on natural images could partially be related to its use of small 3x3 convolution.
>
> Tab.S1 Spectral convergence time computed for patch linear convolutional networks | FFHQ32
>
> |  | Increase scaling | Decease scaling |
> | --- | --- | --- |
> | P=3 | $0.06\lambda^{0.48}$ | $11.88\lambda^{-0.21}$ |
> | P=5 | $0.05\lambda^{0.24}$ | $3.66\lambda^{-0.36}$ |
> | P=7 | $0.05\lambda^{0.01}$ | $3.24\lambda^{-0.37}$ |
> | P=11 | $0.07 \lambda^{-0.22}$ | $2.61 \lambda^{-0.40}$ |
> | P=15 | $0.08\lambda^{-0.28}$ | $2.57\lambda^{-0.40}$ |
> | P=25 | $0.09 \lambda^{-0.30}$ | $2.54 \lambda^{-0.40}$ |
>
> We will add the corresponding figures to Appendix upon acceptance. If requested, we are more than happy to present the figures of these scaling laws or code generating them for reviewers to view them.

---

> > ### Comment · Reviewer_gTYo · 2025-08-04
> >
> > I thank the authors for their careful and detailed rebuttal addressing all questions and remarks. I believe that simplified solvable settings are highly valuable for providing insights into complex systems. As shown by the authors, some trends of more complicated models are captured. Therefore, I am raising my evaluation.

---

> > > ### Author Response · Authors · 2025-08-05
> > > **Thanks for your support and effort in reviewing our paper!**
> > >
> > > Thank you for raising your score and for recognizing the value of our simplified, analytically tractable settings in capturing key trends of complex diffusion models. We deeply appreciate your support!

---

### Official Review · Reviewer_d8oW · 2025-07-03

**Clarity:** 3
**Significance:** 3
**Originality:** 3
**Rating:** 5
**Confidence:** 4

**Summary:**

This paper analyzes the spectral bias in denoising score matching. They analyze the learning dynamics of diffusion models, focusing on how the generated distribution evolves throughout training under some strong assumptions to ensure analytical traceability. Additionally, they also empirically test their insights extracted from the theoretical analysis in a more realistic setting. They provide both theoretical and empirical evidence for spectral bias i.e., weights converge faster along the eigenmodes or Fourier modes with high variance, and the learned distribution recovers the true variance first along the top eigenmodes.

**Questions:**

See weaknesses

**Ethical Concerns:**

["NO or VERY MINOR ethics concerns only"]

**Final Justification:**

I concur with the other reviewers that this is a decent contribution so I suggest acceptance.

**Limitations:**

Limitations are discussed in Section 7.

**Paper Formatting Concerns:**

No concerns

**Quality:**

3

**Strengths And Weaknesses:**

# Strenghts
1. This paper studies an interesting problem with a very sensible methodology. They first start by considering the simple toy setting with linear denoisers where everything is analytically tractable, and then empirically validate the results in a more realistic setting.
2. The presentation is clear. And the paper is well structured.
# Weaknesses
1. It seems that the experimental results deviate more and more from the predicted theory as we move towards the more realistic settings
 -  Even before introduction of Convolutions in Experiment 2 with MLPs on natural images the observed power-law exponent is significantly attenuated(from the theoretical prediction of -1 to -0.48)
 -  In the most realistic experiment(experiment 3) the observed power-law exponent is approximately 0, meaning that the spectral bias is absent.

If in realistic settings the purported phenomenon(spectral bias) is no longer observable then I am not sure what is the upshot of the analysis in this paper. Is it just an idiosycracy of simplistic setting studied in section 4? Or is there reason to believe that it is a more general phenomena for all diffusion models? I think a more detailed discussion connecting the analysis connecting the analytically tractable setting and the realistic setting would be highly beneficial, especially since the empirical analysis is inconclusive.

---

> ### Author Rebuttal · Authors · 2025-07-30
>
> We appreciate the reviewer’s positive assessment of our work and the insightful comment regarding the attenuation of spectral bias in more realistic settings.
>
> We have devoted additional effort and discussion to address this gap between theory and practice, through more extensive simulation of theory and ablation of Unet experiments.
>
> **On the theoretical side**, we showed that the claimed spectral bias is a general effect intrinsic to the denoising score-matching objective, analogous to ridge-regression objectives. We demonstrated this through theoretical analysis of linear, deep linear, linear-convolution, and patch-convolution networks. After submission, we have conducted additional numerical simulations of the distributional learning dynamics for patch-convolution networks (Prop. 5.4). The new results are presented in the new section below. Briefly, in general the scaling relation for linear patch convolution network is weaker, with exponent around -0.40, this shows that the patch convolution already help learning high frequency patterns more efficiently. With proper initialization of the weights, we can obtain learning dynamics closer to reality; further, the exponent strongly depend on the patch size of the filter. Smaller filter leads to generally slower convergence of lower eigenmodes and smaller scaling exponent ~$-0.21$. So this new results shows that, the observed weaker scaling relation in UNet could be related to the small filter size (3x3) of the network, and modulated by the preconditioning and initialization.
>
> **On the empirical side**, MLPs trained on Gaussian or image data exhibit power-law scaling with an exponent of about –0.4 on natural images and around –1 for Gaussian data. However, under the same setup, UNets learn all high-variance modes at roughly the same time and likewise for low-variance modes: variance-increasing modes still converge faster than variance-decreasing modes, so spectral bias persists, but the scaling exponent is effectively zero.
>
> Our current hypothesis is that **certain modern UNet design choices attenuate or circumvent the standard spectral bias**, enabling more efficient convergence to the natural-image spectrum.
> To test this, we systematically ablated various UNet features on FFHQ32—specifically, multi-level/multi-resolution design, depth, and channel width. (Without multi-level design and skip connections, a UNet reduces to a ResNet.)
> Surprisingly, we found that **channel width** is the key factor affecting the spectral signature of learning. Reducing channels from 128 to 8 or 16 lengthened convergence times and revealed a more pronounced spectral bias—consistent with the theory for patch-linear convolution networks. When channel width is held constant, UNets with 1–4 levels exhibit the same near-zero exponent; similarly, ResNets of varying depth share this signature.
>
> These findings highlight a limitation of existing theory: it typically assumes convolutional networks with few channels. Extending analyses to multi- or infinite-channel limits—perhaps via Convolutional Neural Tangent Kernel (CNTK) methods (e.g. arXiv:2203.09255)—could better capture the learning dynamics of practical high-dimensional UNets.
>
> In summary, understanding the evolution of the sample distribution during training of practical diffusion models remains challenging, and we do not claim to solve this problem in one paper. Nonetheless, our contributions are twofold: (1) solving tractable cases of diffusion learning dynamics and demonstrating their relevance to deep nonlinear MLPs, which can be relevant in non-image, e.g. tabular settings; and (2) revealing qualitative differences in practical UNet and ResNet cases. We believe that future theoretical work must incorporate design elements such as local convolution and multi-channel structure—otherwise, generic setups (e.g. two-layer MLPs, random features, linear models) common in diffusion theory (e.g. arXiv:2502.00336; 2505.24769; 2505.17638) will likely fail to account for these spectral-learning phenomena. We hope this message resonates with the theory community.
>
> ----
> **Numerical simulation of the patch convolution learning dynamics**
> We took the FFHQ32 dataset as our example, and used the statistics of patches cropped from it to compute the optimal linear patch filter at each noise scale $w_\sigma^*$. Then we used proposition 5.4 to obtain the learned filter at different training time, then solve the PF-ODE in the spatial domain to get samples. We computed their variance along eigenmodes and the scaling laws.
>
> During numerical simulation, we found there are two key factors affecting the observed power law relation between image eigenvalues and convergence time, namely, the **patch size** of the filter $P$ and the **initialization of the filter weights**.
>
> If filter weights are initialized as identity without scaling, the initial sample variance will be way too high, and the exponent will be around ~ $-0.4$ — already attenuated than the linear case solution. One practically relevant alternative is to initialize the filter weight as identity scaled by  $\sigma_{data}^2 / (\sigma^2 + \sigma_{data}^2)$, inspired by the coefficient of skip connection $c_{skip}$ in EDM preconditioning scheme.
>
> If so, we’d observe modes with both increasing and decreasing variances. The specific scaling relation is affected by the filter patch size (summarized in following table). We found that when the patch size become smaller, scaling exponent of decreasing mode approach zero. from around -0.4 for P=15 to around -0.21 for P=3 (3x3 convolution), and the convergence time generally increases with smaller patch sizes. For the mode with increasing variance, a smaller patch size also attenuated the exponent to $0.0$ (P=7) or even flip the law i.e. $0.48$ for $P=3$. Intuitively, we think, 3x3 convolution ties many modes together in the Fourier space, so during generation, many modes would be learned at the same time.
>
> In summary, even we no longer have analytical prediction of the scaling coefficients, we can still obtain prediction through numerical simulation. It shows that the patch size of the convolution could significantly affect the scaling relation. We hypothesize that the attenuated or flat scaling relation for practical UNet trained on natural images could partially be related to its use of small 3x3 convolution.
>
> Tab.S1 Spectral convergence time computed for patch linear convolutional networks | FFHQ32
>
> |  | Increase scaling | Decease scaling |
> | --- | --- | --- |
> | P=3 | $0.06\lambda^{0.48}$ | $11.88\lambda^{-0.21}$ |
> | P=5 | $0.05\lambda^{0.24}$ | $3.66\lambda^{-0.36}$ |
> | P=7 | $0.05\lambda^{0.01}$ | $3.24\lambda^{-0.37}$ |
> | P=11 | $0.07 \lambda^{-0.22}$ | $2.61 \lambda^{-0.40}$ |
> | P=15 | $0.08\lambda^{-0.28}$ | $2.57\lambda^{-0.40}$ |
> | P=25 | $0.09 \lambda^{-0.30}$ | $2.54 \lambda^{-0.40}$ |
>
> We will add the corresponding figures to Appendix upon acceptance. If requested, we are more than happy to present the figures of these scaling laws or code generating them for reviewers to view them.

---

> > ### Comment · Reviewer_d8oW · 2025-08-05
> >
> > Thank you for your clarifications. I am inclined to accept this work, so I will maintain my score.

---

> > > ### Author Response · Authors · 2025-08-05
> > >
> > > Thanks for your supporting score and your efforts in reviewing our paper!

---

### Note · Authors · 2025-08-16

We sincerely thank all reviewers for their careful reading, constructive feedback, and thoughtful engagement with our work.

All reviewers found the paper to be rigorous, clearly written, and insightful, providing a valuable analytical framework for understanding the spectral bias in diffusion model training.
While initial concerns were raised—particularly about the attenuation of spectral bias in realistic CNN/UNet settings (Reviewer `d8oW`), the restrictiveness of linear assumptions (Reviewer `PL6U`), and technical clarifications on notation and convolutional analysis (Reviewer `d6G6`)—our rebuttal directly addressed each point.
On the theory side, we clarified the equivalence between DSM/EDM formulations, discussed connections to recent random feature analyses.
More importantly, we provided numerical evidence on both theory and empirical side aiming to close the gap. We provided new numerical simulations for patch convolutional theory and observed attenuated spectral bias, and emphasized how UNet hyperparameters (e.g., filter size, channel width) modulate spectral bias. These clarifications and additional results helped contextualize the deviations from theory in realistic models and strengthened the relevance of our contributions.

Reviewers appreciated our transparency, thorough rebuttal, and the complementary role of our work alongside recent theory, with several (>=2) explicitly raising their scores after the discussion. Overall, the consensus is that our simplified but analytically tractable framework captures important trends in diffusion learning dynamics and makes a worthy theoretical contribution to the field.

We once again thank the reviewers for their supportive evaluations and constructive comments, which will help us improve the paper further!

---

### Decision · Program_Chairs · 2025-09-17

**Decision:**

Accept (spotlight)

**Comment:**

This paper develops an analytical framework for analyzing the evolution of the learned distribution during diffusion‐model training. Via Gaussian equivalence,  the authors obtain closed-form gradient-flow dynamics of weights for one- and two-layer linear and linear-convolutional denoisers, showing convergence along principal components (linear) and Fourier modes (convolutional).
It proves a pronounced spectral bias: the convergence time of each mode follows an inverse power law in that mode’s variance. The reviewers appreciate the sound theoretical framework, empirical validation mirroring the theory, and the clear exposition. The authors should incorporate the clarifications and numerical simulations provided during the discussion and update the final manuscript to reflect the thoughtful suggestions made by the reviewers.